# Moiré effect enables versatile design of topological defects in nematic liquid crystals

Xinyu Wang ®[1,4], Jinghua Jiang[2,4], Juan Chen[2,3,4], Zhawure Asilehan[2], Wentao Tang[1], Chenhui Peng ®[2] ✉ & Rui Zhang ®[1] ✉

Recent advances in surface-patterning techniques of liquid crystals have enabled the precise creation of topological defects, which promise a variety of emergent applications. However, the manipulation and application of these defects remain limited. Here, we harness the moiré effect to engineer topological defects in patterned nematic liquid crystal cells. Specifically, we combine simulation and experiment to examine a nematic cell confined between two substrates of periodic surface anchoring patterns; by rotating one surface against the other, we observe a rich variety of highly tunable, novel topological defects. These defects are shown to guide the three-dimensional self-assembly of colloids, which can conversely impact defects by preventing the self-annihilation of loop-defects through jamming. Finally, we demonstrate that certain nematic moiré cells can engender arbitrary shapes represented by defect regions. As such, the proposed simple twist method enables the design and tuning of mesoscopic structures in liquid crystals, facilitating applications including defect-directed self-assembly, material transport, micro-reactors, photonic devices, and anti-counterfeiting materials.

Liquid crystals (LCs) consist of rod- or disk-like molecules, which can self-assemble into well-defined mesoscopic structures with long-range orientational order[1]. This ordering can be locally frustrated due to topological reasons, leading to regions called topological defects[2–4]. These regions are shown to segregate foreign molecules and particles in the system, leading to defect-based applications in, for example, directed self-assembly of molecules and colloids[5], photonic devices[6], biosensing[7], and material transport[8–10]. The capability of engineering defects in LCs is important for the abovementioned applications. Existing defect manipulation methods include magnetic and electric field actuation[9,11,12], optical control[13,14], active stresses[15,16], curvature imposed by boundaries[17–19], patterned substrates[20–26], and chemical interactions[5,27]. However, the topology, morphology, and periodicity of defects these methods can engineer are often limited by the intrinsic symmetry of the system and the imposed pattern. Hence, a versatile method to manipulate defect structures, including their shapes, periodicities, and orientations, remains at large, limiting their further applications.

In recent decades, twistronics have emerged as a new field. By rotating one layer of the lattice of material on top of another to a certain angle, one obtains the so-called moiré pattern or moiré effect with emerging periodicities[28]. Recent interests in moiré-based applications are motivated by strain analysis and metrology technology[29], super-resolution imaging in biological devices[30], and van der Waals heterostructures, e.g., superconductivity[31], magnetism[32], and correlated insulators[33].

Inspired by the above advances, we hypothesize that the moiré effect can also serve as an alternative method to engineer the mesoscopic structure of a material. As a proof of concept, here, we demonstrate that moiré patterns can be used to manipulate the mesoscopic director field and the emerging topological defects in a nematic cell, which we term the "nematic moiré pattern". In the spirit

[1]Department of Physics, The Hong Kong University of Science and Technology, Clear Water Bay, Kowloon, Hong Kong, China. [2]Department of Physics, University of Science and Technology of China, Hefei, Anhui 230026, China. [3]Department of Physics and Materials Science, The University of Memphis, Memphis, TN 38152, USA. [4]These authors contributed equally: Xinyu Wang, Jinghua Jiang, Juan Chen. ✉e-mail: cpeng2@ustc.edu.cn; ruizhang@ust.hk

of the moiré effect, we consider a nematic cell confined by two surfaces with an identical, periodic pattern imposing orientational preference (namely, anchoring) to the nematic; we combine simulation and experiment to study the moiré effect of different periodic anchoring patterns. Our nematic moiré patterns can give rise to a rich variety of periodic disclination structures, including straight lines, helical-like curves, defect networks, and loops. Their three-dimensional (3D) topological structures are modelled by continuum simulations and then further confirmed by confocal microscope experiments. The geometry of these emerging periodic topological structures is sensitive to twist angles and cell gaps and reveals both low- and high-frequency modes of the geometric moiré, the latter of which are often difficult to see in conventional moiré systems. The cross-polarized optical patterns from nematic moirés contain grain- and ring-like features, which are distinct from the optical appearance of isotropic moiré patterns, namely, periodic moiré patterns arising from geometric patterns without anisotropy. The versatility of the nematic moirés is further demonstrated by the fact that a certain one-dimensional (1D) surface pattern can even form a two-dimensional (2D) lattice of disclination loops and optical patterns. Furthermore, a 2D surface pattern formed by a lattice of ±1 defects can give rise to heterogeneous defect structures corresponding to different regimes in the geometric moiré pattern.

It is known that defect lines engendered in LCs can be used to guide the self-assembly of colloidal particles[20,34,35]. This opens up possibilities for a range of interparticle and assembly behaviors, such as modifying the dynamics of defect cores in nematics[36], triggering the gelation of colloidal dispersions[37], and enhancing the thermal stability of the blue phases against external fields[38,39]. Earlier studies have thoroughly characterized various closed knot defects in nematic colloids[40–42]. In this work, we report on the entrapment of colloidal particles along line defects, akin to the entanglement defect structure (a particle attracted by a defect line) shown in[43]. The 3D defect networks generated in thick nematic moiré cells are capable of guiding 3D self-assembly and nucleation of colloidal structures. Interestingly, when a colloidal particle-laden defect loop in a nematic moiré cell undergoes shrinkage, those particles can prevent self-annihilation through jamming. Finally, we show that certain judiciously designed nematic moiré patterns can engender pixelated shapes represented by defect regions. Taken together, our proposed nematic moiré pattern offers a versatile platform to investigate the interplay of topology, geometry, and ordering in LCs and other soft materials systems, facilitates defect-based emergent applications, and opens the door for inverse design of mesoscopic structures of materials.

## Results

### A 1D cusp-like splay-bend pattern generates versatile 1D defect structures

We consider a nematic LC cell of gap size $H$ bounded by two surfaces in the $z$-direction having identical surface patterns imposing preferred orientations to the nematic as the anchoring condition. We first examine a 1D periodic splay-bend pattern of a cusp-like shape on both top and bottom substrates, resembling a 1D cosinusoidal grating pattern (Fig. 1A, B, Fig. S1)[28]. Before rotation, the two identical anchoring patterns on the two surfaces are aligned. The preferred orientation of the nematic on both surfaces can be represented by $\mathbf{n}_s = [\cos\theta, \sin\theta]$, with $\theta \equiv \theta(x) = \pi x/L$ in a Cartesian coordinate system (Fig. 1A), where $L$ is the pattern period (Fig. 1A, Fig. S1). These anchoring conditions can lead to a defect-free ground state in which the director field of the bulk nematic adopts the same orientations preferred by the two substrates.

Next, we rotate the top substrate counterclockwise with respect to the origin of the $xy$ plane by an angle $\Psi$ while fixing the bottom substrate (Fig. 1B). Upon the rotation operation, the bulk nematic is frustrated by the mismatched preferred orientations of the two substrates (Fig. 1A, B). We introduce $\theta^t(x,y)$ and $\theta^b(x,y) \equiv \theta(x)$ to

represent the preferred orientations of the top and bottom surfaces at coordinates $(x, y)$, respectively. The preferred orientation angle difference, defined as $\Delta\theta(x,y) = \theta^t(x,y) - \theta^b(x,y)$, will locally distort the nematic (Supplementary Material 2.2). Because of the preferred angle mismatch between the two substrates, there is a pointwise twist deformation in the bulk achiral nematic. The handedness of the twist is determined by the acute angle the two preferred orientations make. When this angle transitions from acute to obtuse, there will be a twist reversal, at which the bulk nematic is frustrated. Therefore, we expect that disclinations will emerge where the two preferred orientation angles are orthogonal, i.e., $\Delta\theta = m\pi/2$ with $m = \pm1, \pm3, \ldots$[44,45] The contour lines of $\Delta\theta = m\pi/2$ appear as equispaced, parallel lines in the $xy$ plane (Fig. S2).

To elucidate the nematic structure in the above patterned system, we further perform continuum simulations (Materials and Methods) by varying cell gaps $H$ while fixing pattern period $L$ and rotation angle $\Psi = 12°$. For simplification, the one-elastic-constant assumption is applied in our simulations, and a comparison with elastic constants of real materials is discussed in Supplementary Information 6.2.4 and Fig. S37. Interestingly, our simulations uncover three different types of defect structures (Fig. 1C–E, Movie S1, Supplementary Material 2.4). When $H/L < 0.3$, straight line defects (S-state) emerge, appearing as parallel and equispaced lines within the midplane of the cell; their locations and directions match well with the theoretical prediction (Fig. 1C, Fig. S2). The schematics of different local profiles of disclination lines are given in Fig. S3. When $0.3 < H/L < 0.8$, curved line defects (C-state) appear as equispaced, aligned curves, each of which resembles a 3D helix and appears as a wavy line viewed in the $z$-direction (Fig. 1D, Fig. S4). The handedness of the C-state defects can be reversed by changing the rotation direction (Supplementary Information 2.4 and Fig. S5). When $H/L > 0.8$, web-like crossing lines (W-state) emerge, consisting of two groups of equispaced, parallel defect lines located near the two surfaces, exhibiting different orientations, which appear like a web when viewed from the $z$-direction (Fig. 1E, Fig. S4). The transition of the disclinations from straight to curling shape as cell thickness increases can also be understood by making an analogy to the Peach–Koehler force of dislocations in solids under external stresses[46]. Note that as soon as the angle $\Psi$ starts to deviate from $0°$, the disclination lines in the above states come from infinity one by one with their distance decreasing with increasing $\Psi$.

Our subsequent experiments confirmed all three predicted defect structures at different cell gaps using a photopatterning system (Fig. 1I–K, Materials and Methods). Photo-alignment is a versatile tool to pattern confining substrates and stabilize both singular and nonsingular defects in LC cells[44,45,47]. The out-of-plane anchoring strength is on the order of $\sim 10^{-3}\,\text{J/m}^2$ and the in-plane anchoring energy is on the order of $\sim 10^{-4}\,\text{J/m}^2$ (strong anchoring)[48]. Based on the simulated director fields, we apply the Jones-matrix approach to obtain simulated polarized optical microscope (POM) images (Supplementary Material 6.2.5). The POM images of the three defect states also agree with the experiments. These POM images consist of parallel arrays of dark grains, the size and aspect ratio of which are different among the three defect states (Fig. S6). By contrast, isotropic moirés constructed from the cosinusoidal grating patterns appear as a similar spatial distribution of white grains (Figs. 1B, 2A and Fig. S6). The Jones-matrix approach is good for thin samples (Fig. S6A, B), while for thicker ones where focusing, ray deflection, and oblique rays become relevant (Fig. S6C), the director field cannot be seen clearly (Supplementary Information 6.2.5). These POM patterns are insensitive to the choice of the orientation of the cross polarizers (Movie S2). Although the separation distance between neighboring arrays of grains is the same for the two types of moiré patterns, the grain density in the nematic moiré is twice as dense as that in the isotropic moiré (Fig. S6). To probe the 3D defect structure in the W-state predicted by the simulation, laser scanning confocal

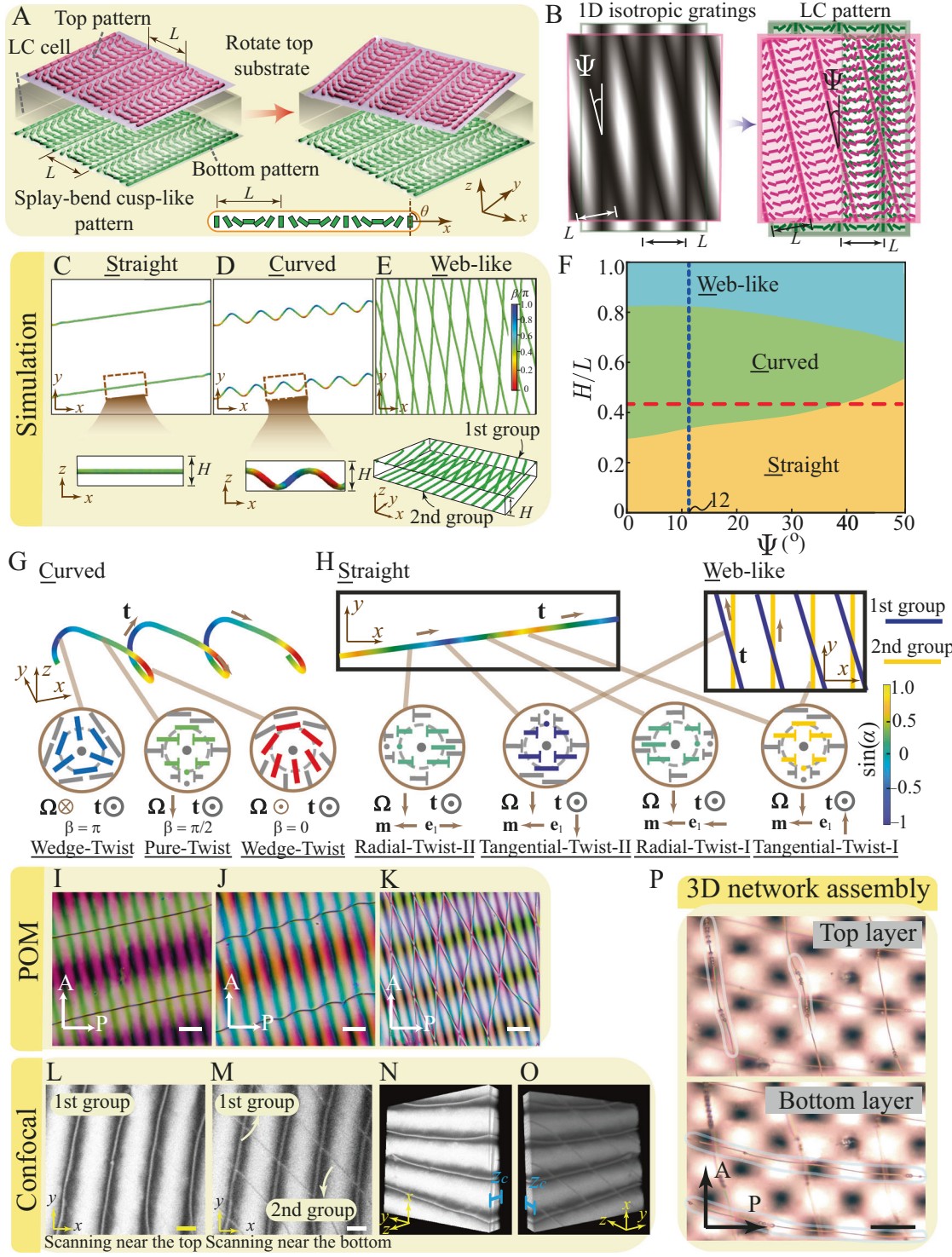

**Fig. 1 | Topological structures and 3D network colloidal assembly in a 1D splay-bend cusp-like nematic moiré pattern. A** Schematic of the system with a patterned top substrate (pink) rotated by an angle Ψ and an identically patterned, fixed bottom substrate (dark green). Short rods represent surface-preferred nematic orientations, and the geometric periodicity of the pattern is $L$. **B** Mapping from the geometric cosinsoidal pattern to the nematic surface pattern. Top and side views of the simulated defect structures at Ψ = 12° when $H/L = 0.3$ (**C**), $H/L = 0.7$ (**D**) and $H/L = 0.9$ (**E**) (colored by angle $\beta$). In (**E**), the 1st group of disclinations is near the top surface, and the 2nd group defect is generated close to the bottom surface. **F** Defect state diagram in terms of $H/L$ and Ψ, blue dotted line for Ψ = 12° and red dotted line for $H/L = 0.43$ (Fig. 2E, F). **G** Periodic, helical topological structure of the C-state (colored by angle $\beta$). **H** Topological structure of the S-state and W-state

(colored by angle $\alpha$). **I**–**K** Corresponding POM images for the S-, C-, and W-states at Ψ ≈ 12° showing good agreement with the simulation. Using confocal microscopy to scan the cell from top to bottom, we observe **L** disclination lines (the first group) close to the top substrate and **M** the disclination lines (the second group) appearing near the bottom substrate. **N**, **O** Two 3D rotation views of the sample to visualize the two groups at the same time. $z_c$ is the spacing between the two groups in the scanning process, $z_c = 40\mu m$. **N** When the two groups of disclinations are visualized from the cell top at the same time, the bright lines are the first group disclination lines, close to the top substrate. **O** Viewing the sample from the cell bottom. Both groups of lines are shown due to the scattering effect, and the second group of lines close to the bottom substrate is in gray because of lower contrast. **P** 3D colloidal assembly in the W-state. Scale bar: 50 μm.

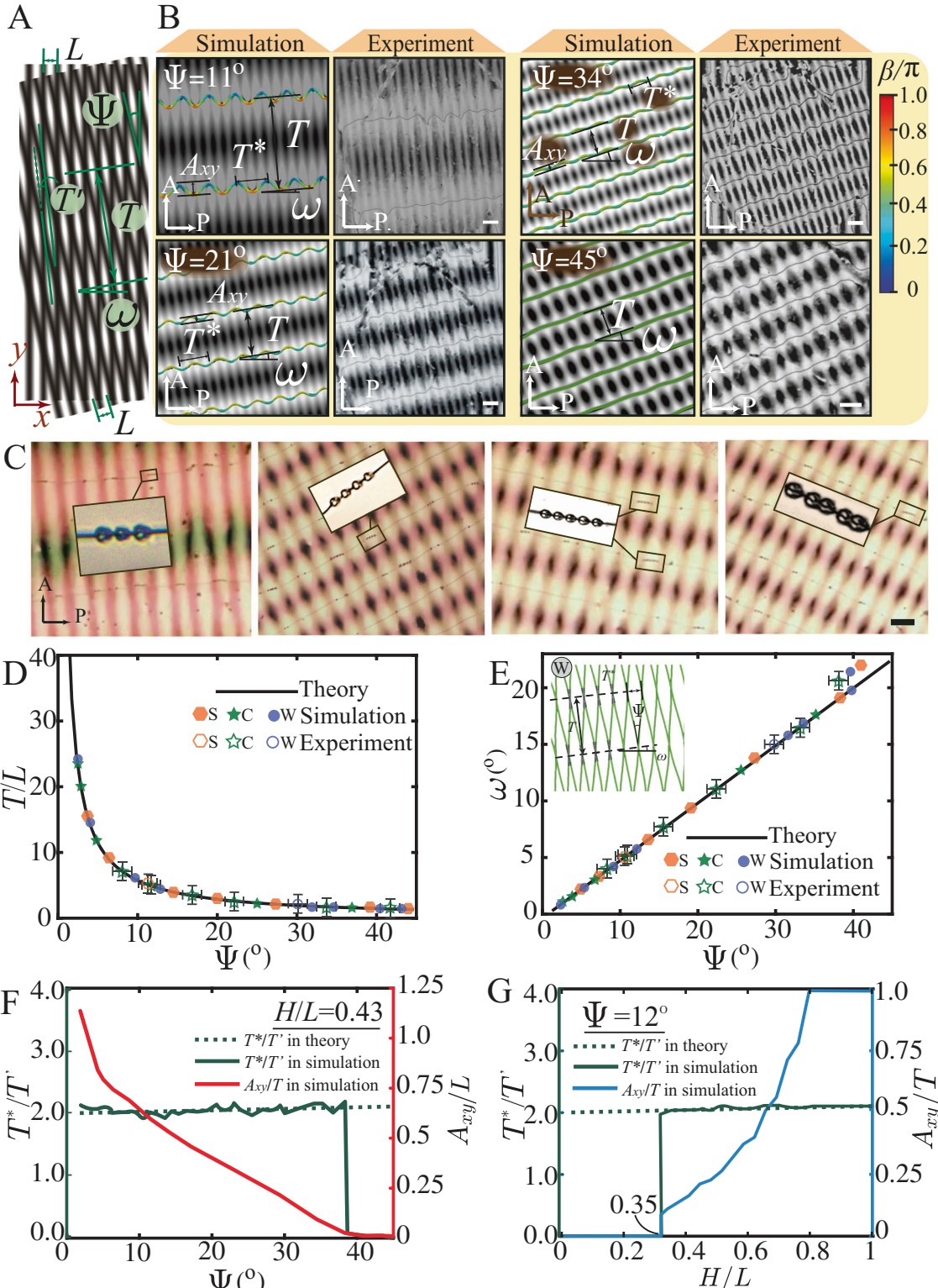

**Fig. 2 | Defect structure characterization and colloidal assembly manipulation in the 1D cusp-like anchoring pattern. A** 1D cosinusoidal geometric moiré lattice showing the $(\pm1, \mp1)$-moiré period $T$ and $(\pm1, \pm1)$-moir$\acute{e}$ $T'$. **B** Simulation and experimental results of the nematic moiré at $\Psi = 11°$, $21°$, $34°$ and $45°$. Disclinations in the simulation are colored by twist angle $\beta$. $T$ is the spacing distance for neighboring defect curves, and $\omega$ is their tilting angle with respect to the $x$ axis. The defect helical structure diameter $A_{xy}$ and pitch $T^*$ are introduced to characterize defect shapes. **C** The periodicity and orientation of the colloidal assembly by the defects can be tuned in the nematic moiré. **D** Moiré period $T/L$ as a function of rotation angle $\Psi$. **E** Moiré tilting angle $\omega$ as a function of rotation angle $\Psi$. The insert is the W-state defect. Crossings are denoted by gray lines, connected by pseudo-lines (black dashed lines). **F** $T^*/T'$ and $A_{xy}/L$ as functions of $\Psi$ while fixing $H/L = 0.43$. **G** $T^*/T'$ and $A_{xy}/T$ as functions of $H/L$ at $\Psi = 12°$. Scale bar: 50 μm. Source data are provided as a Source Data file.

fluorescence microscopy was conducted (Fig. 1L–O). Near the top substrate, there are four disclination lines (the first group) along the same direction, which are bright lines inside thick dark bands (Fig. 1L). With scanning along the $z$-axis, another four thin, gray disclination lines (the second group) appear near the bottom substrate in another orientation (Fig. 1M). Our configuration is similar to the two-layer configuration observed in ref. 49 by confocal microscopy. Therefore, when viewed from the $z$-direction (Fig. 1M), the two groups of defects appear as a web-like structure, confirming the simulation prediction (Movie S3). 3D views of the web-like structure are presented in Fig. 1N, O (Movie S4).

We further perform simulations to construct a defect state diagram by varying the gap-to-pattern ratio $H/L$ and the rotation angle $\Psi$ (Fig. 1F). Consistent with the cell gap effect, we find that the three defect states appear roughly in different ranges of $H/L$ in the diagram (Fig. 1C–E). This state diagram can be understood by considering the relative elastic energy costs of different defect states (Supplementary Material 2.3). When the cell gap is narrow (small $H/L$), the surface anchoring effect is important. The bulk nematic field represented by the orientation angle $\theta(x,y,z)$ adopts surface-preferred orientations. The anchoring conflict between the two surfaces will cause a transition from $\theta(x,y,z) \approx \theta^{b}(x,y)$ near the bottom surface to $\theta(x,y,z) \approx \theta^{t}(x,y)$ near the top surface (Fig. 1C, F). This transition can be a smooth twist in most areas. However, in certain regions where the angle difference $\Delta\theta(x,y)$ becomes the maximum value $\pi/2$, twist-winding disclinations are expected to appear (Supplementary Material 2.2). This explains the agreement between the prediction and thin-cell results. When the two substrates are away from each other (large $H/L$), the surface patterning effect becomes relatively weak and the bulk nematic dominates. Consequently, the system chooses to minimize its elastic free energy by aligning the nematic uniformly in the bulk and forming defects near the substrates to accommodate the surface anchoring pattern. This argument can well explain the emergence of line defects in the W-state: bulk nematic adopts a uniform director field; for each surface, defects should appear where the difference between the surface-preferred orientation and the bulk nematic orientation angle reaches the maximum value of $\pi/2$ (Fig. S4C). Therefore, two layers of disclinations will emerge in thick cells, and the separation distance between neighboring line defects for each layer should coincide with the periodicity $L$ of the anchoring pattern. At medium cell gaps, a delicate competition between the surface patterning and the bulk nematic leads to a complex defect structure, i.e., the C-state defect (Fig. 1D, G, Fig. S4B), the morphology of which is essentially intermediate between the parallel straight lines in the S-state and the web-like lines in the W-state, serving as a transition state between these two defect states (Movie S1). Similar to the W-state (Fig. S4C, Fig. 1C, H, I), the C-state forms a locally uniform director field in the midplane, and the defects are repelled to the two substrates (Fig. S4B).

Our further analysis of the local profiles of these emergent disclinations reveals that they are structurally different from those reported in similar systems[50] (Supplementary Material 2.4). The director profiles of these disclinations can be characterized by two angles[51,52], namely a twist angle $\beta \in [0,\pi]$ and a phase shift angle $\alpha \in [0,2\pi)$. ("Methods", Supplementary Information 2.4, Fig. S3). Pure-twist corresponds to $\beta = \pi/2$; $+1/2$ wedge winding and $-1/2$ wedge winding are characterized by $\beta = 0$ and $\beta = \pi$, respectively. A pure-twist winding with $\beta = \pi/2$ can be further characterized by the angle $\alpha$, which can distinguish between tangential-twist ($\alpha = \pi/2, 3\pi/2$) and radial-twist ($\alpha = 0, \pi$) windings[50]. Our calculations show that line defects in the S- and W-states are of the pure-twist type with $\beta \equiv \pi/2$ (Fig. 1C, E, Fig. S4). Despite the simple geometry of the line defect in the S-state, its local winding periodically varies from radial-twist-I ($\alpha = 0$), to tangential-twist-I ($\alpha = \pi/2$), to radial-twist-II ($\alpha = \pi$), then to tangential-twist-II ($\alpha = 3\pi/2$), and eventually back to tangential-twist-I ($\alpha = 2\pi \equiv 0$) profiles (Fig. 1H, Fig. S7). The angle $\alpha$ of the S-state

defect changes from 0 to $2\pi$ in a lattice period $L$: this contrasts with the line defect with uniform $\alpha$ in similar systems in which the top substrate adopts a uniform anchoring[50]. However, for the two groups of parallel defect lines in the W-state, the local director field along the curve tangent is nearly constant (Fig. S4C, top plane and bottom plane). The two groups of defects are tangential-twist-II (near top) and tangential-twist-I (near bottom) (Fig. 1H). Within the same group, all the parallel line defects share the same local profile and, therefore, the constant angle $\alpha$. In the C-state, the twist angle $\beta$ of the local winding varies along the defect curve (Figs. 1D, G, 2B, Fig. S4B): when the local winding is close to the two surfaces, $\beta$ approaches $\pi/2$ (pure-twist); when the curve passes through the midplane of the cell, the local winding is of wedge type, with $\beta$ approaching 0 and $\pi$ alternatively (wedge-twist). Therefore, the C-state defect belongs to the wedge-twist type. Distinct from 2D liquid crystals, in 3D nematics, wedge disclinations of winding number $\pm 1/2$ can be transformed through a continuous transformation, with the symmetric twist state found in between. This continuous transformation of the local profiles can also be used to explain the state transitions (between the S-, C- and W-states) as we tune the cell gap.

In Fig. 1M, the focus of the microscope is adjusted in the experiments to provide a clear view of the top and bottom parts separately. In the particular case of the defect network state (the W-state), colloids are found to assemble in both groups of disclinations (top layer and bottom layer), giving rise to a 3D configuration (Fig. 1P, Movie S5). This is potentially helpful for reversible reprogramming of colloids as building blocks to achieve multiple functions.

We also show that the geometric details of the emerging disclinations, including neighboring defect spacing $T$, defect orientation angle $\omega$, helical periodicity $T^{*}$, and projected helix diameter $A_{xy}$ (C-state), are fundamentally dictated by the geometric moiré pattern regardless of defect states (Fig. 2A, B, Supplementary Material 2.4). As $\Psi$ increases, the disclinations become closer to each other and their orientations further rotate (Fig. 2B, Movies S6 and S7). Larger $\Psi$ gives rise to more frequent variations in the anchoring angle mismatch $\Delta\theta$ in space, the disclinations therefore appear denser in the system. Both $T$ and $\omega$ can be understood by the geometric moiré pattern[28]. The theoretical values $T = L/\sqrt{2(1 - \cos\Psi)}$ and $\omega = \Psi/2$ for the first-order ($\pm 1, \mp 1$) mode in the geometric moiré pattern (Fig. S8) are quantitatively matched by those measured from all three defect states in both simulations and experiments (Fig. 2D, E). This programmable variation in disclination lines can be directly utilized to template colloidal self-assembly. For instance, when colloidal particles are added to the nematic moiré in the S-state, they tend to self-assemble into linear chains, the position and orientation of which follow the geometric features of the nematic moiré (Fig. 2C, Movie S8). Interestingly, these self-assembled colloidal chains can also impact straight-line disclinations (with a local twist profile) by inducing entangled structures (Fig. 2C insets). In an earlier work, the disclinations around particles were found to be sensitive to the cell gap[40]. Different from the closed entanglement loops around particles[34,41,42], we show the interaction between existing defect curves and particles, as particles are attracted to the straight-line disclinations (the S-state). In our case, the defect curve (S-state) in Fig. 2C has a constant pure-twist profile akin to the assembly structures in ref. 53. The projected helix diameter $A_{xy}$ in the C-state increases as $\Psi$ decreases (Fig. 2F, Movie S6) or $H/L$ increases (Fig. 2G, Movie S1). Upon the transition from the C-state to the W-state, $A_{xy}$ approaches $T$ as the neighboring wavy-like defects intersect and start to form web-like defects (Fig. 2F, G, Movie S1).

Different length scales that emerge from the defect structures stem from moiré modes of different frequencies, the high-frequency ones of which are usually difficult to see in isotropic moiré patterns (Supplementary Material 2.4). For the C-state, there are two emerging geometric parameters associated with the helical-like defects, namely the period of the helix, $T^{*}$, and its helical diameter projected onto the

$xy$ plane, $A_{xy}$ (Fig. 2B). For the W-state, $T^*$ is defined as the spacing between adjacent defect crossings on the same pseudo line. It turns out that $T^*$ in both states can be understood by the periodicity $T'$ for the $(\pm 1, \pm 1)$ mode in the geometric moiré pattern (Figs. S8, S9), as $\frac{T^*}{T'} = 2$ in both states (Fig. 2F, G). Therefore, our nematic moirés provide a simple method to tune the topology and geometry (shapes, periodicities, orientations, etc.) of the disclinations using the geometry of moiré patterns, which can further guide the colloidal assembly.

## A 1D sinusoidal splay-bend pattern generates a 2D lattice of ring defects

In the previous system, a 1D cusp-like pattern can generate 1D periodic defect structures. Here, we demonstrate that an alternative 1D pattern can generate a 2D defect lattice. Specifically, we consider a 1D sinusoidal splay-bend pattern for the two surfaces. Let $\theta(x,y) = \frac{\pi}{2}\sin(2\pi x/L)$

for both surfaces prior to rotation (Fig. 3A, B, Fig. S1). We first use the same theory to plot the predicted disclinations and find a 2D lattice of loops of alternating sizes (Fig. 3D, Supplementary Material 3.1, Figs. S10, S11). This predicted configuration agrees very well with the thin-cell simulation ($H/L = 0.1$) and experiment (Fig. 3H, Fig. S11). As the cell gap increases, both simulation and experiment show that smaller defect loops (Loop-I) shrink and eventually self-annihilate, and larger loops survive (Fig. 3E, G, Figs. S12, S13, Movie S9). Simulations also find that the surviving defect loops gradually switch from a more round-like shape (Loop-II, Fig. 3E, F, Fig. S13B) to a rhombohedral shape (Loop-III, $H/L \in [0.2, 0.6]$, Fig. 3E, G, Fig. S13C) in the planar view. This transition with an out-of-plane buckling is akin to the transition from the S-state to the C-state in the 1D cusp-like pattern (Figs. S12, S13). There is an excellent agreement between the simulated and experimental POM images (Fig. 3H, I), again underscoring the fidelity

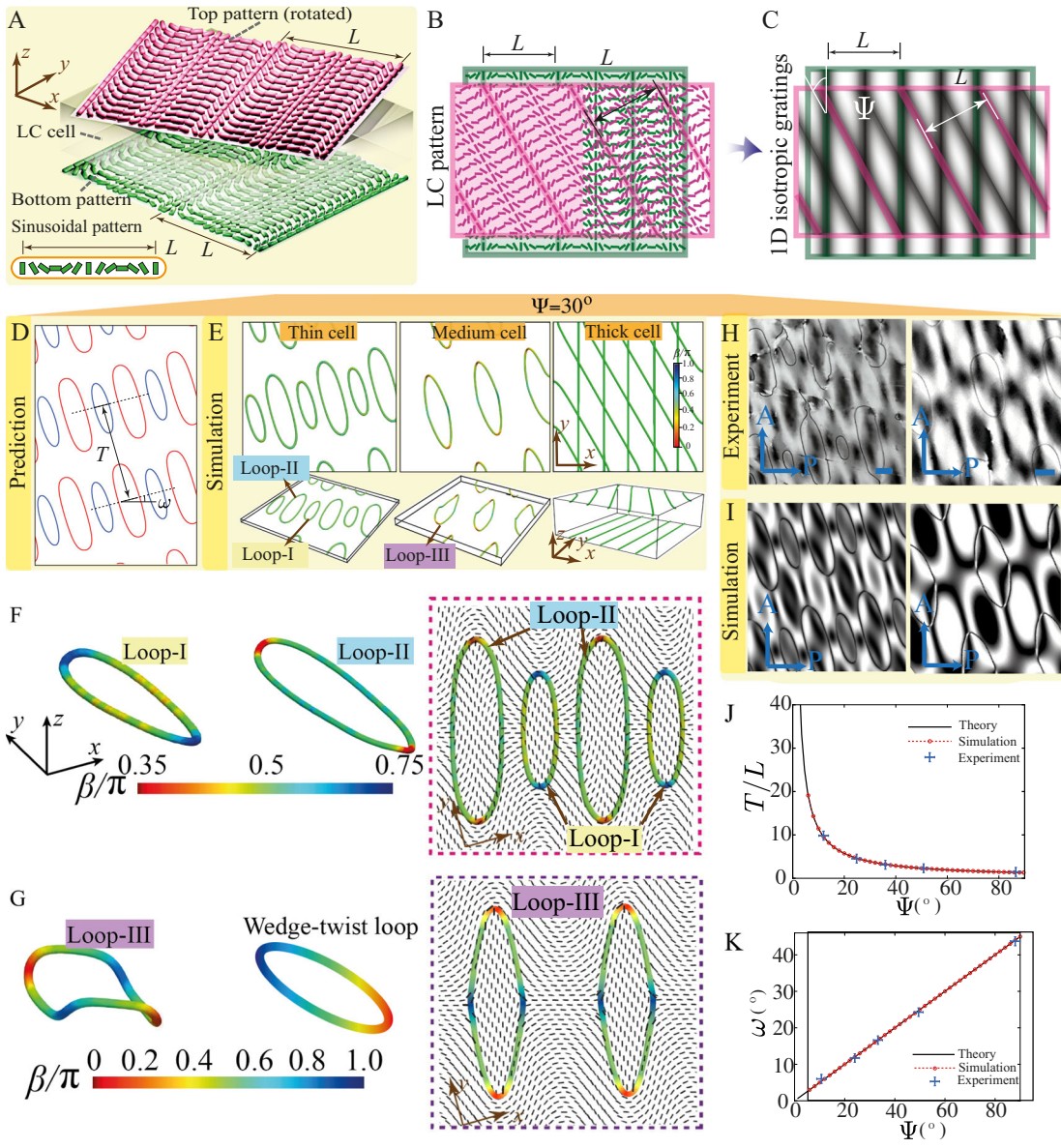

**Fig. 3 | Topological structures and optical patterns generated in a 1D sinusoidal splay-bend nematic moiré pattern. A** Schematic of the system with the dashed thick rods indicating the geometry of the pattern. The inset is the sinusoidal pattern in a period $L$. **B** Superposed surface-preferred director field of the two anchoring patterns and **C** its mapped geometric pattern with identical period $L$ and rotation angle $\Psi$. **D** Theoretically predicted defect prediction for $\Psi = 3°$, showing defect period $T$ and tilting angle $\omega$. **E** From left to right: the simulation of $H/L = 0.1$ has two groups of defect loops, $H/L = 0.3$ only has surviving larger loops, and $H/L = 1.0$ gives rise to web-like defects ($\Psi = 30°$). **F, G** A more detailed comparison of Loop-I, Loop-II, and Loop-III in the simulations and the classical wedge-twist loop. POM images (**H**) and the corresponding simulated images (**I**) for thin cells (left) and medium-gap cells (right) at $\Psi = 30°$. **J** Moiré period $T/L$ in theory, simulations, and experiments. **K** Moiré tilting angle $\omega$ in theory, simulations, and experiments. Scale bar: 50μm. Source data are provided as a Source Data file.

of the simulation method. These novel optical images consist of a lattice of dark grains and rings, in sharp contrast to that formed from the cusp-like pattern and the corresponding isotropic moiré pattern (Figs. 2B, 3C). For even thicker cells ($H/L > 0.6$), a web-like defect is formed (Fig. 3E), akin to the thick-cell results of the cusp-like pattern (Figs. S4C, S12F). As we increase the angle $\Psi$ from $0°$, the defects first enter the simulation window around $\Psi = 6°$, as shown in Fig. S14A. Then we observe one array of closed defect loops at $\Psi = 12°$ (Fig. S14B) and multiple arrays of defect loops with a distance of $T$ after $\Psi \geq 24°$. As $\Psi$ increases, the aspect ratio of the loops decreases, and the loops become rounder, as seen in the simulation and theory (Supplementary Information 3.3, Fig. S14I).

The local topological structures of these defect loops are different from the pure-twist and wedge-twist defect loops (Fig. 3G, Fig. S13) reported in driven and active nematic systems[51,54]. Along a standard wedge-twist loop (Fig. 3G), its $\beta$ profile continuously varies from pure-twist ($\beta = \pi/2$) to $+1/2$ wedge ($\beta = 0$), then back to pure-twist ($\beta = \pi/2$), to $-1/2$ wedge ($\beta = \pi$), and finally returns to pure-twist ($\beta = \pi/2$) (Fig. S3A). For thin cells and medium-gap cells, the three kinds of wedge-twist defect loops characterized by the profile of angle $\beta$ are generated in both simulations and experiments (Fig. 3D–I). Both Loop-I and Loop-II are of elliptical shape and exhibit pure-twist profiles ($\beta = \pi/2$), except for the two end regions along their long axes (Fig. 3F). Loop-I and Loop-II are also different: Loop-I and Loop-II (blue loops in Fig. 3D) exhibit negative and positive anchoring angle differences $\Delta\theta$, respectively (Fig. 3D), which implies that $\Delta\theta$ changes oppositely from the inside to the outside region of the loop (Fig. S10). Moreover, the two end regions along the major axis in a Loop-I have a $+1/2$ wedge-like profile ($\beta < \pi/2$), whereas the end regions along the long axis of a Loop-II have a $-1/2$ wedge-like profile ($\beta > \pi/2$, Fig. 3F, Figs. S12, S13). As the cell transits from a thin gap to a medium gap, a Loop-II can evolve into Loop-III, a nontrivial, distorted, 3D loop (Movie S9). When the tangent vector rotates along the Loop-III for one revolution, it goes through both the $+1/2$ wedge profile and $-1/2$ wedge profile 2 times and passes the pure-twist profile 4 times, which is also distinct from the wedge-twist loop (Fig. 3G). Note that our system uses a patterning technique and quasi-static rotation operation to manipulate topological structures, and defect loops are stable except when they are smaller than ~1200 $\mu m^2$ in the experiments. For energy reasons, the small loop gradually shrinks and eventually annihilates. The topological charges of the three kinds of loops are all zero (Supplementary Information 3.2) and they can self-annihilate if not prevented by laden particles. Since there is no more than a $\pi$ rotation from the top to the bottom patterned substrate, knot formation is topologically forbidden.

Although the spatial variation details of the two 1D splay-bend patterns (cusp-like and sinusoidal) are different, both can produce the moiré period $T$ and the tilting angle $\omega$ in terms of defect geometries and optical patterns (Fig. 3H, I, Fig. S14, Movie S10). Note that $\Psi = 90°$ is an interesting case in which theory predicts a square lattice of round defect loops. This is indeed confirmed in experiments and simulations (Fig. S15), which also identify a metastable state in which rectangle-like disclination loops with their sides along the $x$- or $y$-direction appear (Fig. S15). These defects can emerge when neighboring defect loops coalesce into large ones.

When colloidal particles are incorporated into the sinusoidal pattern system, we observe various loop-shaped colloidal assemblies in the experiment. The expansion of the disclination loops during a twist operation can be used to collect particles, which leads to loop-shaped colloidal self-assembly structures (Fig. 4A). Importantly, when a particle-laden defect loop undergoes shrinkage in the system, colloidal particles can prevent the self-annihilation of the defect through jamming (Fig. 4B). By contrast, a particle-free defect loop will disappear during the same shrinking process (Fig. 4B). We therefore expect that the interplay of dynamic disclinations and colloids in nematic moiré cells will give rise to more intriguing phenomena.

## A 2D defect lattice pattern generates hybrid disclinations

To further demonstrate the versatility of the proposed nematic moiré, we turn to 2D anchoring patterns. Specifically, we consider a surface pattern of a 2D lattice of defects consisting of two interweaving square lattices of bend-type $+1$ and $-1$ defects (Fig. 5A). This 2D pattern is similar to the dot screen pattern in isotropic moirés[28] (Fig. 5B, C). Here, we focus on a thin-film scenario with $H/L = 0.3$ (Fig. S16), in which the system forms disclinations connecting two imprinted defects from opposite surfaces at equilibrium. Before rotation, i.e., $\Psi = 0°$, each pair of same-charge surface defects at the same $(x, y)$ coordinate will be connected by two disclination lines of half-integer charge instead of one disclination line of integer charge due to elastic reason[55]; these two disclinations repel each other elastically, and they appear as loop-like defects in 3D[20] (Fig. S17A), and as shown in Fig. S17C–E, a nematic between loops and patterned surface defect core positions has depressed order and a quasiloop that connects two $+1$ defect cores and two $-1$ defect cores has $\beta \in [0, \frac{\pi}{2}]$ and $\beta \in [-\frac{\pi}{2}, 0]$, respectively (Supplementary Information 4.2).

At a small rotation angle ($\Psi = 5°$), the corresponding dot screen moiré exhibits a tilt super lattice consisting of periodically overlapping lattice points with a moiré period $T$ and orientation angle $\omega$ (Fig. 5B). The emerging defect structure in the nematic moiré is highly correlated with this super lattice, with different regimes in the unit cell engendering defects of different types and shapes (Fig. 5D, Figs. S18, S19). The loop size is small near the unit cell center and gradually increases with the distance from the center, while their windings approach to the pure-twist profile (Fig. S18). Therefore, there is a correlation between the size and the twist angle of the loops (Fig. S19). Experiments have also confirmed this finding (Fig. 5E, F). The periodicity of these defect structures and their orientation angles from both experiments and simulations match well with the geometric moiré theory (Fig. 5G, H, Fig. S20, Movie S11). At large rotation angles, $\Psi \geq 21°$, the disclinations in the nematic moiré pattern appear to be aperiodic (Movie S11, Supplementary Material 4.4, Fig. S21).

Interestingly, a periodic defect structure is observed around a special angle $\Psi = \tan^{-1}3/4 \approx 36.8°$ in the simulation (Fig. 5I), with a lattice constant $\widetilde{T}$ (different from the moiré period $T$) and orientation angle $\widetilde{\omega}$ (different from the moiré tilting angle $\omega$) (Table 1). The rotation angle $\Psi = \tan^{-1}3/4$ represents a Pythagorean triple, leading to a super lattice containing overlapping points from opposite layers (Table 1, Fig. 5I). This intriguing defect configuration was also observed in the experiment (Fig. 5J). Note that $\tan^{-1}(3/4)$ is a singular point in the moiré theory (divergent black curve in the blue region in Fig. 5G)[28] with an infinitely large moiré period $T$ (see the calculations in Supplementary Information 4.4). In the simulation and experiment, the defect structures are found aperiodic for $\Psi \in [34°, 40°]$ except $\Psi \approx 36°$ (Fig. S21). The emerging defect period $\widetilde{T}$ and angle $\widetilde{\omega}$ of the superlattice around the rotation angle $\Psi \approx 36°$ agree between the experiments and simulations and can be explained by the Pythagorean triple theory (Fig. 5I, Table 1, Fig. S22) rather than the moiré theory.

In conventional moiré patterns, the translation of one layer can lead to the translation of the emerging pattern in the orthogonal direction. This feature is also found in our 2D nematic moiré (Movie S12). We further confirm that the rotation center will not alter the defect period and tilting angle for small $\Psi$ (Fig. S24, Supplementary Material 4.6). For the special angle $\Psi = \tan^{-1}3/4$, however, the emergence of the periodic structures comes from the overlapping of defects and therefore, the defect configuration is sensitive to the choice of the rotation center (Fig. S23). Note that the defect structure is also sensitive to the rotation speed. When the rotation is in the quasi-static limit, the system shows periodic disclinations during the rotation

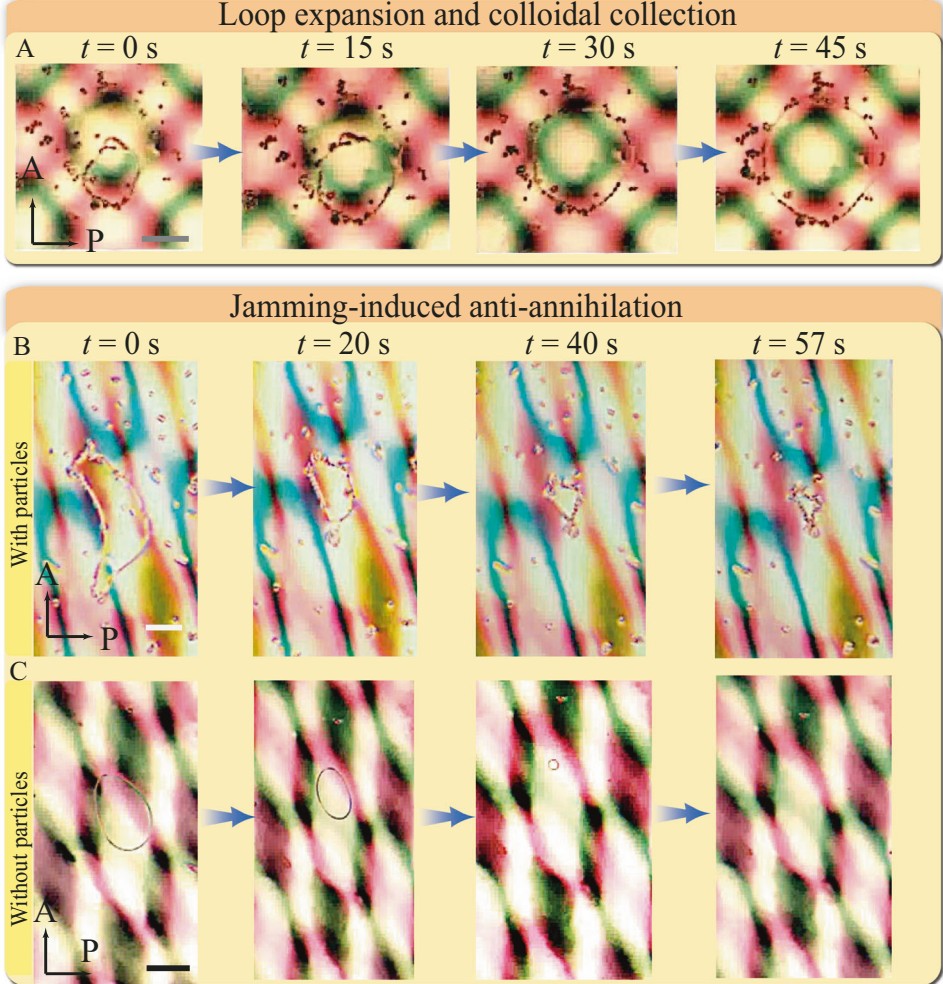

**Fig. 4 | Colloidal assembly in the 1D sinusoidal splay-bend nematic moiré pattern. A** The nucleation and expansion of the disclination loops can collect nearby particles. Ψ = 90°. **B** The colloidal assembly undergoes jamming as defect loops shrink and prevent their self-annihilation. **C** In a similar scenario without colloids, the defect loop is eventually self-annihilated. Ψ = 30°. *t* is time. Scale bar: 50 $\mu$m.

and can return to the initial state after a π-turn (Fig. S17). However, if the rotation speed is fast (Materials and Methods, Supplementary Material 6.2.2), the system will enter a disordered defect state and be stuck in a state comprised of space-filling curve defects after a π-turn, akin to the Truchet pattern (Movie S13). The hysteresis of the defect structure is rooted in the complex free energy landscape of the system. During substrate rotation, there is a competition between the surface-preferred director field and bulk elasticity. Slow rotation will render the system remaining in the free energy minimum constrained by the anchoring condition; for fast rotation, the bulk director has no time to reorient to satisfy the anchoring condition, thereby leading to a different, path-dependent director field. This also explains the rotation rate dependence of the defect structure in the 2D pattern. When the ratio is lower, there are more bulk defects than near-surface defects and the total volume of defects is lower (Fig. S25).

### Applications of nematic moiré patterns

Beyond the versatility of the proposed simple twist method, we proceed to demonstrate the applications of nematic moiré cells. For defect-based applications, existing research efforts have been devoted to the forward design problem: which topological structures with what properties can be formed from a given geometry or a given pattern. Moreover, for a given anchoring pattern, all the possible defect structures are often quite limited to a few metastable states. This

greatly limits the applications of nematic disclinations. Here, we propose using the moiré effect to address the inverse problem of designing arbitrary pixelated defect regions.

To this end, we are inspired by the application of moiré methods in the protection of documents and products[56]. Based on the moiré intensity profiles, new patterns are generated between two specially designed periodic dot screens, one of which is a microstructured image that is located on the document itself (bottom, green layer, Fig. 6A), while the other (top, pink layer, Fig. 6A) plays the role of a revealer. In the design of anti-counterfeiting materials, the superposed new pattern of a highly visible repetitive moiré pattern of a predefined intensity profile shape and color gives the authenticated properties, documents, or products. Here we use simulations to demonstrate that a judiciously chosen nematic moiré pattern can be used to engender arbitrary shapes represented by defect regions. Specifically, we design a circular pattern for the top substrate and a Latin-letter pattern (U, S, or T) for the bottom substrate (Fig. 6B). Combining the understanding that disclination lines appear where the twist angle of the pattern director has a 90° difference in a thin cell[44,45], we propose an inverse design possibility and map the "document" pattern (bottom) and "revealer" pattern (top) to LC patterns. At a small twist angle Ψ = 4°, an amplified letter region of topological defects much larger than the periodicity of the pattern emerges, which consists of many defect loops (Fig. 6C). Therefore, we anticipate that nematic moiré cells can

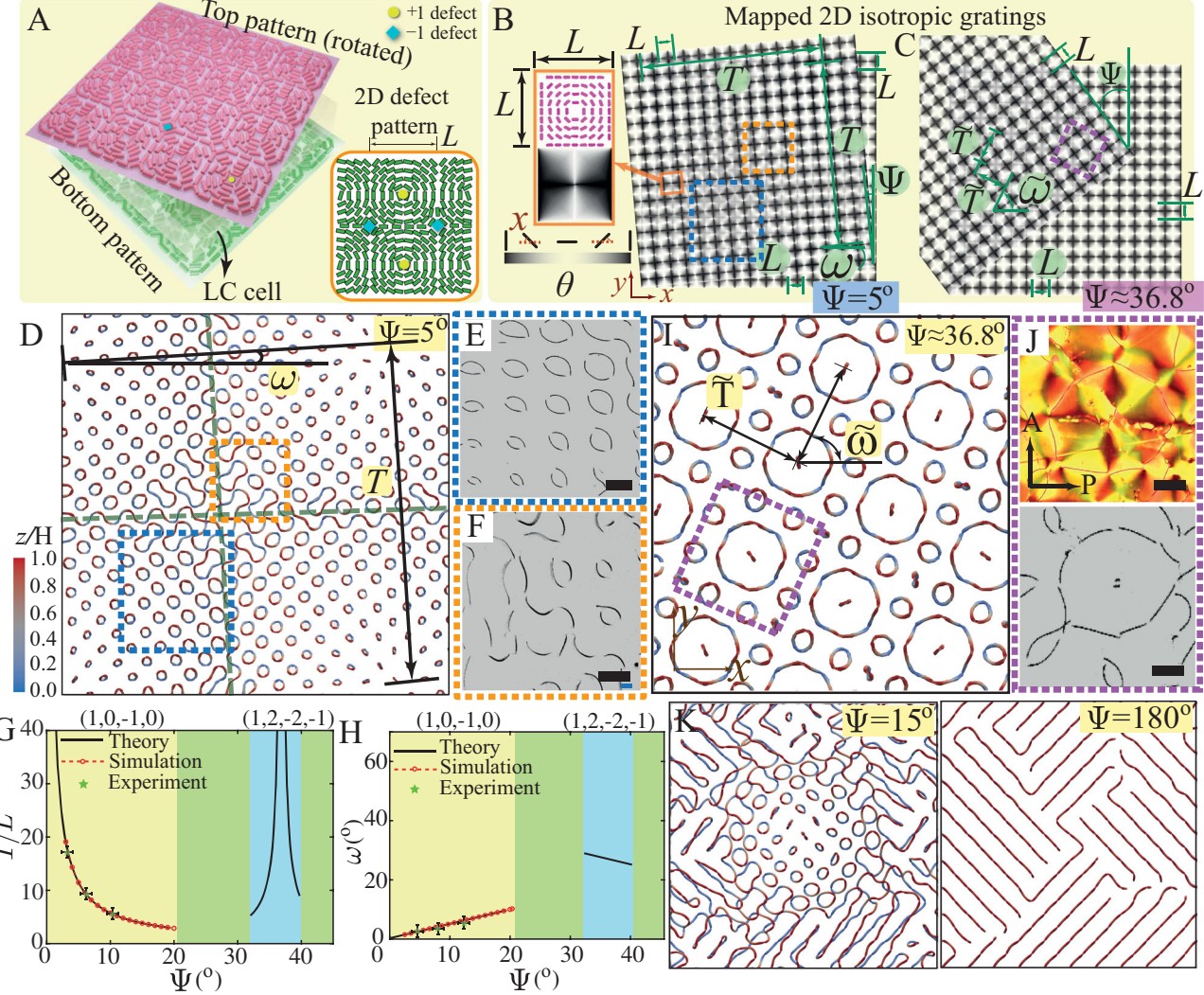

**Fig. 5 | 2D pattern defect structure characterization. A** Schematic of the system with the patterned top substrate in pink and the bottom substrate in dark green. The inset is a schematic of the 2D pattern. **B** The mapping from the dot-screen pattern to the $\pm 1$ defect lattice pattern (left) and the superposed 2D geometric square gratings formed at $\Psi = 5°$. **C** Superposed 2D geometric moiré pattern at $\Psi \approx 36.8°$. **D** Simulated defect structure in the nematic moiré at $\Psi = 5°$. **E**, **F** Experimental bright field snapshots corresponding to two boxed regions in the dashed box in (**B**, **D**). **G** Superposed period $T/L$ from geometric moiré pattern theory, simulation, and experiment. **H** Superposed tilting angle $\omega$ from geometric moiré pattern theory, simulation, and experiment. Yellow is for the $\Psi \le 21°$ regime

with $(1, 0, -1, 0)$ as the dominating moiré, and blue is for the $\Psi \in [34°, 40°]$ regime with dominating moiré $(1, 2, -2, -1)$, with a singular point (infinitely large period) at $\Psi = \tan^{-1}(3/4)$. For the other angle ranges, $\Psi \in [21°, 34°]$ and $\Psi > 34°$, no dominating moiré is observed (using green). **I** Nematic moiré in simulation for $\Psi \approx 36.8°$. Its period and tilting angle are $\tilde{T}$ and $\tilde{\omega}$. **J** Experimental coloured POM snapshots (top) and bright field image (bottom) of a periodic defect structure at $\Psi = 36.8°$ corresponding to the boxed region in (**C**). **K** Fast rotation simulation snapshots at $\Psi = 15°$ (left) and $\Psi = 180°$ (right) with the simulation time ratio being $\tau_s/\tau_{cell} = 35.4$ ("Methods"). Scale bar: 50μm. Source data are provided as a Source Data file.

generate various programmed shapes and can potentially be used for inverse design, printing, self-assembly, photonics, and anti-counterfeiting materials[57].

Lastly, we study how nematic moirés respond to external electric or magnetic fields. A uniform nematic cell can undergo the so-called Frederiks transition if an electric field applied normal to the cell

surface is above a threshold value[1]. Twisting can lower the threshold voltage (Supplementary Material 5.1). For comparison purposes, we choose $\Psi_0 = \Psi = 12°$ for both twisted nematic cells (TNCs) and nematic moiré cells. To characterize the transition in nematic moiré cells, we introduce a tilting angle $\phi$ to represent the angle between **n** and the $xy$ plane. The measured $\bar{\phi}$ from both uniform cells and TNCs in the simulations agree well with the above theory (Supplementary Material 5.2). We consider the 1D cusp-like pattern and compare the average tilting angle $\bar{\phi}$ in the midplane of the cell as a function of the applied voltage $V$ for the three defect states with the same cell gap $H$ in the simulation (Fig. S26). Similar to the effect of twisting in planar cells, spatial distortions of the nematic imposed by the surface patterning can also facilitate the onset of the Frederiks transition (Fig. S26B, Movie S14, Supplementary Material 5.2, 5.3). Indeed, the nematic moiré pattern exhibits a threshold voltage approximately 10% lower than that

**Table 1 | 2D defect pattern disclination period and tilting angle ($\Psi = \tan^{-1}(3/4)$)**

|  | Pythagorean Theory | Simulation | Experiment |
|---|---|---|---|
| $\Psi$ | $\approx 36.87°$ | $36.2° - 37.4°$ | $36.2° \pm 2°$ |
| $\tilde{T}/L$ | $\sqrt{5}$ | $\approx 2.24$ | $2.2 \pm 0.1$ |
| $\tilde{\omega}$ | $65°$ | $65°$ | $65° \pm 2°$ |

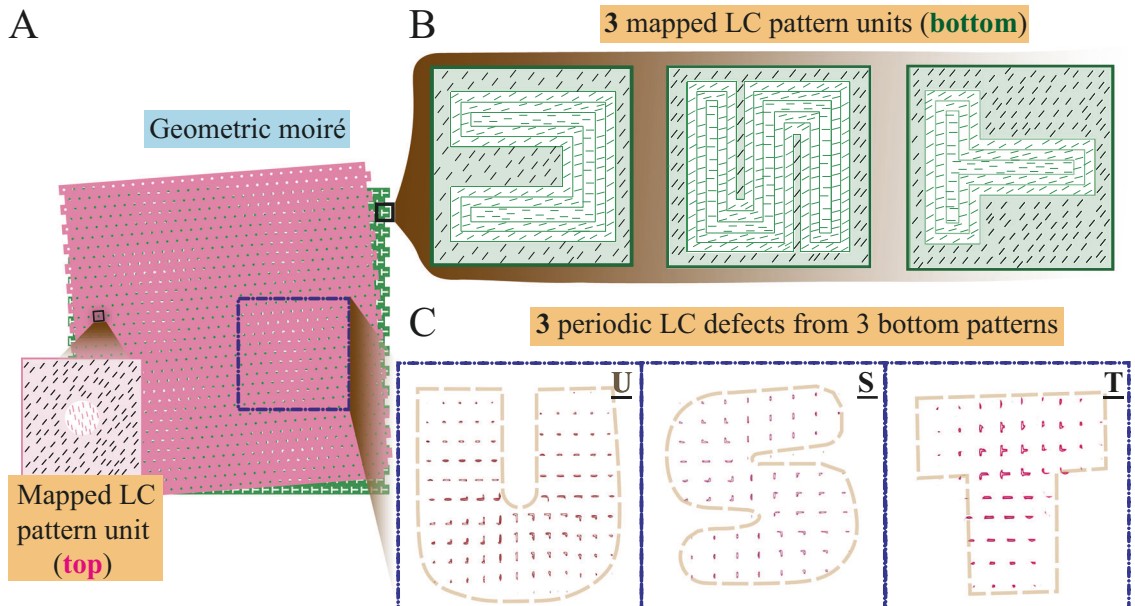

**Fig. 6 | Simulation design of on-demand shapes of defect-region using nematic moiré.** Inspired by geometric moiré, using alternate surface patterns, various Latin letter-shaped defects following the moiré period and tilting angle can be generated. **A** By putting a "revealer" pattern (circular) on top of a "document" pattern (T-shape), an array of amplified T shapes can be revealed in the geometric moiré. One of the generated T shapes is noted by a dashed blue frame. The inset is the circular LC pattern mapped from the circular geometric pattern. **B** Map the "document pattern" at the bottom to the LC pattern (in the shape of U, S, and T). **C** Results of three simulations for $\Psi = 4°$: Overlapping the circular LC pattern on the top and the three Latin letter LC patterns on the bottom sequentially, pixelated regions in the shape of U, S, and T comprising defect loops are generated in the three simulations.

of TNCs, showing the promise of nematic moiré patterns in applications such as displays and responsive materials.

## Discussion

A versatile method to engineer the mesoscopic structure of a material is crucial for its applications. As an important mesoscopic structure, topological defects are ubiquitous in various physical systems and are pivotal in understanding the bulk properties and phase transitions of materials[58–60]. In this work, we propose to harness the moiré effect to achieve a high degree of tunability of this mesoscopic structure in nematic LCs. To demonstrate, we combine continuum simulations and experiments to examine a nematic cell confined by two surfaces with identical, periodic anchoring patterns. Our simulation is a numerical solver of the Ginzburg–Landau equation based on the Landau–de Gennes free energy functional[61]; our experiment adopts a maskless photopatterning technique to control the spatially varying anchoring of the LC from the micrometer to centimeter scale[50,62]. We first consider a one-dimensional (1D) splay-bend anchoring pattern. Before rotation, the two surface anchoring patterns are aligned, and the nematic is in a defect-free state. Upon twisting one surface against the other, we observed a periodic defect structure, the periodicity and shape geometry of which can be explained by the corresponding geometric moiré pattern. The type, shape, and dimensionality of these defect structures are highly tunable and sensitive to the rotation angle and the cell gap. Based on the preferred orientation difference between the two surfaces, a simple theory can correctly predict these disclinations in the thin-cell limit but fails for thicker cells. The simulations and experiments agree very well not only on defect configurations but also on optical structures (POM images), which appear distinct from the optical images of the corresponding isotropic moiré patterns. Importantly, for the 3D defect network structure, the confocal microscope experiment further confirms the simulation prediction. The engendered defect networks can be used to guide the 3D self-assembly of colloidal particles. Conversely, these particles can also alter the defect structure by preventing self-annihilation of loop defects through jamming.

We further investigate a 2D anchoring pattern consisting of a lattice of ±1 surface defects. The emerging topological structure corresponds to the different regimes in the super lattice of the geometric moiré. Interestingly, when the rotation angle coincides with a Pythagorean triple, a rotation center-sensitive, highly periodic defect pattern emerges. A similar photopatterning technique has been adopted in the literature[20,49,63,64], but the rotation operation over two overlapping periodic patterns was not investigated. We also find that the resulting nematic structure is sensitive to the rotation speed: slow rotation can bring the system back to the ground state after a π-rotation, whereas fast rotation can drive the system into disordered defect states, which are stuck in a state comprised of surface defects.

Finally, we demonstrate two applications of nematic moiré cells. First, a judiciously chosen nematic moiré pattern can give rise to arbitrary shapes represented by defect regions. We anticipated that our system could be used for anti-counterfeiting materials[65] and photonic devices[66]. Additionally, our calculations show that the Frederiks transition voltage of the nematic moiré is ∼ 10% lower than that of a planar twist cell with the same twisting angle, which can facilitate display-related applications. This can be understood by the finer elastic distortions incurred by the anchoring pattern.

The proposed nematic system equipped with the moiré effect shows the following features and advantages: (1) distinct periodic topological structures with highly tunable periodicities can be systematically realized in one system by a simple twist method, and their geometries can be well understood by the moiré theory-based analytic model, which can enable the inverse design of these structures; (2) the unique POM images of nematic moirés could enrich their applications in microscopic imaging and strain analysis; and (3) the interplay between the engendered disclinations and colloidal particles contains rich physics and can lead to emerging mesoscopic patterns. If incorporating the nematic moiré effect in the ongoing

defect-based application research, we can foresee much richer applications of the moiré effect in liquid crystals, soft materials systems and beyond.

## Methods

### Patterned surface alignment

The nematic liquid crystal 4′-pentyl-4-cyanobiphenyl (5CB) was purchased from *Jiangsu Hecheng INC*. and used without purification in this experiment. JXLC20000 is a commercialized nematic mixture purchased from *Grandinchem, INC* (Fig. S30). The photosensitive material azo dye SD1 (Fig. S30) was purchased from *DIC INC*. and used without further purification. SD1 was mixed with n,n-dimethylformamide (DMF) solvent at a 0.2 wt% concentration. Glass substrates were washed in an ultrasonic bath with Cavi-clean detergent and then with isopropyl alcohol and dried in an oven at 80 °C for 15 min. Subsequently, the substrates were placed in an ozone chamber under UV exposure for 5 mins. The SD1 solution is spin-coated on the glass substrates at 3000 rpm for 30 s. The glass plates were baked at 95 °C for 20 min. The photosensitive azo dye on the glass substrates is sensitive to and will be oriented perpendicular to the irradiated linearly polarized light[67]. The LC director is aligned with the orientations of azo dye molecules.

The topological pattern is created by using a maskless photo-patterning setup based on a projector display[50,62,68,69], Fig. S27A. To produce the complex director pattern, for example, a director field $\mathbf{n} = [n_x, n_y] = [\cos\theta, \sin\theta]$, where $\theta(x,y) = m\tan^{-1}\left(\frac{y}{x}\right) + \theta_0$, $m = +1$ is an integer topological charge, and the phase $\theta_0 = \pi/2$ sets the distortion with a pure bend (Fig. S27B), this pattern is divided into 36 segments (Fig. S27C). One segment with an open angle of 10° is taken and set to both the segment and linear polarizer along the x-axis, as shown in Fig. S25D. The linearly polarized light can pass through the white segment, while light from all other directions is blocked by the dark background (Fig. S27D). Then, the rotation speed of the polarizer is $R_1$, and the segment is rotated at a speed of $R_2$. The rotation of the polarizer is controlled by a rotary motorized stage, and the rotation of the segments is controlled by the computer. Both rotations are synchronized by a homemade LabVIEW program. If a circular pattern shown in Fig. S27B is needed, $m = \frac{R_1}{R_2} = 1$. Hence, the polarizer is rotated at speed $R_1 = 10°/10s$ to have a 10 s exposure time for each step. Meanwhile, the time step between two segments is 10 s. After a full $2\pi$ rotation of segments, the circular pattern will be created, as shown in Fig. S27E, F. A two-dimensional (2D) lattice of topological defects can be produced by controlling the distance between adjacent defects. The produced 1D cusp-like splay-bend pattern is shown in Fig. S28. Thus, the polarization pattern of light is imprinted into the photosensitive substrate that is used to align the liquid crystal. Please note that this maskless patterning technique by projecting display can produce any pattern of the director field with the typical scale of spatial gradients ranging from approximately tens of micrometers to centimeters.

### Sample preparation

To reduce the influence of light irradiation on the patterned substrate, an additional layer of liquid crystal polymer is coated on the top of the pattern. Monomer RM257, Fig. S26, purchased from *Wilshire*, is mixed with toluene at a concentration of 7 wt% with photoinitiator Irgacure 651 (from Ciba, Inc.) at a concentration of 5 wt% RM257. This solution was spin-coated onto the patterned SD1 substrates at 3000 rpm for 30 s. The substrates were photopolymerized under unpolarized ultraviolet light with an intensity of 1.4 mW/cm² for 30 min. The polymer pattern replicates the pattern of SD1 alignment beneath it.

Two substrates with the same topological patterns are assembled, and the gap is set by 10 – 50 $\mu m$ glass spacers. Specifically, in the experiments involving a 1D cusp-like splay-bend pattern, a 10 $\mu m$ spacer was used for the S-state experiments, a 30 $\mu m$ spacer was used for the C-state experiments, and a 50 $\mu m$ spacer was used for the W-state experiments. In the experiments of the 1D sinusoidal splay-bend pattern, a 10 $\mu m$ spacer was used for the thin cell demonstration and a 30 $\mu m$ spacer for the thick cells. In addition, a 10 $\mu m$ spacer was used in the experiments using a 2D defect lattice pattern. 5CB is injected into the cell by capillary force at 45 °C, which corresponds to the isotropic phase of 5CB. After the sample is cooled to room temperature, the disclination lines form, and the sample will be imaged by polarizing optical microscopy and bright-field optical microscopy. The top substrate is rotated manually to the set rotation angle Ψ. The birefringence for JXLC-20000 is approximately 0.05, and the birefringence for 5CB is approximately 0.2. Hence, the color of the images is different. Both 5CB and JXLC-20000 liquid crystal materials are used in the experiments. JXLC-20000 is used in Fig. 1, and 5CB is used in the rest of the experiments.

5CB is doped with 0.01 wt% silica colloids of radius 2.5 $\mu m$ (*Cospheric Inc.*). Colloids treated with octadecyl-dimethyl-(3-trimethoxysilylpropyl) ammonium chloride (DMOAP) produce perpendicular director alignment and dipolar structures with a hyperbolic hedgehog on one side of the sphere accompanying a spherical particle with normal anchoring[67]. The colloidal dispersion in the LC is injected into the photopatterned cell at room temperature (22 °C). The colloids are manipulated by using a laser tweezers (*JCOPTIX*, China).

### Optical microscopy

We used a 50X-1000X Advanced Upright Polarized light Microscope from *Amscope* with both a 10x Plan, NA = 0.25 objective and 20x Plan, NA = 0.40 objective. Optical microscopy images were captured by a 20 MP USB3.0 BSI C-mount Microscope Camera from *Amscope*. (resolution 5440 × 3648 pixels). The bright-field optical micrographs were taken by a 40X-1000X Upright Fluorescence Microscope with Rotating Multifilter Turret from *Amscope* with both 10x Plan, NA = 0.25 objective and 20x Plan, NA = 0.40 objective.

### Laser scanning fluorescence confocal microscopy

The anisotropic fluorescent dye N,N′-bis(2,5-di-tert-butylphenyl)-3,4,9,10-perylenedicarboximide (BTBP) (from *Sigma–Aldrich*) was mixed in methanol at 0.01 wt%. Due to the low birefringence of LC JXLC-20000, it is used in the confocal measurement to reduce the scattering effect during the imaging process. The BTBP solution is then mixed with JXLC-20000 at a weight ratio of 1:1. Then, methanol was evaporated overnight on a hotplate at 90°C. This LC is filled in the prepared cell. Imaging was performed with a Nikon A1 laser scanning confocal fluorescence microscope using lasers with excitation wavelength of 488 nm and emission wavelength of 530 nm.

### Theoretical model of numerical simulations

The director field (local orientation) of a nematic LC can be represented by a double-headed vector $\mathbf{n}$ ($\mathbf{n} - \mathbf{n}$). The tensorial nematic order parameter $\mathbf{Q}$ is given by $Q_{ij} = S(n_i n_j - \delta_{ij}/3) + \frac{P}{2}(e_i^1 e_j^{(1)} - e_i^{(2)} e_j^{(2)})$, where $S$ is the scalar order parameter of the nematic, $P$ is the degree of biaxiality, $\mathbf{e}^{(1)}$ is the secondary director perpendicular to $\mathbf{n}$, and $\mathbf{e}^{(2)} = \mathbf{n} \times \mathbf{e}^{(1)}$[61]. Our simulation is based on the Landau–de Gennes free energy functional[1,70] in terms of $\mathbf{Q}$. Infinite anchoring is assumed in the simulation. The one-elastic-constant assumption is adopted, and the length unit in the simulation is set to the nematic coherence length (the defect core size), $\xi_N = \sqrt{L_1/A_0}$, where $L_1$ is the elastic constant and $A_0$ is the energy scale of the nematic. In the simulations, we neglect flow effects and focus on the thermodynamic relaxation of the nematic order. The simulation box dimensions are shown in Fig. S33, and considering the boundary effect, the results are shown in a slightly cropped box (Supplementary Information 6.2.1). We apply two

substrate patterns and let the system relax to equilibrium. The total free energy $F$ consists of three bulk terms:

$$F = \int_V (f_{\text{LdG}} + f_{\text{el}} + f_{\text{E}}) dV, \tag{1}$$

where $f_{\text{LdG}}$ is the Landau−de Gennes free energy, $f_{\text{el}}$ is the elastic energy, and $f_{\text{E}}$ is the electric field-induced free energy. $f_{\text{LdG}}$ has the form[1]

$$f_{\text{LdG}} = \frac{A_0}{2}\left(1 - \frac{U}{3}\right)\text{Tr}\left(\mathbf{Q}^2\right) - \frac{A_0 U}{3}\text{Tr}\left(\mathbf{Q}^3\right) + \frac{A_0 U}{4}\text{Tr}\left(\mathbf{Q}^2\right)^2, \tag{2}$$

where parameter $A_0$ is the energy density scale, and parameter $U$ controls the magnitude of $S_0$ of a homogeneous static system through

$$S_0 = \frac{1}{4} + \frac{3}{4}\sqrt{1 - \frac{8}{3U}}. \tag{3}$$

$U$ is set to 3.5, and the equilibrium scalar order parameter $S_0 \cong 0.62$ in the simulation. Defect regions are described by an iso-surface of $S = 0.45$. Using the above Doi expression for $f_{\text{LdG}}$, the biaxiality $P$ is practically 0 in defect-free regions[71–73].

The elastic energy density $f_{\text{el}}$ in our simulations is expressed in terms of the $\mathbf{Q}$-tensor form as

$$f_{\text{el}}^Q = \frac{1}{2}L_1\left(\partial_k Q_{ij}\right)\left(\partial_k Q_{ij}\right) + \frac{1}{2}L_2\left(\partial_k Q_{jk}\right)\left(\partial_l Q_{jl}\right)$$
$$+ \frac{1}{2}L_3 Q_{ij}(\partial_i Q_{kl})\left(\partial_j Q_{kl}\right) + \frac{1}{2}L_4\left(\partial_l Q_{jk}\right)\left(\partial_k Q_{jl}\right). \tag{4}$$

For simplicity, we use a one-constant assumption without special notation (Supplementary Material 6.2.1). The electric field-induced energy density takes the following form:

$$f_E = -\frac{1}{2}\epsilon_0 \epsilon_a \mathbf{E} \cdot \mathbf{Q} \cdot \mathbf{E}, \tag{5}$$

where $\mathbf{E}$ is the electric field and $\epsilon_a$ is the molecular dielectric anisotropy, $\mathbf{Q}_s$ is the patterned surface field defined as $\mathbf{Q}_s = S_0(\mathbf{n}_s \mathbf{n}_s - \frac{\mathbf{I}}{3})$ with $\mathbf{n}_s$ the imposed surface orientation and is fixed. To minimize the thermodynamic potential in Eq. 1, we write a molecular field,

$$\mathbf{H} = -\left[\frac{\delta F}{\delta \mathbf{Q}}\right]^{\text{st}}, \tag{6}$$

where $[...]^{\text{st}}$ is a symmetric and traceless operator. The evolution for bulk points is governed by

$$\partial_t \mathbf{Q} = \Gamma_s \mathbf{H}, \tag{7}$$

where $\Gamma_s$ is a relaxation constant and is related to the rotational viscosity $\gamma_1$ via $\Gamma_s = 2S_0^2/\gamma_1$[74,75], and $t$ is time. Infinite (fixed) anchoring condition is applied to approximate the strong anchoring condition in the experiment[76]. The surface points in the simulation are set to have constant equilibrium bulk order parameter $S_0 \cong 0.62$. We monitor the free energy as a function of simulation time (Fig. S34) and do not find any divergence issue (Supplementary Information 6.2.1).

There are two characteristic time scales associated with the nematic. One is the nematic relaxation time $\tau_{\text{LC}} = \frac{\gamma_1 \xi_N^2}{L_1}$ with $\gamma_1$ defined in the above. The other time scale is the relaxation time associated with the nematic cell[74]: $\tau_{\text{cell}} = \frac{\gamma_1 H^2}{L_1} = \frac{2S_0^2 H^2}{\Gamma_s L_1}$, with $H$ being the cell height. In simulation units, $\tau_{\text{LC}} \approx 7.7$ and $\tau_{\text{cell}} \approx 623$. The simulation time step $\Delta t = 0.5$ is more than ten times smaller than $\tau_{\text{LC}}$ and we set the total

simulation time $\tau_s \gg \tau_{\text{LC}}$ to ensure that our simulation can converge to free energy minimum, where $\tau_s = \Delta t \times N$ and $N$ is the total number of simulation steps. For rotation operation simulations, we choose the total simulation time $\tau_s \gg \tau_{\text{cell}}$ to approximate a quasi-static process. We have also checked the nematic structure during rotation if $\tau_s \gg \tau_{\text{cell}}$ is not satisfied (Movie S13).

The simulation is conducted via a finite difference approach, as described in ref. 61. Following ref. 61, our simulation can be mapped to the nematic 5CB at room temperature and atmospheric pressure by choosing $\xi_N \approx 6.63$ nm, $A_0 = 1.17 \times 10^5$ J/m$^3$, $\gamma_1 = 0.078$ Pa·s, and $\epsilon_a = 11.5$. This gives rise to $\tau_{\text{LC}} \approx 0.667$ $\mu$s and $\tau_{\text{cell}} \approx 54$ $\mu$s. In the simulation, surface pattern period $L = 30$ in simulation units, corresponding to $\sim 199$ nm, which is compared to 75 $\mu$m in the experiment. Despite the system size mismatch, the simulation and experiment agree very well.

### Topological analysis of disclination lines

For each curve, we define vector $\mathbf{t}$ as its tangent, and $\boldsymbol{\Omega}$ is the rotation vector for each director profile along the curve. $\mathbf{m}$ is normal to both $\mathbf{t}$ and $\boldsymbol{\Omega}$. $\beta$ is the angle between $\boldsymbol{\Omega}$ and $\mathbf{t}$, which ranges in $[0, \pi]$. If $\beta = \pi/2$, the winding type of a local profile is pure-twist; if $\beta = 0$ or $\pi$, a defect curve point is claimed to be the wedge-twist type, $\beta = 0$ being the $+1/2$ wedge and $\beta = \pi$ being the $-1/2$ wedge[2,50,51]. The schematics of angle $\beta$ are presented in Fig. S3A. Write $\mathbf{n}$ in terms of $\mathbf{t}$ and $\mathbf{m}$,

$$\mathbf{n} = \cos\frac{1}{2}\varphi \mathbf{m} + \sin\frac{1}{2}\varphi(\cos\beta \mathbf{t} \times \mathbf{m} + \sin\beta \mathbf{t}), \tag{8}$$

where $\varphi$ is the azimuthal angle of the local profile. Furthermore, pure-twist can be characterized as angle $\alpha$ in another triad $[\mathbf{t}, \mathbf{e}_1, \mathbf{e}_2]$[52]. Vector $\mathbf{e}_1$ is on the plane that is normal to $\mathbf{t}$, pointing from the defect core center along $\varphi = 0$, and $\mathbf{e}_2$ is written as $\mathbf{e}_2 = \mathbf{t}\mathbf{e}_1$. Then, vector $\mathbf{m}$ can be expressed as

$$\mathbf{m} = \cos\alpha \mathbf{e}_1 + \sin\alpha \mathbf{e}_2, \tag{9}$$

and $\alpha$ is the phase shift angle between vector m and radial line $\varphi = 0$. As $\alpha = 1/2\pi$ or $\alpha = 3/2\pi$, the local profile is of the tangential twist type, and $\alpha = 0$ or $\pi$ corresponds to the radial twist type. The schematics of angle $\alpha$ are presented in Fig. S3B.

## Data availability

Source data used to make the plots in main figures and Supplementary Figs. are provided with this paper. Source data are provided with this paper.

## Code availability

Codes used to make the plots in main figures and Supplementary Figs. are provided with this paper.

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

## Acknowledgements

We thank the fruitful discussions with Shengzhu Yi, Dr. Xiaoguang Wang, and Dr. Jack Binysh. C.P. acknowledged Dr. Rui He and Dr. Junhua Yuan for help with confocal microscopy measurements. R.Z. acknowledges the Hong Kong Research Grants Council grant no. 26302320. C.P. acknowledges National Natural Science Foundation of China Grant no. 62375254. J.J. acknowledges National Natural Science Foundation of China Grant no. 62305323.

## Author contributions

R.Z. conceived the research. X.W. and W.T. performed the theoretical and simulation calculations. J.J. and J.C. performed the experiments. R.Z. and C.P. directed the research. All authors participated in discussing and writing the manuscript.

## Competing interests

The authors declare no competing interests.
