## [Peer Review File · Nature Communications]

REVIEWER COMMENTS

Reviewer #1 (Remarks to the Author):

The authors combining theory, simulations, and experiments present an interesting study of complex topological nematic structures controllably created via Moiré effect between two parallel relatively rotated equally patterned surfaces. The paper is well written although due to its length and huge supplement with a large number of movies not easy to follow. The paper deserves publication in Nature Communications after considering the below listed remarks.

MAIN TEXT

Page 1

It would be good to write the title in a more compact form!

Page 2

Writing of particular complex forms via controlled creation of defects is interesting but I would not specifically stress writing letters in the abstract.

While listing manipulation methods consider also a partially related recent paper by R. Selinger in the Arxiv.

Page 4

The presented English letters are in fact Latin letters. I believe that in principle any script could be realized.

Page 5

For the readers it would be good to explain why the orthogonality of top and bottom surface directors allows formation of disclinations. Why hysteresis can occur, etc.

Results of structural simulations presented in numerous Figures do not include any information how disclination thickness is determined. In most of the published studies it is related to the degree of nematic order. Here only coloring is clearly described. This should be clearly stated in Figs and introduced in Materials and Methods.

Further it would be good to explain that in thin cells when the angle ψ starts to deviate from zero disclination lines come from infinity one by one with their distance decreasing with growing ψ .

It would be good to stress that deformations and the value of H/L where they start to appear, probably strongly depend on the anisotropy of elastic constants. What determines the left/right type of disclination helices?

Page 6

The Fig S4 mentioned in the relation to simulated optical microscope images opens some questions. Usually Jones approach do not properly present disclination lines as cores are too tiny. What was done to get rather thick dark lines in simulations of S and C states? In the Methods I do not see any note on this point!

Page 7-8

The analysis of topology of disclinations on these pages refers to the introduction of disclination topology in the Materials and Methods. That text is fine for specialists but a broader readership needs a graphical presentation of all disclination profiles with typical α and β angles that appear in this paper. It should go beyond what is now in Fig 1 G, H. The new figure should probably go in the Supplement or Materials and Methods if allowed.

Page 8

Most of impurities are attracted (not segregated) by nematic disclinations. It would be good to refer also to the knowledge disclosed by some earlier studies that analyze under what conditions these effects occur in nematics, cholesterics, and blue phases.

Page 9

Mentioning straight-line disclinations inducing entangled colloid structures is from Fig. 2C insets not clear without relating this observation to previous studies of disclination entangled, linked, and knotted structures.

Page 10

In the previous case the lines do not change much when ψ angle is small but their separation decreases with increasing ψ angle. How is here when ψ angle starts from zero? Probably increasing ψ loops enter the observation window from the infinity and become closed when they are small enough to fit in the window. In simulations it is important what you do with boundary conditions. It would be good to show how the aspect ratio change with growing ψ ? All this should be explained in more details in the supplement using theory or simulations!

Topological charge of the loop should be mention. Here loops should have charge zero so that can disappear if not prevented by surroundings.

Page 11

Also, for this discussion the above requested (Page 7-8) illustration of disclination profiles is crucial!

In the simulations presented in Fig3 I the disclination loops looks like artificially added similarly as on above mentioned Fig S4.

Page 12

The presented surface defects require strict planar anchoring. How strong it is in the experimental setup an what was used in simulations?

Page 13

More details are needed for $\psi=0$ structures: There are plus and minus defects and must be a coherence with neighbors how disclinations form quasi loops. It would be good to add a simulation with disclination loop profiles. What happens with a nematic between loops and surface defects where order is depressed?

In simulations, beside the dependence on the step size of the relaxation procedure also metastable states and related hysteresis are problematic. One usually adds quasi thermal noise to reduce such effects. What the authors has done?

Page 19

Here I repeat: Anchoring strength for in-plane and out of plane deviations should be presented as it certainty strongly effects experimental structures in thin samples. What was assumed in simulations?

Page 20-21

The paper is quite long therefore the Section "Theoretical model of numerical simulations" should be more compact as the approach is also well known. The technical details like different elastic constants and Frank form should go to the Supplement.

The statistical average in presented Q tensor definition is based on molecular orientation vectors and not directors and is not consistent with further use. For the phenomenological approach you need to write Q in terms of directors and order parameter S that is later mentioned but not introduced. Its values are useful for the presentation of disclinations. Usually also biaxiality is included. As biaxial effects are neglected this should be clearly stated. At this point including Refs 1, 41 and 50 is appropriate.

Page 22

What was taken for the anchoring strength? Are for these materials in plane and out-of- plane strengths equal?

It would be good to say a little more on the numerical procedure and limitations like resolution, sample size, and boundary conditions.

Page 23

The paper is quite long therefore also the Section "Simulation method for generating POM images" should be more compact as the approach is also well known. The technical details should be moved to the Supplement. It should be good to stress that Jones approach is not good in simulating disclination lines and has no use for bright field simulations. Probably one of available open codes for simulation of microscopic pictures of nematic structures should be used for comparison and for the bright field cases. The recent one from G. Poy looks useful. This could also resolve the previous question (Pages 6 and 11) about the artificial presentation of disclination lines.

Page 24

The title should be "Topological analysis of disclination lines"!

Page 29

Are disclination lines in Fig1 I, J and K artificially enhanced as the lines in Fig 2 B are much more natural?

Pages 29-30

The introduction of the Symbol T is not the best as T is already used for the period and T_{bold} is used for tangential vectors of disclination lines. On top it is has no sense as it is better to somewhere mention that as soon as ψ is not zero and disclinations appear the director is at least locally non-planar.

Pages 32 and 34

Disclination lines were artistically added as Jones approach can not properly image disclination lines. This should be clearly stated or a better simulation should be used (See remark on the page 23).

Page 35

Here T is used for time what is not the best, as it is used for period before!

Page 36

In Fig.5 A,B,C,D some details like boxes and text are difficult to recognize.

T in the section D of the Fig 5 has no purpose.

Page 38

Latin letter

SUPPLEMENT

Page S8

In the Fig S3 some features are to tiny to be clearly visible.

Page S9

Usually Jones approach do not properly present disclination lines as cores are to tiny. What was done to get rather thick dark lines in S and C state simulations? In methods I do not see any note on this point!

Page S14

Eq. for $\Delta\theta$: It would be good to show the ψ toward zero limit.

Page S15

Bright field images can be simulated using open codes (Page 23).

It would be good to explain the background of differences in loop behaviors.

Pages 33-34

One usually adds quasi thermal noise to reduce such effects.

Reviewer #2 (Remarks to the Author):

This paper presents the extensive results of a detailed simulation and experimental investigation of versatile topological defect structures in patterned nematic liquid crystal cells designed and controlled by the Moiré effect. By precisely rotating one patterned surface against the other, the authors generate versatile periodic defect arrays ranging from 1D line defects to 2D lattices of defect loops and hybrid disclinations. The topological structures are explained in detail using numerical simulations, polarized light microscopy images, and geometrically based arguments. The text is well written and accessible to non-specialists, and the main figures are well organized and supported by a comprehensive supplement. The results demonstrate an unexplored approach to the precise generation and tunability of topological defects in achiral nematic medium with considerable potential for application. I would say that such ideas have been around for a while, and now the authors have implemented them in an amazing way and made a giant leap forward in the manipulation of topological defects. In my opinion, the work definitely deserves publication, provided that the comments described below are taken into account and adequately addressed.

The data are of high quality, technically sound and valid. The presentation is above average even for high profile journals, the conclusions are supported with both simulation and experimental results, and many details are clearly explained in the extensive supplementary material. The movies are excellent, adding to the understanding of the dynamic response and showing the differences in rotational motion. It is obvious that this research was led by an experienced simulation group that found a similarly capable experimental team to perform this work in parallel. However, there is room for improvement and I have few suggestions to add to the submitted version of the manuscript.

First, the title is long, fuzzy, and somewhat confusing; I think the version in the supplement is better, because the novelty lies in a versatile design of numerous topological defect configurations by a kind of Fourier transform. The topic of colloidal assembly is of secondary importance here and not so new to the liquid crystal community, while the inverse design of shapes is commented on only briefly at the end of the article as a possible application of the nematic moiré. Therefore, I suggest the authors reconsider the title. In addition to references 4, 8, 19, 43, and 44, the authors should be aware of the following recent publications: Phys. Rev. E 97, 020701(R) (2018), Phys. Rev. Lett. 123, 097801 (2019), Crystals 10, 840 (2020), Phys. Rev. Lett. 130, 178101 (2023). I am also surprised that some visible papers on colloidal assembly in a twisted nematic cell have not been cited, for example, Science 333, 62 (2011) and PNAS 112, 1675 (2015), as there are several parallels between the systems. The simplest example of the Moiré effect is a twisted nematic cell that, in the presence of homeotropic microspheres, generates topologically charged disclination loops that can be entangled and separated by alternating twist domains. There, as in an earlier work by Ravnik et al, Phys. Rev. Lett. 99, 247801 (2007), periodic 1D and 2D disclination loops sensitive to cell gap and particle spacing have been discussed. Here, the loops are topologically neutral and there is no more than a π rotation from the top to the bottom patterned substrate, so I assume that knot formation is topologically forbidden. In my view, more complex 2D lattices of ring defects are closer to space-filling curves or even some kind of Truchet contour, since there are no real crossings that could lead to entanglement. However, I would appreciate if the authors could elaborate more about the statistics of the generated loops in terms of their number or density, length, creation/annihilation processes with respect to the rotation angle (and speed) in the case of 2D patterns; this should not be too difficult with the selected simulation results. I also wonder how the chirality (twist angle) is distributed across the sample for different loop types. It looks like this is determined by the surface patterns and does not depend on the loop size. Also, the presented Moiré effect is likely limited to rotationally symmetric patterns, or is there a hint of aperiodic motifs that could be realized in a similar manner. With this in mind, I would appreciate if the authors were more specific about the »arbitrary« shapes that can be realized by their method. I am also not sure about »magic angle«, which are more commonly used in twistronics, and what properties of the system might be associated with that. Finally, I find the potential applications interesting, although the meaning of Fig. 6 needs further discussion in the main text.

Reviewer #3 (Remarks to the Author):

The authors report on an interesting experimental and numerical study of liquid crystal structures that are formed when the confining surfaces are patterned and are turned by an angle with respect to each other. The work is inspired by Moire graphene photonics and brings an interesting novelty to the field of soft matter in general and liquid crystals in particular. This work is therefore original, timely, novel, and aims, to a certain degree, at bridging the graphene and soft matter communities.

The experiments are carefully conducted and the state of the art approach is used to generate surface patterns. The methodology used is quite complete and encompasses the various techniques that are needed to resolve the topology of the defects formed in different scenarios. The volume of the experimental work is satisfactory and covers all of the open questions that the reader might ask. The experiments are very well supported by numerical simulations, both of the complex structures and optics of these structures. The agreement between the experiments and the modelling is good and gives rather high level of confidence to this work.

Besides using the Moire approach to LC structures, much of the novelty aims at the topology of the complex LC structures that are formed as a result of confinement, surface patterning and the Moire twisting angle. Depending on the thickness of the LC layer and the type of surface pattern, the authors present a good number of novel defect structures, such as arrays of straight or curved lines, web-like defect line pattern or isolated defect loops. The topology of the lines and loops is carefully analyzed using numerical modelling. Finally, trapping and assembly of colloidal particles in defect lines is demonstrated, which is, however, not a real novelty and was reported in a number of previous articles.

This work has the potential for publication in Nature Communications provided some revisions are made:

1. In the abstract and in some parts of the main text it is stressed that "As such, the proposed simple twist method enables the design and tuning of arbitrary mesoscopic structures in liquid crystals, facilitating applications including defect-directed self-assembly, material transport, micro-reactors, photonic devices, and anti-counterfeiting materials." or in line 181: "This allows reversible reprogramming of colloids as building blocks to achieve multiple functions." In my opinion this is greatly exaggerated and should be made much more realistic. What is shown is trapping of colloidal particles in specific defect line patterns, which are not "arbitrary" or could be pre-designed. Moreover, Figure 6 shows only numerical modelling that is not supported by experimental images. This makes the proposed applications not quite convincing.

2. The referencing to previous work is not quite complete. For example, trapping of colloidal particles in defect lines demonstrated by others, is not referenced (line 177). This was demonstrated by D. Pires, J.-B. Fleury, and Y. Galerne, Phys. Rev. Lett. 98, 247801(2007) and Skarabot et al (PRE 77, 061706(2008)).

3. The confocal images in Fig. 1L are not consistent with the simulation and also are not adequately described in the text: "With scanning along z-axis, another four disclination lines near the bottom substrate show up, which finally form the web-like structures when viewed from the z-direction (Fig. 1L) and are agreed with the simulation." What I see from the 3D confocal structure (right panel in Fig.1L) is a kind of wall that extends from one surface to another, and not two distinct layers with disclination lines. This should be made clear in the revision.

4. The local topology and structure of disclinations and closed loops can be further characterized by two angles, as introduced in Refs. 34 and 35. However, it is not mentioned, in the case of closed loops (Fig. 3G), what is their net topological charge. I assume it is zero, because they can disappear, but it would be helpful to include the concept of the net charge as well.

There are also some technical deficiencies that need to be fixed:

1. In line 81, the "gap size H" is not indicated in Figure 1A.

2. In line 83, periodic splay-bend pattern is mentioned to be seen in Figure 1A. However, due to the perspective used in this 3D image, this is not well recognizable. It is suggested to add a top view of such a pattern as an inset.

3. In line 86, the tilt angle θ is introduced, This angle is not indicated in Figure 1A.

4. Line 91: " Ψ " is not in Fig. 1A, but in Fig. 1B.

5. Figure 3E: the structures are shown from thick cell to thin cell, just opposite to Figure 1.C, D and E. I suggest to make this coherent, it will be less confusing for the reader.

Reviewer #4 (Remarks to the Author):

The authors carry out experimental and numerical studies on the design of the structures of a nematic liquid crystal cell sandwiched by two patterned surfaces. They prepare two identical surfaces and rotate the top surface with respect to the bottom one to make a "moire" pattern. By one-dimensional (1D) patterned surfaces, they produce different types of configurations with an array of disclinations, both experimentally and numerically, depending on the cell thickness and the detail of the pattern. Their structural properties are explained by a simple theoretical argument. They also demonstrate that the disclination array can serve as a template for the arrangement of colloidal particles. Their strategy is extended also to 2D patterned surfaces.

The authors do not pay due respect for the previous studies presenting the basic and important ideas underlying the present work. Their "theory" predicting the position of disclination lines by the twist angle of the director in the thickness direction has been already exploited by Ozaki's group [Sunami et al., Phys. Rev. E 97, 020701(R); Ouchi et al., Phys. Rev. Lett. 123, 097801]. They used exactly the same strategy to design almost arbitrary shapes of disclination lines. The authors' statement "This theory can help the inverse design of arbitrary defect structures in future works" (page 15) is therefore invalid.

Disclination lines have long been used for manipulating colloidal particles, and one of the first important contributions was made by Galerne's group [Pires et al., Phys. Rev. Lett. 98, 247801; Fleury et al., Phys. Rev. Lett. 103, 267801]. Ref. 19 (Yoshida et al., Nature Commun. 6, 7180) also demonstrates the formation of colloidal arrays using disclination lines.

The motivation of the surface pattern design in Fig. 6 (encoding of characters) and how this design works are not at all clear. I am also wondering whether the characters are really visible under a microscope.

In the last part the authors comment that their strategy may be applicable to display applications because of lower threshold voltage for Frederiks transition. However, I cannot convince myself that their inhomogeneous moire structures with a periodicity of a few tens of microns are really useful for display application. If the periodicity is pushed to a smaller value to avoid the problem of inhomogeneity, the surfaces could just act as an effectively uniform substrate. In the main text the authors give just a vague argument and pushes out all the details, but the description in the main text should be comprehensive enough.

I also have a concern about the confocal images (Fig. 1L). How they should be interpreted is not explained in the main text or Methods (Should one see local liquid crystal orientation as in POM?). How one should understand the right panel of Fig. 1L (confocal image) is not presented, either. The authors state that the disclination lines are localized in the vicinity of the surfaces, but the center and the right confocal images do not seem to support their claim; rather there appear to be domain walls spanning the thickness direction.

Regarding the 2D pattern case, the authors say that the choice of the rotation center does not affect the defect configuration (p13 of the main text, and p24 of the Supplementary PDF). However, the 2D system lacks continuous translational symmetry, and indeed Movie S10 demonstrates the effect of the choice of the rotation center on the detailed structure of the disclinations.

As the authors state in the Introduction, moire patterns have been attracting great attention in various fields of condensed matter physics, and therefore the present study might appeal to broad interest. However, considering the fact that the underlying ideas towards the design of disclinations have been already exploited, and the lack of clarity in the interpretation of the confocal images, I do not think this work merits publication in a high-profile journal such as Nature Communications.

I also have several concerns about the presentation as listed below:

* The authors put too much information using small characters in individual small panels of the figures without caring about visibility. For example, the geometrical parameters in Fig. 2A, B are hardly visible. Superimposed top and bottom surface patterns in Figs. 1B, 3B, S2B, and S8B are hardly recognizable. Visibility of Fig. S3 is quite poor.

* The meaning of "isotropic moire" is not clear. Does it simply mean moires without a liquid crystal?

* (page 6) Is the top surface really at $z=0$?

* Some results are presented without statements on their implications. In pages 7 and 8, the authors just list up the local structural properties without any insights given (I also note that "local topology" is not suitable because topology refers to properties invariant under continuous transformations). What can readers learn from the plot of A_{xy} (Fig. 2F,G) ?

* (Page 8) Is wedge winding really characterized by $\beta \neq \pi/2$?

* (Page 12) The dynamics of defect loops is not clearly explained. How and when does the shrinkage of disclination loop occur in the absence of colloidal particles?

* The detailed structure of the 2D imprinted pattern is not explained.

* In Fig. 5B and D, does a dashed orange (or blue) square in the two panels really correspond to the same region?

* [Fig. 5 G, H] What are the green and blue regions, and the blue lines in the latter?

* [Theoretical model of numerical simulations (page 20-22, and the Supplementary PDF and movies)
]

* The one-constant approximation usually refers to setting $K_1=K_2=K_3$, not $L_1=L_2=L_3=L_4$. In the Supplementary PDF (p26), the authors inconsistently assumes the former.

* (Just after eq. 8) What do the authors mean by the average in the definition of the surface field Q_s ?

* If the authors are interested only in equilibrium (or metastable) structures the definition of the time t is not important, and it is safely regarded as "the relaxation step". However, they present the "dynamics" of disclination lines in response to the rotation of one surface or the applied field (Supplementary videos S11 and S12). Then the unit of time should be clearly presented. It is not clear either what is the relation between τ_{cell} and τ_0 .

* Just after eq. (10), the sentence "For surface points, ..." does not seem to make sense.

* The anchoring strength W , the field strength E and the dielectric anisotropy ϵ_a are not given.

* In Supplementary video 12, it seems that the field is turned on and off somewhere, but it is not mentioned in the caption.

* What was the cell gap H chosen in the simulations in Fig. S22? Is it the same in all the simulations? (If so, two structures are metastable).

* In Tables S1 and S2, the value of the free energy is meaningless unless the system size is specified. I am also wondering why the value of the free energy in Table S1 is so different from that in Table S2.

* (Section 6.2.1 of the Supplementary PDF) It is not clear what the "four different conditions" are. Does θ specify the azimuthal angle of the uniform alignment of the liquid crystal? What are the symbols like (1,0,0) in Figs. S29 and S30?

* (Section 6.2.2 of the Supplementary PDF) The definition of τ_s is different from that in the Methods section.

* [p 23] \mathbf{V} could be referred to as the Jones vector. Instead of Ref. 51, a standard textbook or review on liquid crystal optics (or optics in general) could be cited.

* [Caption of Fig. S1] I could not find the definition of the angle θ_g .

* The definition of the 2D rotation matrix in eq. (S2) is not consistent with that in eq. (12).

* [Supplementary PDF, pages 17 and 20, and Fig. S17] Regions I and III are not consistently defined.

* The authors should carefully check cross references:

* (page 8) Does Ref. 31 really concern the sentence "The angle α of the S-state changes from 0 to 2π in a lattice period L : this contrasts with the line defect with uniform α in similar systems in which the top substrate adopts a uniform anchoring"?

* (page 18) Fig. S25D should be Fig. S27D.

* (Supplementary PDF, p12) Fig. 2D should be Fig. 2E.

* (Supplementary PDF, page 24) Movie S11 should be Movie S10.

* There should be more standard references that replace Refs. 48 (orientation of azo dye in response to linearly polarized light. For example, Chigrinov, Kozenkov and Kwok, "Photoalignment of Liquid

Crystalline Materials: Physics and Applications") and 49 (hedgehog defect accompanying a spherical particle with normal anchoring. For example, Poulin et al., Science 275, 1770).

* The text is well written, but contains several typographical/grammatical errors:

* (page 6) "are agreed with" -> "agree with"

* (page 24, just before eq. 16) "angle" is doubled.

* (page 25) "We thank the fruitful discussions S.Y. ..." looks strange.

* (Page 5 of the Supplementary PDF) " $\theta(x,y,z)$ in the vicinity of the top surface" is not properly located.

* (page 14 of the Supplementary PDF) "this predict" -> "this prediction"

We thank the four Reviewers for their time carefully reading our manuscript and providing us with constructive and helpful feedback on our manuscript. According to their comments, we have substantially revised it. All the text edits are shown in blue mark-ups in the revised manuscript. In the following, we provide a point-to-point response to the Reviewers' comments.

Reviewer #1 (Remarks to the Author):

The authors combining theory, simulations, and experiments present an interesting study of complex topological nematic structures controllably created via Moiré effect between two parallel relatively rotated equally patterned surfaces. The paper is well written, although due to its length and huge supplement with a large number of movies not easy to follow. The paper deserves publication in Nature Communications after considering the below listed remarks.

We thank the Reviewer for the positive and helpful comments. We have addressed all the remarks made by the Reviewer, as elaborated below.

MAIN TEXT

Page 1

It would be good to write the title in a more compact form!

Following the Reviewer's suggestion, we have used a more compact title: "Moiré effect enabled versatile design of topological defects in nematic liquid crystals."

Page 2

Writing of particular complex forms via controlled creation of defects is interesting but I would not specifically stress writing letters in the abstract.

We agree with the Reviewer. We have removed the word "letters" in the revised manuscript.

While listing manipulation methods consider also a partially related recent paper by R. Selinger in the Arxiv.

We thank the Reviewer for bringing up this related work. We have cited this arXiv paper in the revised manuscript.

Page 4

The presented English letters are in fact Latin letters. I believe that in principle any script could be realized.

We thank the Reviewer for the point. In the revised manuscript, we have used "pixelated regions" instead.

Page 5

For the readers, it would be good to explain why the orthogonality of top and bottom surface directors allows formation of disclinations. Why hysteresis can occur, etc.

Following the Reviewer's suggestion, we have added the following discussion with relevant references to the revised manuscript to explain why orthogonal surface directors are related to the generation of disclinations, their hysteresis, and the limitation of the reasoning:

Page 5:

Because of the preferred angle mismatch between the two substrates, there is a point-wise twist deformation in the bulk achiral nematic. The handedness of the twist is determined by the acute angle the two preferred orientations make. When this angle transitions from acute to obtuse, there will be a twist reversal, at which the bulk nematic is frustrated. Therefore, disclinations can emerge in regions where the two preferred orientations are orthogonal (44, 45).

Page 15:

The hysteresis of the defect structure is rooted in the complex free energy landscape of the system. During substrate rotation, there is a competition between surface-preferred director field and bulk elasticity. Slow rotation will render the system remaining in the free energy minimum constrained by the anchoring condition; for fast rotation, the bulk director has no time to reorient to satisfy the anchoring condition, thereby leading to a different, path-dependent director field. This also explains the rotation rate dependence of the defect structure in the 2D pattern.

Page 18:

a simple theory can correctly predict these disclinations in the thin-cell limit, but fails for thicker cells.

Results of structural simulations presented in numerous Figures do not include any information how disclination thickness is determined. In most of the published studies it is related to the degree of nematic order. Here only coloring is clearly described. This should be clearly stated in Figs and introduced in Materials and Methods.

Following the Reviewer's suggestion, we have added more explanations in the revised Materials and Methods section about how we determine the disclination thickness: "According to Eq. 3, the equilibrium scalar order parameter $S_0 = 0.62$ in the simulation. Defect regions are described by an iso-surface of $S = 0.45$."

Further it would be good to explain that in thin cells when the angle ψ starts to deviate from zero disclination lines come from infinity one by one with their distance decreasing with growing ψ .

We have added the following explanation in the revised manuscript: “Note that as soon as the angle Ψ starts to deviate from 0° , disclination lines in the above states come from infinity one by one with their distance decreasing with growing Ψ .”

It would be good to stress that deformations and the value of H/L where they start to appear, probably strongly depend on the anisotropy of elastic constants.

We agree with the Reviewer that the defect states strongly depend on the anisotropy of the elastic constants of the nematic. We have performed additional simulations and added a new Fig. S35 and the following discussions in the revised Supplement Material 6.2.4:

In this work, we focus on one-elastic-constant assumption, i.e., $K_1 = K_2 = K_3 = K_{24}$. Our additional calculations show that the deformations near the state boundaries strongly depend on the anisotropy of the elastic constants. Specifically, we have performed simulations using the elastic constants of room temperature 4-Cyano-4'-pentylbiphenyl (5CB), $K_1 = 6$ pN, $K_2 = 3.9$ pN, $K_3 = 8.2$ pN, and $K_{24} = 7$ pN for the 1D cusp-like splay-bend pattern. We fix the rotation angle at $\Psi = 12^\circ$ and tune the cell gap H .

When the cell gap is narrow and the surface patterning dominates, both two sets of elastic constants exhibit the same transition between the S-state and C-state at $H/L = 0.35$. Since 5CB has a relatively lower twist modulus, twist deformation is the most energetically favored. From the analysis of the S-, C-, and W-state in Fig. S3, the S-state and W-state have periodic pure-twist local profile and twist deformation, whereas the C-state has a wedge-twist profile with the splay and bend deformation in the wedge regions. As shown in Fig. S35, the transition between the C-state and W-state of 5CB happens at $H/L = 0.62$, in contrast to $H/L = 0.8$ when one-elastic-constant is assumed. For the C-state, A_{xy}/T varies linearly from 0 to 1 under the one-constant assumption; whereas when 5CB elastic constants are adopted, A_{xy}/T increases slowly for $\frac{H}{L} \in [0.35, 0.6]$ and shows a sudden jump for $\frac{H}{L} \in [0.6, 0.62]$. The difference in the A_{xy}/T variation in the C-state confirms that the C-state is less energetically favored if we use the 5CB elastic constants. Therefore, the anisotropy of the elastic constants can alter the defect state boundaries, and this can be explained by the competition between the splay, bend, and twist energies. Nevertheless, the two choices of the elastic constants do not qualitatively change the defect state diagram. Without loss of generality, we focus on the one-constant approximation in this work.

Fig S35 The effect of elastic constants anisotropy on the defect state boundary. Simulation results of the 1D cusp pattern (at $\Psi = 12^\circ$) using (A) 5CB elastic constants and (B) one-elastic-constant.

What determines the left/right type of disclination helices?

The rotation direction can tune the handedness of the disclination helix. For the disclination helices reported in Figure 1 and 2, we rotated the top substrate counterclockwise and obtained right-handed helices. When a clockwise rotation is applied to the top substrate, left-handed disclination helices are generated. We have included the above discussion and the new Fig. S5 in the revised Supplement Material 2.4.

Fig S5 Left-handed helical structure found in the cusp pattern by a clockwise rotation. Rotation angle $\Psi = 12^\circ$. (A) $x - y$ view. (B) Angled view.

The Fig S4 mentioned in the relation to simulated optical microscope images opens some questions. Usually Jones approach do not properly present disclination lines as cores are too tiny. What was done to get rather thick dark lines in simulations of S and C states? In the Methods I do not see any note on this point!

We thank the Reviewer for this question. We would like to clarify that we did not artificially enhance the disclination lines in our simulated optical images. As a matter of fact, our Jones-matrix approach gives rise to sharp defects in very thin cells for the cusp and sinusoidal patterns we have considered. Take Fig. S4 as an example: the S- and C-state are generated in thin cells with $\frac{H}{L} = 0.2$ and $\frac{H}{L} = 0.4$, respectively. Otherwise, for the C-state using $\frac{H}{L} = 0.65$ (below) and the W-state in Fig. S4 (now Fig. S6 in the revised Supplement Information), the defect cores are less clear. In Fig. R1 shown in below, we use Jones-matrix approach to generate optical images with different $\frac{H}{L}$, and we find that as H increases, the visibility of the cores decreases, and they become invisible when $\frac{H}{L} > 0.5$.

Fig. R1 Simulated optical images using Jones-matrix-based calculations. From left to right: the cell gap $H/L = 0.2, 0.36, 0.48$ and 0.65 .

As suggested by the Reviewer, we have also used the open-source code for the colored POM images in Fig. 1J–K, as presented in Fig. R2. Similar to our code, dark cores can be generated when the cell is relatively thin. For thick cells in the W-state, the open-source gives a better match with the experiment than the Jones-matrix approach does (Fig. S6 in the revised Supplement Information).

Fig. R2 The comparison of optical patterns generated in the experiment and in the simulations (Guilhem Poy. Nemaktis. <https://github.com/warthan07/Nemaktis>). Scale bar: 50 μm .

Furthermore, we incorporated more discussions about the limitations of the Jones-matrix-based calculations in the revised Supplementary Information: Jones-matrix approach is not good in simulating the optical appearance of disclination lines and cannot be used to conduct bright-field simulations. Therefore, we have included a simulated bright field image in the revised Fig. S11 using G. Poy's code.

Fig. S11 Defect prediction, simulation, and experiment for the sinusoidal pattern at $\Psi = 40^\circ$. (A) The theoretical prediction with larger loops colored in red and smaller loops colored in blue. (B) Simulation of two cell thicknesses, a thinner cell of $H/L = 0.008$ (agrees with the prediction) and a thicker cell of $H/L = 0.1$. The bright-field image is generated using the

open-source code by Guilhem Poy [8]. (C) Medium cell results in the experiments agree with the $H/L = 0.1$ simulation. POM image (left) and bright field image (right). Scale bar: $50 \mu\text{m}$.

Page 7-8

The analysis of topology of disclinations on these pages refers to the introduction of disclination topology in the Materials and Methods. That text is fine for specialists but a broader readership needs a graphical presentation of all disclination profiles with typical alpha and beta angles that appear in this paper. It should go beyond what is now in Fig 1 G, H. The new figure should probably go in the Supplement or Materials and Methods if allowed.

Following the Reviewer's suggestion, a new Fig. S3 illustrating α and β shown in below has been added to the revised Supplementary Information.

Fig. S3 Director profiles of the windings in disclination lines. (A) A local $+1/2$ wedge winding with $\beta = 0$ continuously transforms into a $-1/2$ wedge with $\beta = \pi$ through an intermediate pure-twist winding with $\beta = \pi/2$. Unit vector \mathbf{t} denotes the tangent direction of the winding at the disclination curve. Unit vector $\mathbf{\Omega}$ is the rotation axis of the local directors (rods) in the cross-section of the winding. Unit vector \mathbf{m} is orthogonal to both \mathbf{t} and $\mathbf{\Omega}$. (B) For a pure-twist winding with $\beta = \pi/2$, tangential-Twist-I with $\sin \alpha = 1$ continuously transforms into Tangential-Twist-II with $\sin \alpha = -1$ through a Radial-Twist with $\sin \alpha = 0$. Vector \mathbf{e}_1 is within the cross-section plane pointing from the defect core to where $\varphi = 0^\circ$ or the director is along \mathbf{m} ($\mathbf{n} = \cos \frac{1}{2} \varphi \mathbf{m} + \sin \frac{1}{2} \varphi (\cos \beta \mathbf{t} \times \mathbf{m} + \sin \beta \mathbf{t})$), see **Topological analysis of disclination lines** in **Materials and Methods** for more details.

Page 8

Most of impurities are attracted (not segregated) by nematic disclinations. It would be good to refer also to the knowledge disclosed by some earlier studies that analyze under what conditions these effects occur in nematics, cholesterics, and blue phases.

We have added the following discussion citing some earlier works on this matter in the revised manuscript on Page 4:

It is known that defect lines engendered in LCs can be used to guide the self-assembly of colloidal particles (20, 34, 35). This opens up possibilities for a range of interparticle and assembly behaviors, such as modifying the dynamics of defect cores in nematics (36), triggering the gelation of the colloidal dispersions (37), and increasing the thermal stability of the blue phases against external fields (38, 39). Earlier studies have thoroughly characterized various closed knot defects in nematic colloids (40-42). In this work, we report the entrapment of colloidal particles along line defects, akin to the entanglement defect structure shown in (43).

Page 9

Mentioning straight-line disclinations inducing entangled colloid structures is from Fig. 2C insets not clear without relating this observation to previous studies of disclination entangled, linked, and knotted structures.

We thank the Reviewer for this suggestion. Interestingly, these self-assembled colloidal chains can also impact the straight-line disclinations (with a local twist profile) by inducing entangled structures (Fig. 2C insets). The disclinations around particles are sensitive to the cell gap (40). Different from the closed entanglement loops around particles (34, 41, 42), we show the interaction between existing defect curves and particles, as particles are attracted to the straight-line disclinations (the S-state). In our case, the defect curve (S-state) in Fig. 2C has a constant pure-twist profile akin to the assembly structures in (53). These references have been added in the revised manuscript for the discussion of the entangled defect structures.

Page 10

In the previous case the lines do not change much when psi angle is small but their separation decreases with increasing psi angle. How is here when psi angle starts from zero? Probably increasing psi loops enter the observation window from the infinity and become closed when they are small enough to fit in the window. In simulations it is important what you do with boundary conditions. It would be good to show how the aspect ratio change with growing psi? All this should be explained in more details in the supplement using theory or simulations!

Topological charge of the loop should be mention. Here loops should have charge zero so that can disappear if not prevented by surroundings.

We thank the Reviewer for these suggestions. We have calculated the aspect ratio of the series of simulations with $H/L = 0.1$ in the revised Supplement Information. As we increase the angle Ψ from 0° , the defects first enter the simulation window around $\Psi = 6^\circ$, as shown in Fig. S14A. Then we observe one array of closed defect loops at $\Psi = 12^\circ$ (Fig. S14B) and multiple arrays of defect loops with a distance of T after $\Psi \geq 24^\circ$. We have also characterized the change in the aspect ratio of the loops in the Supplement Information. We introduce parameters A_1 , A_2 , B_1 , and B_2 to represent the shapes of Loop-I and Loop-II, respectively (Fig.

S13A). The Loop-II has a larger projected area (in the x-y plane) and aspect ratio than the Loop-I. While the theory over-estimates the defect loop sizes (Fig. S14H), it makes accurate prediction of their aspect ratios (Fig. S14K). As a result, the theory overestimates the self-annihilation point. Take Loop-I as an example, theory predicts $\Psi = 80^\circ$, compared to $\Psi = 58^\circ$ in the simulation (Fig. S14H).

Fig. S14 Defect structure at different Ψ ($H/L = 0.1$) for the sinusoidal pattern. (A–F) Top view of the defect structure for $\Psi = 6^\circ$ (A), $\Psi = 12^\circ$ (B), $\Psi = 24^\circ$ (C), $\Psi = 30^\circ$ (D), $\Psi = 50^\circ$ (E), and $\Psi = 70^\circ$ (F). (G) The change of normalized period T^*/T' over rotation angle Ψ . (H) Projected area in the xy -plane of Loop-I and Loop-II in simulations and theories. (K) The aspect ratio of Loop-I and Loop-II in simulations and theories.

According to the Reviewer's comment, we have also added the discussion about the topological charge of the loops in the revised Supplement Information. Disclination loops, like hedgehog point defects, carry an integer hedgehog charge [4, 9]:

$$d = \frac{1}{4\pi} \int_{S^2} d\theta d\phi \mathbf{n} \cdot [\partial_\theta \mathbf{n} \times \partial_\phi \mathbf{n}].$$

We find that the topological charge of the three kinds of loops is zero and they can all disappear or self-annihilate upon continuous rotation or cell gap change.

Page 11

Also, for this discussion the above requested (Page 7-8) illustration of disclination profiles is crucial!

We have included illustration schematics in the revised Supplement Information (Fig. S3). Please refer to Fig. S3 shown in the above for more details.

In the simulations presented in Fig3 I the disclination loops looks like artificially added similarly as on above mentioned Fig S4.

We didn't artificially enhance the results as replied in Page 5. Using the Jones-matrix approach, we can only see sharp dark cores in thin cells. For example, in Fig. 3, the optical images were obtained using with $\frac{H}{L} = 0.1$ (thin cell) and $\frac{H}{L} = 0.3$ (medium cell). If we increase the ratio to $\frac{H}{L} = 0.5$, the defect cores are less visible and some regions of the defect cores become light-colored (Fig. R3).

Fig. R3 Jones-matrix–based simulated optical image using the sinusoidal pattern at $\frac{H}{L} = 0.5$.

Page 12

The presented surface defects require strict planar anchoring. How strong it is in the experimental setup and what was used in simulations?

We thank the Reviewer for this question. In the experiment, the anchoring strength is in the order of $\sim 10^{-3} - 10^{-4}$ J/m² (strong anchoring) (48). The anchoring strength W is set to be infinite in the simulations (the surface preferred director \mathbf{n}_s is fixed) to approximate the strong anchoring condition in the experiment. The good match between the experiment and

simulation in different scenarios validates the strong anchoring assumption used in the simulation. We have added the above clarifications in the revised manuscript.

Page 13

More details are needed for $\psi=0$ structures: There are plus and minus defects and must be a coherence with neighbors how disclinations form quasi loops. It would be good to add a simulation with disclination loop profiles. What happens with a nematic between loops and surface defects where order is depressed?

We have modified Fig. S17 and added more details about the quasi-loop defects in the Supplement Information 4.2.

Fig. S17: Loop-like defect array at $\Psi = 0^\circ$ using the 2D defect pattern. (A) Arrays of loop-like defects in the simulation. Details of the dashed box: (B) Experimental bright field snapshot corresponding to the boxed region in (A) (xy view); (C–E) Different angled views of the simulation results in which angle β is used to characterize the defects.

Fig S17A shows the configuration of the quasi-loop defects in 3D colored by the normalized z coordinate. In the planar view of the experiment, a quasi-loop defect appears as two dots due to the fact that the defects are split from the surfaces. We further characterize topological structures using angle β in the simulation.

We would like to also thank the question about the nematic between loop defects and surface defects. As shown in Fig S17C-E, a quasi-loop that connects two $+1$ defect cores has $\beta \in [0, \pi/2]$: near the two surfaces, β approaches $\pi/2$ while $+1/2$ wedge profile appears in the

bulk with $\beta \sim 0$. On the other hand, a quasi-loop that connects two -1 defect cores exhibits $\beta \in [\pi/2, \pi]$: near the two surfaces, β approaches $\pi/2$ while $-1/2$ wedge profile appears in the bulk with $\beta \sim \pi$. These two arc-like defects repel each other elastically and appear as two short lines when viewed from top (Fig. S17B).

In simulations, besides the dependence on the step size of the relaxation procedure also metastable states and related hysteresis are problematic. One usually adds quasi thermal noise to reduce such effects. What the authors have done?

We thank the Reviewer for this question, and we have added the discussions in the revised Supplement Information. To get the actual relaxed state with the minimum free energy and to avoid being stuck in local minimums, we try different initial conditions (ansatzes) as an equivalent way to account for the effect of thermal noise. The discussions have been added in Section 6.2.2 in the Supplement Information. In our simulation with electric fields, we added random noise to the initial director field (very small, nonzero component n_z) to study the Frederiks transition, which shows a good match with the theory (Section 5 in SI). Without the initial noise, the transition requires a four times larger transition threshold in the simulations.

Page 19

Here I repeat: Anchoring strength for in-plane and out of plane deviations should be presented as it certainty strongly effects experimental structures in thin samples. What was assumed in simulations?

In the experiment, we applied strong anchoring and the polar anchoring (out of plane) energy for the used material is in the order of 10^{-3} J/m² and the azimuthal (in plane) anchoring energy is in the order of 10^{-4} J/m² (48). The good match between the experiment and simulation validates our infinite anchoring approximation. In the simulations, we assumed infinite anchoring strength for both the top and bottom surface, and we don't evolve the \mathbf{n}_s (the surface molecular orientation) in the energy minimization calculation to approximate the strong anchoring condition in the experiment (70).

Page 20-21

The paper is quite long therefore the Section "Theoretical model of numerical simulations" should be more compact as the approach is also well known. The technical details like different elastic constants and Frank form should go to the Supplement.

We thank the Reviewer for this advice. We have moved more technical details in the "Theoretical model of numerical simulations" to the Supplement (6.1.1).

The statistical average in presented Q tensor definition is based on molecular orientation vectors and not directors and is not consistent with further use. For the phenomenological approach you need to write Q in terms of directors and order parameter S that is later mentioned but not introduced. Its values are useful for the presentation of disclinations.

Usually also biaxiality is included. As biaxial effects are neglected this should be clearly stated. At this point including Refs 1, 41 and 50 is appropriate.

We agree with the Reviewer and have revised the manuscript accordingly. We have rewritten the definition of \mathbf{Q} and included Refs 1, 41, and 50 (now 1, 58 and 67). We modified \mathbf{Q} in the simulation methods and mentioned that we neglected the biaxiality. "The tensorial nematic order parameter \mathbf{Q} is given by $Q_{ij} = S(n_i n_j - \delta_{ij}/3) + \frac{P}{2}(e_i^{(1)} e_j^{(1)} - e_i^{(2)} e_j^{(2)})$, where the secondary director $\mathbf{e}^{(1)}$ is perpendicular to \mathbf{n} , $\mathbf{e}^{(2)} = \mathbf{n} \times \mathbf{e}^{(1)}$, and S is the nematic degree of order (59). In this work, we consider uniaxial nematics that are weakly deformed and biaxial effects can be ignored. Thus, we have $P = 0$ and $Q_{ij} = S(n_i n_j - \delta_{ij}/3)$ (1, 61, 69)."

Page 22

What was taken for the anchoring strength? Are for these materials in plane and out-of-plane strengths equal?

We thank the Reviewer for the questions. We have clarified the sentences in the revised manuscript. Infinite anchoring strength was taken as the assumption in the simulations. Therefore, in-plane and out-of-plane deviations are not allowed. In the experiment, we applied strong anchoring and the polar anchoring (out of plane) energy for the used material is $\sim 0.5 \times 10^{-3} \text{ J/m}^2$ and the azimuthal (in plane) anchoring energy is $\sim 0.6 \times 10^{-4} \text{ J/m}^2$ (48).

It would be good to say a little more on the numerical procedure and limitations like resolution, sample size, and boundary conditions.

We thank the Reviewer for this comment. We have incorporated more simulation details in the Materials and Methods and the Supplement Information. The shortcoming of the simulation is that the size is much smaller compared to the experiment. The characteristic length scale of the simulations is set to 6.63 nm (nematic coherence length). We use 30 lattice points for the period L and thus $L = 30 \times 6.63 \text{ nm} = 198.9 \text{ nm}$. However, $L = 75 \text{ }\mu\text{m}$ is used in the experiment. We compare simulations and experiments at the same ratio, H/L . Despite the mismatch in system size, simulations and experiments agree very well in terms of defect structures and optical patterns. We don't apply noise in the simulation, and we use different ansatzes to find the minimum free energy states instead. We don't impose periodic boundary conditions since the rotated pattern doesn't have translational symmetry. We set infinite anchoring for the top and bottom substrate.

Page 23

The paper is quite long therefore also the Section "Simulation method for generating POM images" should be more compact as the approach is also well known. The technical details should be moved to the Supplement. It should be good to stress that Jones approach is not good in simulating disclination lines and has no use for bright field simulations. Probably one of available open codes for simulation of microscopic pictures of nematic structures should be used for comparison and for the bright field cases. The recent one from G. Poy looks useful.

This could also resolve the previous question (Pages 6 and 11) about the artificial presentation of disclination lines.

We thank the Reviewer for the comments. We have moved the Section "Simulation method for generating POM images" into the Supplement (6.2.5) and added the limitations of the Jones-matrix approach.

As we replied on Page 5, we didn't present an artificially enhanced simulated optical image. We have tried the open-source code for the colored POM images in Fig.1J-K, as presented in Fig.R2. Similar to our code, dark cores can be generated when the cell is relatively thin. For thick cells in the W-state, the open-source code gives a better match with the experiment than the Jones-matrix approach.

Page 24

The title should be "Topological analysis of disclination lines"!

We have changed the title to "Topological analysis of disclination lines".

Page 29

Are disclination lines in Fig1 I, J and K artificially enhanced as the lines in Fig 2 B are much more natural?

We thank the Reviewer for this question. No, we didn't artificially enhance Fig. 1I, J, and K. In the experiments, we used two different types of liquid crystal materials which are JXLC-20000 and 5CB. In Fig1 I, J, and K, we used liquid crystal JXLC-20000, while in Fig 2B, 5CB was used. The birefringence for JXLC-20000 is about 0.05, and the birefringence for 5CB is about 0.2. Hence, the color of images is different. In the material and methods part, we have highlighted the following sentences to clarify this.

Both liquid crystal materials of 5CB and JXLC-20000 are used in the experiments. JXLC-20000 is used in Fig. 1, and 5CB is used in the rest of the experiments.

Pages 29-30

The introduction of the Symbol T is not the best as T is already used for the period and T_{bold} is used for tangential vectors of disclination lines.

We thank the Reviewer for this comment. We have removed the sentence in the caption and changed \mathbf{T} to \mathbf{t} for tangential vectors of disclination lines for clarification.

On top it is has no sense as it is better to somewhere mention that as soon as ψ is not zero and disclinations appear the director is at least locally non-planar.

We thank the Reviewer for this question. We have added the discussion in the second paragraph of the **Results** in the revised manuscript. "Upon the rotation operation, the bulk

nematic is frustrated by the mismatched preferred orientations of the two substrates (Fig. 1A, B) and disclinations appear as $\Psi \neq 0^\circ$ ". We have also clarified that we observe planar directors in patterned cells in all of the simulations, except when we apply an electric field that's along the z-axis (Section 5 in the Supplement Information).

Pages 32 and 34

Disclination lines were artistically added as Jones approach can not properly image disclination lines. This should be clearly stated or a better simulation should be used (See remark on the page 23).

We thank the Reviewer again for the comment. As we have replied in Page 5–6, we didn't artificially enhance the results. In our Jones-matrix calculations, we can only get sharp defect cores for thin cells. The simulated defect lines are less visible in thicker cells, where the director changes more smoothly. We have also added the limitations of the Jones-matrix approach in the revised Supplement Information (6.2.5).

Page 35

Here T is used for time what is not the best, as it is used for period before!

We have removed "T" in the caption of Fig. 1 and modified Fig. 4 to avoid confusing the reader.

Page 36

In Fig.5 A,B,C,D some details like boxes and text are difficult to recognize.

We have modified Fig. 5A–D for better visualization.

T in the section D of the Fig 5 has no purpose.

We have clarified the description in the revised manuscript. T in the section D of Fig. 5 is to show the moiré period of the defects in the 2D pattern.

Page 38

Latin letter

We have changed that.

SUPPLEMENT

Page S8

In the Fig S3 some features are too tiny to be clearly visible.

We have improved Fig. S3 (Fig. S4 in the revised Supplement Information) as below.

Fig. S4 The 3D view and director details of the S-, C- and W-state. (A) the S-state has a constant pure-twist local profile. (B) The configuration of the C-state is a helical-like, 3D curve. (C) The W-state has two groups of defects. The group in pink is close to the top surface, and the other group in blue is near the bottom.

Page S9

Usually Jones approach do not properly present disclination lines as cores are too tiny. What was done to get rather thick dark lines in S and C state simulations? In methods I do not see any note on this point!

We thank the Reviewer again for this question. As we have replied on Page 5–6, we didn't artificially enhance the results. In our Jones-matrix calculations, we can only get sharp defect cores for thin cells. The simulated defect lines are less visible in thicker cells, where the director changes more smoothly. We have also added the limitations of the Jones-matrix approach in the revised Supplement Information (6.2.5).

Page S14

Eq. for deltatheta: It would be good to show the psi toward zero limit.

One more sentence has been added in the revised Supplement Information. "Note that when $\Psi = 0$, $\Delta\theta = 0$ and defects are not generated."

Page S15

Bright field images can be simulated using open codes (Page 23).

We thank the Reviewer for this suggestion. We have included a simulated bright field image in the revised Fig. S11B using G. Poy's code.

It would be good to explain the background of differences in loop behaviors.

We have included more explanations of loop behaviours in the revised Supplement Information: "Movie S7 shows the evolution of loops with the change of cell gap. With the imposed patterns at the top and bottom substrates, a position-dependent twist is induced in the bulk achiral nematic. For a thin cell, the surface anchoring effect is important, and the twist-winding disclinations appear where the angle difference $\Delta\theta(x, y)$ is $\pi/2$. When the two substrates are away from each other (large H/L), the surface patterning effect becomes relatively weak and the bulk nematic dominates. Smaller loops are less stable and gradually self-annihilate. As the cell becomes even thicker, the system chooses to minimize its elastic free energy by aligning the nematic uniformly in the bulk and forming defects near the substrates to accommodate the surface anchoring pattern. The emergency of the W-state is observed."

Pages 33-34

One usually adds quasi thermal noise to reduce such effects.

We thank the Reviewer for this comment. We have used different initial conditions including a random initial configuration, and compare their free energies instead. We have added the discussions about quasi-thermal noise in Supplement (6.2.2). "We didn't add quasi-thermal noise in the simulations. Instead, to get the actual relaxed state with the minimum energy and avoid being stuck in local minimums, we try different initial conditions (ansatzes) as a way to test the system's reaction to noise." In our simulation with electric fields, we added random noise to the initial director field (very small, nonzero component n_z) to study the Frederiks transition, which shows a good match with the theory (Section 5 in SI). Without the initial noise, the transition requires a four times larger transition threshold in the simulations.

Reviewer #2 (Remarks to the Author):

This paper presents the extensive results of a detailed simulation and experimental investigation of versatile topological defect structures in patterned nematic liquid crystal cells designed and controlled by the Moiré effect. By precisely rotating one patterned surface against the other, the authors generate versatile periodic defect arrays ranging from 1D line defects to 2D lattices of defect loops and hybrid disclinations. The topological structures are explained in detail using numerical simulations, polarized light microscopy images, and

geometrically based arguments. The text is well written and accessible to non-specialists, and the main figures are well organized and supported by a comprehensive supplement. The results demonstrate an unexplored approach to the precise generation and tunability of topological defects in achiral nematic medium with considerable potential for application. I would say that such ideas have been around for a while, and now the authors have implemented them in an amazing way and made a giant leap forward in the manipulation of topological defects. In my opinion, the work definitely deserves publication, provided that the comments described below are taken into account and adequately addressed.

The data are of high quality, technically sound and valid. The presentation is above average even for high profile journals, the conclusions are supported with both simulation and experimental results, and many details are clearly explained in the extensive supplementary material. The movies are excellent, adding to the understanding of the dynamic response and showing the differences in rotational motion. It is obvious that this research was led by an experienced simulation group that found a similarly capable experimental team to perform this work in parallel. However, there is room for improvement and I have few suggestions to add to the submitted version of the manuscript.

We appreciate the Reviewer for the positive evaluation. We have incorporated all the suggestions made by the Reviewer as elaborated in below.

First, the title is long, fuzzy, and somewhat confusing; I think the version in the supplement is better, because the novelty lies in a versatile design of numerous topological defect configurations by a kind of Fourier transform. The topic of colloidal assembly is of secondary importance here and not so new to the liquid crystal community, while the inverse design of shapes is commented on only briefly at the end of the article as a possible application of the nematic moiré. Therefore, I suggest the authors reconsider the title.

We agree with the Reviewer. Following the Reviewer's suggestion, we have used a more compact title: Moiré effect enabled the versatile design of topological defects in nematic cells.

In addition to references 4, 8, 19, 43, and 44, the authors should be aware of the following recent publications: Phys. Rev. E 97, 020701(R) (2018), Phys. Rev. Lett. 123, 097801 (2019), Crystals 10, 840 (2020), Phys. Rev. Lett. 130, 178101 (2023). I am also surprised that some visible papers on colloidal assembly in a twisted nematic cell have not been cited, for example, Science 333, 62 (2011) and PNAS 112, 1675 (2015), as there are several parallels between the systems. The simplest example of the Moiré effect is a twisted nematic cell that, in the presence of homeotropic microspheres, generates topologically charged disclination loops that can be entangled and separated by alternating twist domains. There, as in an earlier work by Ravnik et al, Phys. Rev. Lett. 99, 247801 (2007), periodic 1D and 2D disclination loops sensitive to cell gap and particle spacing have been discussed.

We thank the Reviewer for bringing up these relevant publications. All these references have now been added to the revised manuscript with discussions.

About surface patterning and topological defects:

“Liquid crystals (LCs) consist of rod- or plank-like molecules, which can self-assemble into well-defined mesoscopic structures with long-range orientational order (1). This ordering can be locally frustrated due to topological reasons, leading to regions called topological defects (2-4).”

“The handedness of the twist is determined by the acute angle the two preferred orientations make. When this angle transitions from acute to obtuse, there will be a twist reversal, at which the bulk nematic is frustrated. Therefore, we expect that disclinations will emerge where the two preferred orientation angles are orthogonal, i.e., $\Delta\theta=m\pi/2$ with $m=\pm 1, \pm 3\dots$ (44, 45).”

“Photo-alignment is a versatile tool to pattern confining substrates and stabilize both singular and non-singular defects in LC cells (44, 45, 47).”

About colloidal assembly and entanglement:

“In earlier work, the disclinations around particles were found to be sensitive to the cell gap (40). Different from the closed entanglement loops around particles (34, 41, 42), we show the interaction between existing defect curves and particles, as particles are attracted to the straight-line disclinations (the S-state).”

Here, the loops are topologically neutral and there is no more than a pi rotation from the top to the bottom patterned substrate, so I assume that knot formation is topologically forbidden. In my view, more complex 2D lattices of ring defects are closer to space-filling curves or even some kind of Truchet contour, since there are no real crossings that could lead to entanglement.

We thank the Reviewer for these comments. We have added these discussions to the revised manuscript. “However, if the rotation speed is fast (Materials and Methods, Supplementary Material 6.2.2), the system will enter a disordered defect state and be stuck in a state comprised of space-filling curve defects after a π -turn, akin to the Truchet pattern (Movie S13).”

However, I would appreciate if the authors could elaborate more about the statistics of the generated loops in terms of their number or density, length, creation/annihilation processes with respect to the rotation angle (and speed) in the case of 2D patterns; this should not be too difficult with the selected simulation results.

We thank the Reviewer for the helpful suggestions. We have included more statistics about the quasi-loop defects for the 2D pattern. Those quasi-loops in fact consist of two separate curves, as presented clearer in Fig.5E and F. During substrate rotation, defect creation/annihilation may occur. However, since each defect core on the two patterns can generate two split curves, the total number of defect curves is constant during the rotation. Here we count the ratio between bulk defects (defect curve connecting both substrates) and near-surface defects (defect curve with its two ends on the same substrate), and plot it against the twist angle Ψ in Fig. S25. From Fig. S25B and C, we learn that when the rotation is sufficiently fast, the system doesn't have time to relax to satisfy the new anchoring pattern, and there are more near-surface defects. Since we apply $H/L = 0.3$ in the simulations, defects are shorter if they go through the bulk from top to bottom. In quasi-static simulations with the system remaining in the stable state, more defect lines choose to connect through the

bulk (Fig. S25B, C), and as a result, its defect volume is lower than $\tau_s / \tau_{cell} = 35.4$ and $\tau_s / \tau_{cell} = 236$ (Fig. S25D). Here, τ_{cell} is the characteristic time scale for the relaxation of the nematic director field, $\tau_{cell} = \frac{\gamma_1 H^2}{L_1} = \frac{2S^2 H^2}{\Gamma_s L_1}$ with γ_1 being the rotational viscosity, H being the cell gap, and L_1 being the elastic constant. The simulation time $\tau_s = N\tau_0$ is expressed in the simulation time unit τ_0 (N is the number of simulation steps). To achieve a quasi-static simulation, $\tau_s \gg \tau_{cell}$ has to be satisfied. More details about the simulation methods can be found in “Theoretical model of numerical simulations”.

Fig. S25 The ratio of bulk defects to near-surface defects. (A) A planar view and side view of a fast rotation with two kinds of defects labelled. The ratio of the number of defects near the surfaces to the number of defects going through the cell changes with the rotation angle Ψ : (B) $\tau_s / \tau_{cell} = 35.4$ (rate-I); (C) $\tau_s / \tau_{cell} = 236$ (rate-II) and quasi-static. (D) Defect volumes of the three cases also change with the rotation angle.

I also wonder how the chirality (twist angle) is distributed across the sample for different loop types. It looks like this is determined by the surface patterns and does not depend on the loop size.

We thank the Reviewer for this comment. We have added the following new discussion to the revised manuscript and Supplement Information:

The twist angle can depend on not only the surface patterns but also the loop size. In Fig. S19, we characterize the twist angle of the defect windings using the angle β . In Region III (the vicinity of the unit cell center), due to the anchoring mismatch, quasi-loops with $+1/2$ ($-1/2$) wedge profile near the midplane are formed at the center of the unit cell (Fig. S18) while the loops are elongated with β approaching $\pi/2$ as they are further away from the center (the color change of the loops in enlarged view in Fig. S19A). Therefore, there is a correlation between the loop size and the twist angle of its windings in the ± 1 pattern system.

A

Fig. S19 Simulation details of the defect structure for the 2D pattern at $\Psi = 5^\circ$. (A) Top view of defects at $\Psi = 5^\circ$.

Also, the presented Moiré effect is likely limited to rotationally symmetric patterns, or is there a hint of aperiodic motifs that could be realized in a similar manner.

We thank the Reviewer for this question and have added the discussions in the revised Supplement Information. In the 2D ± 1 defect pattern, for $\Psi \in [21^\circ, 34^\circ]$ and $\Psi > 34^\circ$, no dominating moiré is obtained, which is the same as the phenomena in geometric moirés (28) and defects show aperiodic configurations in Fig. S21.

Fig S21 Aperiodic defect configurations in the 2D ± 1 defect pattern. From left to right: $\Psi = 29^\circ, 39^\circ$ and 44° .

I am also not sure about »magic angle«, which is more commonly used in twistrionics, and what properties of the system might be associated with that.

Indeed, the “magic angle” in twistrionics depends on the material property. In our LC system, we find a special angle, $\tan^{-1}(3/4)$ for the 2D pattern, which can give rise to emerging periodic defect structures, which shows singular periodicity in the moiré theory (Fig. 5G, H).

We have added the discussion in the revised manuscript, and to avoid confusion to the readers, we have replaced “magic angle” with “special angle” in most discussions.

With this in mind, I would appreciate if the authors were more specific about the »arbitrary« shapes that can be realized by their method. Finally, I find the potential applications interesting, although the meaning of Fig. 6 needs further discussion in the main text.

We thank the Reviewer for the question about “arbitrary” shapes. There are two implications for “arbitrary” defect shapes in our work: 1) We can rely on the two parameters, namely cell gap and twist angle, to achieve a variety of periodic defect structures with highly tunable periodicities using one specific anchoring pattern; 2). Combining the moiré theory and the insight that 90-degree anchoring difference in a thin cell generates defects, we are able to design arbitrary, pixelated region comprising defect loops (more details in the two following paragraphs).

We also thank the Reviewer for this comment on Fig. 6. We have modified Fig. 6 and have added more discussions for Fig. 6 in the revised manuscript. Understanding the structure of topological defects is important to many branches of physics, as defects play a pivotal role in physical processes such as self-assembly and phase transitions. For defect-based applications, existing research efforts have been devoted to the “forward design” problem: which topological structures with what properties can be formed from a given geometry or a given pattern. Moreover, for a given anchoring pattern, all the possible defect structures are often-times quite limited to a few metastable states. This greatly limits the applications of nematic disclinations. Here, we propose to use the moiré effect to address the inverse problem of designing arbitrary pixelated defect-regions.

The design of Fig. 6 is inspired by the application of moiré methods in the protection of documents and products [56]. Based on the moiré intensity profiles, new patterns are generated between two specially designed periodic dot screens, one of which being a micro-structured image that is located on the document itself (bottom, green layer, T shape, Fig. 6A), while the other plays the role of a revealer (top, pink layer, circle shape, Fig. 6A). In the design of anti-counterfeiting materials, the superposed new pattern of a highly visible repetitive moiré pattern of a predefined intensity profile shape and colour reveals the authenticated properties, documents or products. Following the understanding that the disclination lines appear where the twist angle of the pattern director has 90-degree difference in a thin cell (43, 44), we propose the inverse design possibilities and map the “document” pattern (bottom) and “revealer” pattern (top) to LCs. For a unit on the top pattern (Fig. 6A inset), we let the central circle have $\theta = 100^\circ$ and the other region be $\theta = 45^\circ$; for a unit on the bottom pattern (Fig. 6B), θ changes continuously from the central line to the edges of letter shapes, varying from 0° to 45° . We let the surrounding regions in both patterns share the same angle θ so that only where the circle part on the top and the letter part on the bottom overlaps, defect loops can be generated. As shown in Fig. 6C, an amplified region (in the shape of U, S, and T) consists of multiple defect loops, and we call it a pixelated shape. Like the $2D \pm 1$ defect pattern, we can also tune the period and tilting angle of the pixelated regions. Another advantage of this approach is each small loop can be individually controlled and programmed with the surroundings. With that, we can embed other shapes in the “document pattern” and reveal amplified, pixelated regions in the

corresponding shapes. We therefore anticipate that nematic moiré cells can generate various programmed shapes, and can be potentially used for inverse design, printing, self-assembly, photonics, and anti-counterfeiting materials.

Fig. 6. Simulation design of on-demand topological structures using nematic moiré. Inspired by geometric moiré, using alternate surface patterns, various Latin letter-shaped defects following the moiré period and tilting angle can be generated. (A) By putting a “revealer” pattern (circular) on top of a “document” pattern (T-shape), an array of amplified T shapes can be revealed in the geometric moiré. One of the generated T shapes is noted by a dashed blue frame. Inset is the circular LC pattern mapped from the circular geometric pattern. (B) Map the “document pattern” at the bottom to the LC pattern (in the shape of U, S, and T). (C) Results of three simulations: Overlapping the circular LC pattern on the top and the three Latin letter LC patterns on the bottom sequentially, pixelated regions in the shape of U, S, and T comprising defect loops are generated in the three simulations, respectively.

Reviewer #3 (Remarks to the Author):

The authors report on an interesting experimental and numerical study of liquid crystal structures that are formed when the confining surfaces are patterned and are turned by an angle with respect to each other. The work is inspired by Moire graphene photonics and brings an interesting novelty to the field of soft matter in general and liquid crystals in particular. This work is therefore original, timely, novel, and aims, to a certain degree, at bridging the graphene and soft matter communities.

The experiments are carefully conducted and the state of the art approach is used to generate surface patterns. The methodology used is quite complete and encompasses the various

techniques that are needed to resolve the topology of the defects formed in different scenarios. The volume of the experimental work is satisfactory and covers all of the open questions that the reader might ask. The experiments are very well supported by numerical simulations, both of the complex structures and optics of these structures. The agreement between the experiments and the modelling is good and gives rather high level of confidence to this work.

Besides using the Moire approach to LC structures, much of the novelty aims at the topology of the complex LC structures that are formed as a result of confinement, surface patterning and the Moire twisting angle. Depending on the thickness of the LC layer and the type of surface pattern, the authors present a good number of novel defect structures, such as arrays of straight or curved lines, web-like defect line pattern or isolated defect loops. The topology of the lines and loops is carefully analyzed using numerical modelling. Finally, trapping and assembly of colloidal particles in defect lines is demonstrated, which is, however, not a real novelty and was reported in a number of previous articles.

This work has the potential for publication in Nature Communications provided some revisions are made:

We thank the Reviewer for the positive comments and insightful suggestions. We have incorporated all the revisions suggested by the Reviewer elaborated in below.

1. In the abstract and in some parts of the main text it is stressed that "As such, the proposed simple twist method enables the design and tuning of arbitrary mesoscopic structures in liquid crystals, facilitating applications including defect-directed self-assembly, material transport, micro-reactors, photonic devices, and anti-counterfeiting materials." or in line 181: "This allows reversible reprogramming of colloids as building blocks to achieve multiple functions." In my opinion this is greatly exaggerated and should be made much more realistic. What is shown is trapping of colloidal particles in specific defect line patterns, which are not "arbitrary" or could be pre-designed. Moreover, Figure 6 shows only numerical modeling that is not supported by experimental images. This makes the proposed applications not quite convincing.

We thank the Reviewer for this comment. We have removed the word "arbitrary" from the last sentence in the revised abstract and have also weakened the tune in the revised main text to the following "This is potentially helpful for reversible reprogramming of colloids as building blocks to achieve multiple functions." In fact, our previous work (Jiang et al. PNAS 120.16 (2023): e2221718120) has demonstrated the possibility to transport and reconfigure colloidal structures through the light-triggered rotation of a much simpler, uniform pattern of the top surface. On the one hand, the interaction between colloidal particles and defects in nematic moiré patterns encompasses a great deal of physics which is beyond the focus of the current work. On the other hand, a thorough experimental realization of entrapped colloids pattern requires a large pattern area, as a small tilt angle leads to large emerging periodicity. We therefore leave the comprehensive study of colloids in nematic moiré patterns to future works.

We have added more discussions for Figure 6 in terms of the design details and working mechanisms in the revised manuscript. The letters are much larger than the feature size of

the pattern, and therefore should be visible under the microscopy in the experiments. Based on the successful matching between the simulation and experiment as demonstrated in Figs. 1–5 for different 1D and 2D patterns, these letters should also be observed in the experiment. However, due to the current fabrication size limitation, we leave the experimental verification for a future work.

2. The referencing to previous work is not quite complete. For example, trapping of colloidal particles in defect lines demonstrated by others, is not referenced (line 177). This was demonstrated by D. Pires, J.-B. Fleury, and Y. Galerne, Phys. Rev. Lett. 98, 247801(2007) and Skarabot et al (PRE 77, 061706(2008)).

We thank the Reviewer for pointing out the related references. We have added these papers to our reference to help readers better understand the entanglement around particles. “It is known that defect lines engendered in LCs can be used to guide the self-assembly of colloidal particles (20, 34, 35)... In this work, we report the entrapment of colloidal particles along line defects, akin to the entanglement defect structure (a particle attracted by a defect line) shown in (43).”

3. The confocal images in Fig. 1L are not consistent with the simulation and also are not adequately described in the text: "With scanning along z-axis, another four disclination lines near the bottom substrate show up, which finally form the web-like structures when viewed from the z-direction (Fig. 1L) and are agreed with the simulation." What I see from the 3D confocal structure (right panel in Fig.1L) is a kind of wall that extends from one surface to another, and not two distinct layers with disclination lines. This should be made clear in the revision.

We thank the Reviewer for this question. In order to clarify this, we have added two supplementary videos (see new SM3 and new SM4) for the scanning process of confocal experiments, and we have also modified Fig 1E, L-O. In Movie S3, the starting point is where the 1st group of defects is generated; there are four disclinations (bright lines inside thick dark bands) along the same direction (vertical direction), which are close to the top substrate, as shown in Fig. 1L below. They appear bright. With scanning along the z-axis, another four disclination lines near the bottom substrate show up (Fig. 1M below). The spacing between these two sets of disclinations lines, z_c , is about 40 μm . With rotation views of the scanning sample in Movie S4, the disclination close to the top substrate are brighter (Fig. 1N), and the two groups of disclination lines form web-like structures when viewed from the back of the sample (Fig. 1O). Please note that the disclination lines close to the top substrate scatter the laser light so that their signal extends from one surface to the other, like a wall (Fig. 1O); this feature is similar to the confocal result captured in another disclination web structure [Wang et. al Nature Communication 2017, 10.1038/s41467-017-00548-x]. Their two-layer defect lines are also wall-like in the confocal image due to the strong scattering effect.

Fig. 1 **Topological structures and 3D network colloidal assembly in 1D splay-bend cusp-like nematic moiré pattern.** (E) Top and side views of the simulated defect structures at $\Psi = 12^\circ$ when $H/L = 0.9$ (E) (colored by angle β). The 1st group gives disclinations near the top surface and the 2nd group defect is close to the bottom surface. Scan the cell from top to bottom, we observe (L) disclination lines (1st group) close to the top substrate and (M) the disclination lines (2nd group) appearing near the bottom substrate. (N)-(O) Rotation views of the sample. z_c is the spacing between the two groups in the scanning process. Note that the web structure can be seen from the back of the sample due to the scattering of the disclination lines close to the top surface.

4. *The local topology and structure of disclinations and closed loops can be further characterized by two angles, as introduced in Refs. 34 and 35. However, it is not mentioned, in the case of closed loops (Fig. 3G), what is their net topological charge. I assume it is zero, because they can disappear, but it would be helpful to include the concept of the net charge as well.*

We thank the Reviewer for the comments. We have characterized the local profile of defect loops using the angle β shown below.

Fig. S13 Local profiles of four topological loops characterized by β . (A) Loop-I. (B) Loop-II. (C) Loop-III. (D) The typical wedge-twist loop.

We have included the discussions about the topological charge of the closed loops. We have also added the discussions about the topological charge of the loops in the revised Supplement Information. Disclination loops, like hedgehog point defects, carry an integer hedgehog charge [1, 6]:

$$d = \frac{1}{4\pi} \int_{S^2} d\theta d\phi \mathbf{n} \cdot [\partial_\theta \mathbf{n} \times \partial_\phi \mathbf{n}]$$

The topological charge of the three kinds of loops is zero, and they can all disappear they can all disappear or self-annihilate upon rotation or cell gap change.

There are also some technical deficiencies that need to be fixed:

- 1. In line 81, the "gap size H" is not indicated in Figure 1A.*
- 2. In line 83, periodic splay-bend pattern is mentioned to be seen in Figure 1A. However, due to the perspective used in this 3D image, this is not well recognizable. It is suggested to add a top view of such a pattern as an inset.*
- 3. In line 86, the tilt angle theta is introduced, This angle is not indicated in Figure 1A.*
- 4. Line 91: "Psi" is not in Fig. 1A, but in Fig. 1B.*
- 5. Figure 3E: the structures are shown from thick cell to thin cell, just opposite to Figure 1.C, D and E. I suggest to make this coherent, it will be less confusing for the reader.*

We thank the Reviewer for these comments. We have modified the manuscript accordingly

1. We have labelled "H" in Figure 1C to make it clearer and revised line 81.
2. We have added an inset in Fig. 1A to show the periodic splay-bend cusp-like pattern.
3. We have labelled angle θ in the new inset in Fig. 1A.
4. We have modified Line 91 and changed "Fig. 1A" to "Fig. 1B".
5. We have rearranged Figure 3E. Now the structures are ordered from thin cell to thick cell instead.

Reviewer #4 (Remarks to the Author):

The authors carry out experimental and numerical studies on the design of the structures of a nematic liquid crystal cell sandwiched by two patterned surfaces. They prepare two identical surfaces and rotate the top surface with respect to the bottom one to make a "moire" pattern. By one-dimensional (1D) patterned surfaces, they produce different types of configurations with an array of disclinations, both experimentally and numerically, depending on the cell thickness and the detail of the pattern. Their structural properties are explained by a simple theoretical argument. They also demonstrate that the disclination array can serve as a template for the arrangement of colloidal particles. Their strategy is extended also to 2D patterned surfaces.

The authors do not pay due respect for the previous studies presenting the basic and important ideas underlying the present work. Their "theory" predicting the position of disclination lines by the twist angle of the director in the thickness direction has been already exploited by

Ozaki's group [Sunami et al., Phys. Rev. E 97, 020701(R); Ouchi et al., Phys. Rev. Lett. 123, 097801]. They used exactly the same strategy to design almost arbitrary shapes of disclination lines. The authors' statement "This theory can help the inverse design of arbitrary defect structures in future works" (page 15) is therefore invalid.

We thank the Reviewer for suggesting the relevant literature and we have carefully cited them in the revised manuscript. We have also added discussions in the revised manuscript regarding these references:

“Because of the preferred angle mismatch between the two substrates, there is a point-wise twist deformation in the bulk achiral nematic. The handedness of the twist is determined by the acute angle the two preferred orientations make. When this angle transitions from acute to obtuse, there will be a twist reversal, at which the bulk nematic is frustrated. Therefore, we expect that disclinations will emerge where the two preferred orientation angles are orthogonal, i.e., $\Delta\theta = m\pi/2$ with $m = \pm 1, \pm 3, \dots$ (44, 45).”

We admit that the theory is not new. However, previous works have not considered using the moiré effect, i.e., twisting two identical, periodic surface patterns, to engender defects. Our method can give rise to rich defect structures, whose geometry (including periodicity), tunable through the twisting angle, can be well understood by the moiré theory. Our work also shows that the defect-position theory also works well in our nematic moiré patterns if the cell thickness is thin. Our simulation and experiment further point out that the theory breaks down in thicker cells. Therefore, our work presents a thorough examination of the theory in nematic moiré patterns. Moreover, the defects generated in our nematic moiré are structurally new, as we have discussed in Fig. 1 and 3. As such, we have proposed using the moiré effect to rationally design and engineer topological defects with tunable geometries, which can facilitate emergent applications as discussed here and in the relevant literature.

We have also clarified the sentence, "This theory can help the inverse design of arbitrary defect structures in future works" in below:

“Combining the moiré theory and the insight that 90-degree anchoring difference in the thin limit generates defects, we are able to design arbitrary, pixelated region that comprises defect loops, as demonstrated in Fig. 6.”

Disclination lines have long been used for manipulating colloidal particles, and one of the first important contributions was made by Galerne's group [Pires et al., Phys. Rev. Lett. 98, 247801; Fleury et al., Phys. Rev. Lett. 103, 267801]. Ref. 19 (Yoshida et al., Nature Commun. 6, 7180) also demonstrates the formation of colloidal arrays using disclination lines.

We thank the Reviewer for suggesting those references. We have added those papers to our reference list with discussions. “It is known that defect lines engendered in LCs can be used to guide the self-assembly of colloidal particles (20, 34, 35)... In this work, we report the entrapment of colloidal particles along line defects, akin to the entanglement defect structure (a particle attracted by a defect line) shown in (43).”

The motivation of the surface pattern design in Fig. 6 (encoding of characters) and how this

design works are not at all clear. I am also wondering whether the characters are really visible under a microscope.

We have added the following to elaborate on the motivation of Fig. 6:

“Understanding the structure of topological defects is important to many branches of physics, as the defects play a central role in physical processes such as self-assembly and phase transition. A versatile design of highly tunable defects in nematics can be useful for a range of applications, including defect-directed self-assembly, material transport, micro-reactors, photonic devices, and anti-counterfeiting materials. Previously, tremendous research efforts have been devoted to the “forward design” problem: which topological structures with what properties can be formed from a given container or a given pattern. Moreover, for a given anchoring pattern, all the possible defect structures are often-times quite limited to a few metastable states. This greatly limits the applications of nematic disclinations. Here, we propose to use the moiré effect to design arbitrary pixelated defect-regions.”

We have added the following to explain how the design of Fig. 6 works in detail:

“The design of Fig. 6 is inspired by the application of moiré methods in the protection of documents and products [56]. Based on the moiré intensity profiles, new patterns are generated between two specially designed periodic dot screens, one of which being a micro-structured image that is located on the document itself (bottom, green layer, Fig. 6A), while the other (top, pink layer, Fig. 6A) plays the role of a revealer. In the design of anti-counterfeiting materials, the superposed new pattern of a highly visible repetitive moiré pattern of a predefined intensity profile shape and color gives the authenticated properties, documents or products. Combining the understanding that the disclination lines appear where the twist angle of the pattern director has 90-degree difference in a thin cell [44, 45], we propose possibilities of inverse design and map the “document” pattern (bottom) and “revealer” pattern (top) to LC patterns. For a unit on the top pattern (Fig. 6A inset), we let the central circle have $\theta = 100^\circ$ and the other region be $\theta = 45^\circ$; for a unit on the bottom patterns (Fig.6B), θ changes continuously from the central line to the edges of letter shapes, varying from 0° to 45° . We let the surrounding regions in both patterns share the same angle θ so that only where the circle part on the top and letter part on the bottom overlaps, defect loops can be generated. As shown in Fig 6C, an amplified region (in the shape of U, S, and T) consists of multiple defect loops, and we call it a pixelated shape.”

In the simulation, the characteristic size of the pattern is roughly $1.2 \mu\text{m}$. The experiment is using an even larger pattern size. Therefore we believe they are visible under a microscope.

In the last part the authors comment that their strategy may be applicable to display applications because of lower threshold voltage for Frederiks transition. However, I cannot convince myself that their inhomogeneous moire structures with a periodicity of a few tens of microns are really useful for display application. If the periodicity is pushed to a smaller value to avoid the problem of inhomogeneity, the surfaces could just act as an effectively uniform substrate. In the main text the authors give just a vague argument and pushes out all the details, but the description in the main text should be comprehensive enough.

We thank the Reviewer for the comments. We have added more text to clarify the display applications in the revised manuscript. The main mechanism of applying our system for display is using twist deformation of LC to reduce the transition voltage. Theoretically, the transition threshold in a planar twisted nematic cell (TNC) is [De Gennes P. G., Prost J. 1993. Oxford university press.]

$$E_c^2(\Psi_0) = E_c^2(0) \left[1 + \frac{K_3 - 2K_2}{K_1} \left(\frac{\Psi_0}{\pi} \right)^2 \right]$$

where $E_c(0)$ is the Frederiks transition threshold for a unit cell. From the above equation, we can learn that an LC cell with material constant $2K_2 > K_3$ and a large twist angle Ψ should induce a lower threshold than a uniform cell, which is also verified in our simulations. Further, at the same twist angle, our patterned cells show smaller transition voltages than TNC cells (in Fig. S26A).

In our experiments, the size of the patterned area on a substrate is about 1 mm^2 , which is similar to the size of a pixel on a conventional screen. Further, the moiré period is controllable and scale-invariant. Therefore, with a proper fabrication technique, it should be possible to expand our system to a much larger size and fabricate a display device using our moiré structures. We have added the above discussions in the revised manuscript.

I also have a concern about the confocal images (Fig. 1L). How they should be interpreted is not explained in the main text or Methods (Should one see local liquid crystal orientation as in POM?). How one should understand the right panel of Fig. 1L (confocal image) is not presented, either. The authors state that the disclination lines are localized in the vicinity of the surfaces, but the center and the right confocal images do not seem to support their claim; rather there appear to be domain walls spanning the thickness direction.

We thank the Reviewer for this question. We have added two supplementary videos (see the new SM3 and SM4) for the scanning process of confocal experiments, and we have modified Fig 1E, L-O for clarification. In Movie S3, the starting point is where the 1st group of defects is generated; there are four disclinations (bright lines inside thick dark bands) along the same direction (vertical direction), which are close to the top substrate, as shown in Fig. 1L below. They appear bright. With scanning along the z-axis, another four disclination lines near the bottom substrate show up (Fig. 1M). The spacing between these two sets of disclinations lines, z_c , is about $40 \mu\text{m}$ (the cell thickness is about $50 \mu\text{m}$). Therefore, the disclinations are localized near the two surfaces. With rotation views of the scanning sample in Movie S4, the disclinations close to the top substrate are brighter (Fig. 1N), and the two groups of disclination lines form web-like structures when viewed from the back of the sample (Fig.1O). Please note that the disclination lines close to the top substrate scatter the laser light so that their signal extends from one surface to the other (Fig. 1O, Movie S4), and the configuration is two groups of disclination lines, rather than a domain wall. Our result is similar to the confocal result in another disclination web [Wang et. al Nature Communication 2017, 10.1038/s41467-017-00548-x], which also consists of two layers of disclination lines. Their two-layer defect lines are also wall-like in the confocal image due to the strong scattering effect.

Fig. 1 **Topological structures and 3D network colloidal assembly in 1D splay-bend cusp-like nematic moiré pattern.** (E) Top and side views of the simulated defect structures at $\Psi = 12^\circ$ when $H/L = 0.9$ (E) (colored by angle β). The 1st group of disclinations is near the top surface and the 2nd group defect is generated close to the bottom surface. Use the confocal microscopy to scan the cell from top to bottom and we observe (L) disclination lines (1st group) close to the top substrate and (M) the disclination lines (2nd group) appearing near the bottom substrate. (N)-(O) Rotation views of the sample. z_c is the spacing between the two groups in the scanning process. Note that the web structure can be seen from the back of the sample due to the scattering of the disclination lines close to the top surface.

Regarding the 2D pattern case, the authors say that the choice of the rotation center does not affect the defect configuration (p13 of the main text, and p24 of the Supplementary PDF). However, the 2D system lacks continuous translational symmetry, and indeed Movie S10 demonstrates the effect of the choice of the rotation center on the detailed structure of the disclinations.

We thank the Reviewer for this comment. We have clarified the relevant discussion in the revised manuscript. Mathematically, choosing a different rotation center is equivalent to translating the moiré pattern. It is known from the moiré theory that the relative translation of the two patterns does not qualitatively change the emergent pattern except inducing a translation along the orthogonal direction. We have demonstrated in our 2D pattern that the translation of the top surface can lead to the translation of the defect structure along the orthogonal direction (Movie S12), which preserves the defect period and tilting angle at a small rotation angle.

We have also performed more simulations to clarify the effect of the rotation center. We demonstrate two kinds of periodic defect configurations, $\Psi = \tan^{-1}(3/4)$ and $\Psi = 5^\circ$ ($\Psi \leq 21^\circ$). For the former, the change of rotation center can affect the defect configurations since the special period comes from the periodic overlapping of $+1(-1)$ defect points on the patterns, as shown in Fig. S23; For the latter, the choice of rotation center affects the generated defect configuration in the simulation window, but the nematic pattern (e.g. period and tilting angle) is not varying with the rotation center.

Fig. S23 Defect configurations for different rotation centers at $\Psi = \tan^{-1}(3/4)$. Inset is a schematic showing the rotation center point. (A) The center of the box as the rotation center. (B) The -1 defect at the corner as the rotation center, $(C_x, C_y) = (0, 0)$. (C) A point in the middle of two neighboring $+1$ (-1) defect pairs, $(C_x, C_y) = (L/2, 0)$. (D) A point in the middle of a $+1$ and a -1 as the rotation center, $(C_x, C_y) = (L/4, L/4)$. (E) Another point in the middle of two neighboring $+1(-1)$ defect pairs, $(C_x, C_y) = (0, L/2)$. (F) A $+1$ defect as the rotation center, $(C_x, C_y) = (L/2, L/2)$. (G) Another -1 defect as the rotation center, $(C_x, C_y) = (L, L)$.

Fig. S24 Defect configurations for different rotation centers at $\Psi = 5^\circ$. Inset is a schematic showing the rotation center point. (A) The center of the box as the rotation center. (B) The -1 defect at the corner as the rotation center, $(C_x, C_y) = (0, 0)$. (C) A point in the middle of two neighboring $+1$ (-1) defect pairs, $(C_x, C_y) = (L/2, 0)$. (D) A point in the middle of a $+1$ and a -1 as the rotation center, $(C_x, C_y) = (L/4, L/4)$. (E) Another point in the middle of two neighboring $+1$ (-1) defect pairs, $(C_x, C_y) = (0, L/2)$. (F) A $+1$ defect as the rotation center, $(C_x, C_y) = (L/2, L/2)$. (G) Another -1 defect as the rotation center, $(C_x, C_y) = (L, L)$.

As the authors state in the Introduction, moire patterns have been attracting great attention in various fields of condensed matter physics, and therefore the present study might appeal to broad interest. However, considering the fact that the underlying ideas towards the design of disclinations have been already exploited, and the lack of clarity in the interpretation of the confocal images, I do not think this work merits publication in a high-profile journal such as Nature Communications.

We appreciate the Reviewer for recognizing the novelty of our work. The design of disclinations using patterned surfaces has been discussed in several prior publications. In these works, the key insight or design principle is a simple defect-position theory, which states that disclinations in the patterned nematic cell should appear where there is a twist reversal due to a 90° difference in the surface-preferred angles. In the revised version we have appropriately cited these works. As we point out, although the simple theory has been proposed before, it is for the first time to be examined in the context of nematic moiré patterns in which two identical, periodic surface patterns are relatively twisted by an angle; in fact, our simulation and experiment show that the theory still works for thin cells, but fails to predict the rich defect structures in thicker cells. Therefore, our work presents a thorough examination of the theory in the context of nematic moiré patterns. Building upon these efforts, we propose a rational inverse design principle, i.e., the moiré effect, to create and engineer highly tunable defect structures. By changing the twisting angle and the cell gap, we can achieve novel defects structures with tunable geometries, which can be understood by the moiré theory. As is discussed in our work, these defects are structurally different from those reported in existing works. Therefore, our design principle is not merely the simple theory, but the moiré effect, based on which we are able to generate novel defects. Following the Reviewer's comments and suggestions, we have incorporated all the necessary changes. Specifically, we have clarified our confocal experimental results, cited all the relevant papers, and elaborated on Fig. 6 in the revised manuscript.

I also have several concerns about the presentation as listed below:

** The authors put too much information using small characters in individual small panels of the figures without caring about visibility. For example, the geometrical parameters in Fig. 2A, B are hardly visible. Superimposed top and bottom surface patterns in Figs. 1B, 3B, S2B, and S8B are hardly recognizable. Visibility of Fig. S3 is quite poor.*

We thank the Reviewer for pointing this out. We have enlarged the font size of geometrical parameters in Fig. 2A, B. We have redesigned Figs. 1B, 3B, S2B and S8B (now Fig. S10B). Fig. S3 is also modified to improve its visibility (now Fig. S4).

** The meaning of "isotropic moire" is not clear. Does it simply mean moires without a liquid crystal?*

We have added more explanations in the revised manuscript. "Isotropic moiré" in our manuscript means the most well-known moiré effects in the black-and-white geometric gratings. Those gratings (or lattices) don't exhibit orientation-dependent material anisotropy. Here, we explore moiré effects using nematic liquid crystals, which exhibit birefringence. Therefore, to better distinguish this difference and to better demonstrate the features that only exist in liquid crystal moiré, we specify those previous moirés generated in isotropic systems as "isotropic moiré". Isotropic moiré patterns in our work refer to periodic moiré patterns arising from geometric lattices without anisotropy.

** (page 6) Is the top surface really at z=0?*

We removed the descriptions to avoid misunderstanding.

** Some results are presented without statements on their implications. In pages 7 and 8, the authors just list up the local structural properties without any insights given (I also note that "local topology" is not suitable because topology refers to properties invariant under continuous transformations). What can readers learn from the plot of A_{xy} (Fig. 2F,G) ?*

We appreciate the Reviewer's comments. Additional discussions following the topology characterization of the defects have now been included in the revised manuscript. Distinct from 2D liquid crystals, in 3D nematics, a $+1/2$ and a $-1/2$ wedge can be transformed into each other through a continuous transformation, with the symmetric twist state found in between. This continuous transformation of the local profiles can be used to explain the defect state transitions (between the S-, C- and W-state) as we tune the cell gap.

We have also added more discussions about A_{xy} . The amplitude of the defects A_{xy} in the C-state increases as Ψ decreases (Fig. 2F, Movie S2) or H/L increases (Fig. 2G, Movie S1). Upon the transition from the C-state to the W-state, A_{xy} approaches T as the neighboring wavy-like defects intersect and start to form web-like defects (Fig. 2F, G, Movie S1).

We agree with the Reviewer. "Local topology" can be confusing. We now use "local profile" instead.

** (Page 8) Is wedge winding really characterized by $\beta \neq \pi/2$?*

We have clarified this sentence. $+1/2$ wedge winding and $-1/2$ wedge winding are characterized by $\beta = 0$ and $\beta = \pi$, respectively.

** (Page 12) The dynamics of defect loops is not clearly explained. How and when does the shrinkage of disclination loop occur in the absence of colloidal particles?*

We have added more explanations. In the thin cell experiments, Loop-I is unstable when the generated shape is smaller than ($\sim 1200 \mu\text{m}^2$). Due to the energy reason, the small loop gradually shrinks and eventually annihilates. In Fig. 3H (right), the smaller loops also shrinks

during the equilibration process (after about 30s after taking the photo). We characterize the shrinking process in Fig. R4 shown in below.

Fig. R4 Loop shrinkage without particles (Fig. 4C).

** The detailed structure of the 2D imprinted pattern is not explained.*

We have modified Fig. 5A in below (a planar view of the 2D imprinted pattern is added), and we have changed the text.

** In Fig. 5B and D, does a dashed orange (or blue) square in the two panels really correspond to the same region?*

Yes, they correspond to the same regions. We have clarified that in the revised caption.

** [Fig. 5 G, H] What are the green and blue regions, and the blue lines in the latter?*

We add an explanation in the Supplementary Material and the caption of Fig.5. Yellow color is for the $\Psi \leq 21^\circ$ regime with $(1,0,-1,0)$ as the dominating moiré, and blue color is for the $\Psi \in [34^\circ, 40^\circ]$ regime with dominating moiré $(1,2,-2,-1)$. For the other angle ranges, $\Psi \in [21^\circ, 34^\circ]$ and $\Psi > 34^\circ$, no dominating moiré is observed (using green color)

** [Theoretical model of numerical simulations (page 20-22, and the Supplementary PDF and movies)]*

** The one-constant approximation usually refers to setting $K_1=K_2=K_3$, not $L_1=L_2=L_3=L_4$. In the Supplementary PDF (p26), the authors inconsistently assumes the former.*

We thank the Reviewer for this comment. We have moved this part to the Supplementary Material 6.2.1 and have changed it.

** (Just after eq. 8) What do the authors mean by the average in the definition of the surface field Q_s ?*

We thank the Reviewer for this point. We have changed the expression of Q_s to resolve the confusion, $Q_s = S(\mathbf{n}_s \mathbf{n}_s - \frac{\mathbf{I}}{3})$.

** If the authors are interested only in equilibrium (or metastable) structures the definition of the time τ is not important, and it is safely regarded as "the relaxation step". However, they present the "dynamics" of disclination lines in response to the rotation of one surface or the applied field (Supplementary videos S11 and S12). Then the unit of time should be clearly presented. It is not clear either what is the relation between τ_{cell} and τ_0 .*

The characteristic time scale for the relaxation of the nematic director field is $\tau_{\text{cell}} = \frac{\gamma_1 H^2}{L_1} = \frac{2S^2 H^2}{\Gamma_s L_1}$ with γ_1 being the rotational viscosity, H being the cell gap, and L_1 being the elastic constant. This is an intrinsic time scale in the simulation. The simulation time $\tau_s = N\tau_0$ is expressed in the simulation time unit τ_0 (N is the number of simulation steps). To achieve a quasi-static simulation, $\tau_s \gg \tau_{\text{cell}}$ has to be satisfied. Otherwise, defect structure could be different and path-dependent during the rotation operation. We have now emphasized the above in the revised manuscript.

** Just after eq. (10), the sentence "For surface points, ..." does not seem to make sense.*

We agree with the Reviewer. We have removed that sentence.

** The anchoring strength W , the field strength E and the dielectric anisotropy ϵ_a are not given.*

We have added more descriptions. W is infinitely large in simulations. The dielectric anisotropy is $\epsilon_a = 11.5$ since the material in the experiments is 5CB. E is varied in our simulations in simulations. We fix the cell height H for the simulations in the Supplementary Material section 5.2 and we keep increasing E .

** In Supplementary video 12, it seems that the field is turned on and off somewhere, but it is not mentioned in the caption.*

We thank the Reviewer for this comment. In video 12 (now video 14), we let the system form the S-state first (the starting frame); then, we turn on the electric field and let the system

relax. Eventually, the directors in the middle layer are aligned to the electric field direction (z). We have added the text “Electric field on” in the revised Movie S12 (now S14).

** What was the cell gap H chosen in the simulations in Fig. S22? Is it the same in all the simulations? (If so, two structures are metastable).*

We have clarified that part. To better compare the transition voltage, all the simulations in Fig. S22 have the same cell gap H (and the same cell size), $H \approx 30 \times 6.63 \text{ nm} = 198.9 \text{ nm}$. But the lattice constant is tuned to induce different defect states (S, C, and W), according to the state diagram in Fig. 1F. The states are thus not metastable.

** In Tables S1 and S2, the value of the free energy is meaningless unless the system size is specified. I am also wondering why the value of the free energy in Table S1 is so different from that in Table S2.*

We have added the simulation sizes in Section 6.2.2 in the SI. Yes, the energies in Table S1 should be higher since we use a thicker box with a larger cell gap H to get the W-state (compared to the C-state).

** (Section 6.2.1 of the Supplementary PDF) It is not clear what the “four different conditions” are. Does θ specify the azimuthal angle of the uniform alignment of the liquid crystal? What are the symbols like (1,0,0) in Figs. S29 and S30?*

We have modified the sentences in Section 6.2.1. “The first three initial conditions have uniform bulk director, $\theta = 0$, $\theta = \pi/2$, and $\theta = \pi/4$ (θ is the in-plane tilting angle of the director with the x -axis); the fourth condition is a random condition. For clarification, we changed (1,0,0) to $\theta = 0$ in Figs. S33 and S34.

** (Section 6.2.2 of the Supplementary PDF) The definition of τ_s is different from that in the Methods section.*

The definition should be the same. We have modified Section 6.2.2 of the Supplementary PDF to make it clearer.

** [p 23] \mathbf{V} could be referred to as the Jones vector. Instead of Ref. 51, a standard textbook or review on liquid crystal optics (or optics in general) could be cited.*

We have cited a book and a review for matrix \mathbf{V} (We have moved that part to Supplementary 6.2.5).

** [Caption of Fig. S1] I could not find the definition of the angle θ_g .*

We have modified this part in the revised Supplement Information. θ is the tilting angle of the surface director \mathbf{n}_s . Different from the angle θ , θ_g varies between 0° (mapped to white color) and 90° (mapped black color), and $\theta_g = ||\theta - 90^\circ| - 90^\circ|$.

** The definition of the 2D rotation matrix in eq. (S2) is not consistent with that in eq. (12).*

We have moved the previous Eq. (12) to the Supplement and renamed it as $\mathbf{R}_0(\alpha)$.

** [Supplementary PDF, pages 17 and 20, and Fig. S17] Regions I and III are not consistently defined.*

We have corrected them in the Supplementary PDF.

** The authors should carefully check cross references:*

** (page 8) Does Ref. 31 really concern the sentence "The angle alpha of the S-state changes from 0 to 2pi in a lattice period L: this contrasts with the line defect with uniform aloha in similar systems in which the top substrate adopts a uniform anchoring"?*

We have changed Ref. 31 to Ref. 34.

** (page 18) Fig. S25D should be Fig. S27D.*

** (Supplementary PDF, p12) Fig. 2D should be Fig. 2E.*

** (Supplementary PDF, page 24) Movie S11 should be Movie S10.*

We have fixed those.

** There should be more standard references that replace Refs. 48 (orientation of azo dye in response to linearly polarized light. For example, Chigrinov, Kozenkov and Kwok, "Photoalignment of Liquid Crystalline Materials: Physics and Applications") and 49 (hedgehog defect accompanying a spherical particle with normal anchoring. For example, Poulin et al., 275, 1770).*

We have replaced those Refs. 48 and 49 (now Refs. 67 and 68) with the suggested literatures.

** The text is well written, but contains several typographical/grammatical errors:*

** (page 6) "are agreed with" -> "agree with"*

** (page 24, just before eq. 16) "angle" is doubled.*

** (page 25) "We thank the fruitful discussions S.Y. ..." looks strange.*

** (Page 5 of the Supplementary PDF) " $\theta(x,y,z)$ in the vicinity of the top surface" is not properly located.*

** (page 14 of the Supplementary PDF) "this predict" -> "this prediction"*

We have corrected those typographical/grammatical errors.

We are again grateful to the Reviewers for their careful and thorough reading of this manuscript.

REVIEWER COMMENTS

Reviewer #1 (Remarks to the Author):

The authors have substantially improved the manuscript. Unfortunately, there are several technical errors. In the main text are missing links to the explanations in the Supplementary information. There are only two points where more explanation should be added mostly in Supplements and Methods: optical simulations and numerical simulations. With the suggested improvements I expect that the paper will be ready to go on. The remarks listed below are inserted in the rebuttal letter related to pages of the original version of the manuscript, and only a few are directly related to the line numbers of the new version.

Page 2

.
. .
.

While listing manipulation methods consider also a partially related recent paper by R. Selinger in the Arxiv.

We thank the Reviewer for bringing up this related work. We have cited this arXiv paper in the revised manuscript.

The reference is still missing!

Page 5

.
. .
.

It would be good to stress that deformations and the value of H/L where they start to appear, probably strongly depend on the anisotropy of elastic constants.

We agree with the Reviewer that the defect states strongly depend on the anisotropy of the elastic constants of the nematic. We have performed additional simulations and added a new Fig. S35 and the following discussions in the revised Supplement Material 6.2.4:

In this work, we focus on one-elastic-constant assumption, i.e., $K_1=K_2=K_3=K_{24}$. Our additional calculations showNevertheless, the two choices of the elastic constants do not qualitatively change the defect state diagram. Without loss of generality, we focus on the one-constant approximation in this work.

In the main text, it should be mentioned that justification for the single constant approximation is given in Supplementary Information 6.2.4 and illustrated in Fig. S35.

What determines the left/right type of disclination helices?

The rotation direction can tune the handedness of the disclination helix. For the disclination helices reported in Figure 1 and 2, we rotated the top substrate counterclockwise and obtained right-handed helices. When a clockwise rotation is applied to the top substrate, left-handed disclination helices are generated. We have included the above discussion and the new Fig. S5 in the revised Supplement Material 2.4. ...

Also, here a similar hint should be given in the main text.

Page 6, 11, 23, 32, 34, and S9

The Fig S4 mentioned in the relation to simulated optical microscope images opens some questions. Usually Jones approach do not properly present disclination lines as cores are too tiny. What was done to get rather thick dark lines in simulations of S and C states? In the Methods I do not see any note on this point!

For the Jones approach, dark areas correspond to the regions where polarized ray in the layer mostly passes areas where the director is either parallel or orthogonal to the polarization. Therefore, a

disclination can be indirectly seen if the director field above the line satisfies the mentioned condition. As lines look very thin the director should have a narrow range where this condition is realized. In a Mauguin limit for a very particular director distortion, the polarization can also keep the right orientation. The numerically obtained director field can disclose this unfortunately, the presented profiles in the Supplements are too crude to see the director field. It would be good to add a discussion of this background. For thin samples, the Jones approach is good while for the thicker ones where focusing, ray deflection, and oblique rays become relevant the open code also used is better.

There is another problem which causes possible troubles. The numerical solution for the nematic structure is scaled for a factor of ~ 300 . There is no problem with the director field. The problem is that the defect core with a depressed order parameter cannot be simply scaled. It would be nice to test how relevant this is for POM. For a qualitative description, it is certainly not relevant but for optical applications, it is certainly relevant. It would be good to comment on it at some point.

Page 7-8

The analysis of topology of disclinations on these pages refers to the introduction of disclination topology in the Materials and Methods. That text is fine for specialists but a broader readership needs a graphical presentation of all disclination profiles with typical α and β angles that appear in this paper. It should go beyond what is now in Fig 1 G, H. The new figure should probably go in the Supplement or Materials and Methods if allowed.

Following the Reviewer's suggestion, a new Fig. S3 illustrating α and β shown in below has been added to the revised Supplementary Information.....

In appropriate points in the main text, Supplementary Information 2.4 and Figs S3, S4, and S5 should be mentioned.

Page 10

In the previous case the lines do not change much when ψ angle is small but their separation decreases with increasing ψ angle. How is here when ψ angle starts from zero? Probably increasing ψ loops enter the observation window from the infinity and become closed when they are small

enough to fit in the window. In simulations it is important what you do with boundary conditions. It would be good to show how the aspect ratio change with growing ψ ? All this should be explained in more details in the supplement using theory or simulations!

Topological charge of the loop should be mention. Here loops should have charge zero so that can disappear if not prevented by surroundings.

We thank the Reviewer for these suggestions. We have calculated the aspect ratio of the series of simulations with $H/L=0.1$ in the revised Supplement Information. As we increase the angle Ψ from 0° , the defects first enter the simulation window around $\Psi=6^\circ$, as shown in Fig. S14A....

At appropriate points in the main text, Supplementary Information 3.2 & 3.3 and Figs. S12, S13, & S14 should be mentioned.

Page 11

Also, here the same remark as for Pages 7-8 applies!

Page 12

The presented surface defects require strict planar anchoring. How strong it is in the experimental setup and what was used in simulations?

We thank the Reviewer for this question. In the experiment, the anchoring strength is in the order of $\sim 10^{-3}-10^{-4}$ J/m² (strong anchoring) (48). The anchoring strength W is set to be infinite in the simulations (the surface preferred director n_{sis} fixed) to approximate the strong anchoring condition in the experiment. The good match between the experiment and simulation in different scenarios validates the strong anchoring assumption used in the simulation. We have added the above clarifications in the revised manuscript.

This is OK if there are no surface defects! See remark related to page 19

Page 13

More details are needed for $\psi=0$ structures: There are plus and minus defects and must be a coherence with neighbors how disclinations form quasi loops. It would be good to add a simulation with disclination loop profiles. What happens with a nematic between loops and surface defects where order is depressed?

We have modified Fig. S17 and added more details about the quasi-loop defects in the Supplement Information 4.2.

We would like to also thank the question about the nematic between loop defects and surface defects. As shown in Fig S17C-E,...

Also, here including hints about Supplementary Information 4.2. and Fig S17 in the main text should be done.

Page 19, it applies also to page 22

Here I repeat: Anchoring strength for in-plane and out of plane deviations should be presented as it certainly strongly effects experimental structures in thin samples. What was assumed in simulations?

In the experiment, we applied strong anchoring and we don't evolve the n_s (the surface molecular orientation) in the energy minimization calculation to approximate the strong anchoring condition in the experiment (70).

Listing W values of the nematic anchoring strength and stressing that for simulation of the director fields the approximation of infinite W is good enough is OK only if there are no surface defects. The problem is the nematic order parameter S . If you assume that it is constant, local free energy diverges and the approximation of infinite W does not work anymore. Usually, finite W is used but it is true that this substantially complicates the numerical procedure. A possible (not elegant) way out

is assuming that in the defect core S is zero. It should be clearly explained what was done here, probably in Methods and Supplementary.

Page 20-21 and 22

The paper is quite long therefore the Section "Theoretical model of numerical simulations" should be more compact as the approach is also well known. The technical details like different elastic constants and Frank form should go to the Supplement.

We thank the Reviewer for this advice. We have moved more technical details in the "Theoretical model of numerical simulations" to the Supplement (6.1.1).

This is OK

Further in this segment, I comment according to the lines of the revised text:

522: Add here an expression for the nematic coherence length that now appears later. Additionally, you must stress that flow effects are neglected and you will follow only relaxation of the orientational ordering.

537-539: Here I repeat the above question related to Page 19.

542-543: Here you should describe that Γ relates to the well-known Leslie's viscosities and sets up the typical timescale of the dynamical processes in the nematic at a selected length scale. From Eq.8 usually, a simple expression for a characteristic time is obtained: the ratio of the square of the particular length divided by $\Gamma \times L^2$.

546-547: On page 6 etc. I have stressed that there is a problem with scaling, which is OK for the director field but could not be simply applied to the defect cores with a depressed order parameter. This should be commented

547-548 Describing the selection of lattice point distance related to the periodicity is not enough. The best would be to present the size of the complete simulation box with lattice point distances in all three orthogonal directions. Here also boundary conditions should be explained and the expected weakness of the solution close to free boundaries. It looks like they are cut in the presented results. More details for all structures should be in the Supplement 6.2.1.

548-553: The discussion on time scales does not look OK. Including undefined terms, not clearly describing dimensionless quantities, mentioning Lattice Boltzmann simulations used for flows, etc. Using the above-mentioned expression for times related to characteristic scales: nematic coherence length, cell thickness, and lateral system size should be related to the needed time interval and number of repetitions. Usually time intervals are a few times shorter than the nematic coherence time. In periodic structures starting with small perturbations computing time must surpass the time related to the layer thickness, while starting from a random state would require times longer than the one related to the lateral size.

554: Here it would be good to mention that the finite difference relation approach was used. Your code or public code etc. Some details can go to the supplement.

Pages S33-S34

One usually adds quasi thermal noise to reduce such effects.

We thank the Reviewer for this comment. We have used different initial conditions including a random initial configuration, and compare their free energies instead. We have added the discussions about quasi-thermal noise in Supplement (6.2.2). ... In our simulation with electric fields, we added random noise to the initial director field (very small, nonzero component n_z) to study the Frederiks transition, which shows a good match with the theory (Section 5 in SI). Without the initial noise, the transition requires a four times larger transition threshold in the simulations.

In Supplementary 5 this is not added in detail so that one cannot reproduce the procedure.

Reviewer #2 (Remarks to the Author):

In the revised manuscript, the authors have fully addressed the comments, questions, and suggestions of four reviewers. The title, many parts of the main text, the list of references, the supplementary information, and the videos have been substantially revised and clarified. I thank the other reviewers for their accurate and comprehensive reports, which helped the authors improve the manuscript in many ways. I also appreciate the authors' efforts to review the effects of anisotropy of elastic constants, the use of an additional open-source package for simulating optical patterns, and their provision of clear explanations of noted deficiencies. The authors are aware of the limitations of their approach, and on the other hand, they also suggest how to use the moiré effect to address the inverse problem of designing arbitrary pixelated defect regions. I have no further comments or suggestions for improvement, and I think the manuscript is good enough to be published in Nature Communications in its present form.

Reviewer #3 (Remarks to the Author):

The authors have answered all comments and questions that were raised in my previous report and I recommend acceptance of this manuscript in its present form.

Reviewer #4 (Remarks to the Author):

The authors have addressed many of the concerns raised by the four referees including me. However, there are still many problems in the presentation (several issues should have been commented in the 1st report), and therefore I do not recommend the publication of this manuscript in its present form.

As one of the referees says, this manuscript is too long. I do not accept the authors' mere statements "something has been added in the revised manuscript" that let the referees look through the whole 42-page manuscript or 44-page supplementary material.

* Present tense ("enables") is natural for the title.

* [Page 10] Axy is referred to as "projected helix diameter" in lines 208-209, and "amplitude of the defects" in line 226. This could be confusing.

* [Page 15, lines 330-331] The statement " $\tan^{-1}(3/4)$ is a singular point in the moire theory." does not practically say anything. What results from the fact that $\tan^{-1}(3/4)$ is a singular point? I also note that it has been already stated in lines 326-327 that $\tan^{-1}(3/4)$ is a special angle.

* [Materials and Methods] The authors do not mention anything about from which company they bought JXLC-20000 (or how they synthesized it), or why they used it instead of 5CB in confocal microscope observation.

* [Page 23] It is well known that biaxiality arises in the vicinity of the defect core when the orientational order is simulated by a second-rank tensor (See e.g., Lyuksyutov, Sov. Phys. JETP 48, 178 (1978); Schopohl and Sluckin, Phys. Rev. Lett. 59, 2582 (1987); Penzenstadler, and Trebin, J. Phys. 50, 1027 (1989)). I agree with the authors that the order is practically uniaxial in the bulk. Nevertheless, if they are doing calculations with the constraint of $P=0$, their calculations are different from standard ones carried out in previous studies such as Ref. 61.

* [Pages 24-25, eq. (6)] I understand that the Rapini-Papoular form of the surface energy is in fact not used in their calculations if W is assumed to be infinite.

* [Page 25] The authors say that τ_0 (and hence τ_S) does not have physical meaning. But still they compare τ_S with τ_{cell} ; the latter is given by physical parameters. This comparison does not make sense.

* [Confocal images in Fig. 1] A fundamental issue is not addressed: What should appear dark (or bright) in their confocal images?

The difference between the panels N and O should be clearly stated in the caption or the main text.

* [Fig. 5] The authors say that in Fig. 5B and D, the dashed orange (or blue) square corresponds to the same region. Obviously the relative position of the orange square with respect to the blue one is different.

The explanation of the black theoretical lines in the blue regions of Fig. 5G and H is still missing (In the 1st report I erroneously mentioned the black lines as blue ones). They are not compared with simulations or experiments.

* [Supplementary PDF, Sec. 5.1] Three different symbols, L, d and H, have been used for the cell thickness.

* Fig. S14 does not have panels I and J.

* I recommend the authors to present Figs. S17D and E from a different viewpoint: I misunderstood that two close loops were located vertically.

* Fig. S23 is not referred to in the main text.

* What one can learn from Fig. S25 is not at all mentioned.

* [Fig. S26] If the cell thickness is the same for the three simulations as the authors say, the difference in the lattice constant must be clearly state there.

* [Figs. S4, S17, S33, S34] The color scale should be for β/π (as in Fig. 1), not β .

* [Figs. S27-29] The "V" in "0.82V" should be roman, not italic.

* [Supplementary PDF, Sec. 6.2.5] The authors response "we did not artificially enhance the disclination lines in our simulated optical images" seems to contradict with the introduction of n'_E with a comment, "We apply this modification because the the Jones-matrix approach is not good in simulating thin disclination lines."

The value of λ is not given in the calculated POM images.

- * There are still typographical/grammatical errors:
- * [page 2, line 30] "planck" -> "plank"? (or, "disk-like" is more commonly used in the community.)
- * [page 15, line 334] "later" is not a verb.
- * [page 25] The phrase "comparing to $L=75\mu\text{m}$ in the experiment" does not seem to be placed appropriately in this sentence.
- * [Page 34, line 808] "Use" -> "Using"?
- * [Supplementary PDF, page 17] "emergency" -> "emergence"?
- * [Supplementary PDF, page 17] "they can all disappear" appears twice.
- * [Supplementary PDF, page 31] The semicolon that follows "in the simulations" should be colon.
- * [Supplementary PDF, page 41] "Fig. S4" -> "Fig. S4"

Point-by-point Response to the second round of Referee reports

We sincerely thank the Editor and all the Reviewers for your valuable time and constructive feedback. We are sorry for the remaining issues. According to your suggestions, we have made extensive corrections to our previous version. In particular, we have added line numbers and marked all the new changes in red in the revised manuscript. With the changes detailed below, we hope you will find the manuscript and our response satisfactory. For clarity, the reviewers' comments are in *blue*, and the new changes we have made are in *red* in the revised manuscript and the revised Supplementary Information (the changes from the first round are kept *blue* in the main text).

REVIEWER COMMENTS

Reviewer #1 (Remarks to the Author):

The authors have substantially improved the manuscript. Unfortunately, there are several technical errors. In the main text are missing links to the explanations in the Supplementary information. There are only two points where more explanation should be added mostly in Supplements and Methods: optical simulations and numerical simulations. With the suggested improvements I expect that the paper will be ready to go on. The remarks listed below are inserted in the rebuttal letter related to pages of the original version of the manuscript, and only a few are directly related to the line numbers of the new version.

We would like to thank Reviewer #1 for the careful checks and detailed suggestions. We have addressed Reviewer #1's concerns and also included in the revised main text the hints to all the Supplementary Information sessions and figures as detailed below. We believe that the quality of our work has been further improved with those modifications.

Page 2

While listing manipulation methods consider also a partially related recent paper by R. Selinger in the Arxiv.

We thank the Reviewer for bringing up this related work. We have cited this arXiv paper in the revised manuscript.

The reference is still missing!

We actually did have added the paper using patterned surfaces. It's [26] in the reference list: C. Long *et al.*, Frank-Read Mechanism in Nematic Liquid Crystals. *arXiv preprint arXiv:2212.01316* (2022). We have now highlighted it in the revised manuscript below:

Changes in the main text:

Page 2 Line 36–38

Existing defect manipulation methods include magnetic and electric field actuation (9, 11, 12), optical control (13, 14), active stresses (15, 16), curvature imposed by boundaries (17-19), **patterned substrates (20-26)**, and chemical interactions (5, 27).

Page 30 Line 690

26. C. Long et al., Frank-Read Mechanism in Nematic Liquid Crystals. arXiv preprint arXiv:2212.01316 (2022).

Page 5

It would be good to stress that deformations and the value of H/L where they start to appear, probably strongly depend on the anisotropy of elastic constants.

We agree with the Reviewer that the defect states strongly depend on the anisotropy of the elastic constants of the nematic. We have performed additional simulations and added a new Fig. S35 and the following discussions in the revised Supplement Material 6.2.4:

In this work, we focus on one-elastic-constant assumption, i.e., $K_1=K_2=K_3=K_{24}$. Our additional calculations showNevertheless, the two choices of the elastic constants do not qualitatively change the defect state diagram. Without loss of generality, we focus on the one-constant approximation in this work.

In the main text, it should be mentioned that justification for the single constant approximation is given in Supplementary Information 6.2.4 and illustrated in Fig. S35.

We thank Reviewer #1 for this criticism. We have now included the hint to the new SI section and Fig. S35 (now Fig. S37) in the revised main text detailed below.

Changes in the main text:

Page 5 Line 111–113

For simplification, the one-elastic-constant assumption is applied in our simulations, and a comparison with elastic constants of real materials is discussed in Supplementary Information 6.2.4 and Fig. S37.

What determines the left/right type of disclination helices?

The rotation direction can tune the handedness of the disclination helix. For the disclination helices reported in Figure 1 and 2, we rotated the top substrate counterclockwise and obtained right-handed helices. When a clockwise rotation is applied to the top substrate, left-handed disclination helices are generated. We have included the above discussion and the new Fig. S5 in the revised Supplement Material 2.4. ...

Also, here a similar hint should be given in the main text.

We thank Reviewer #1 for this comment. We have now included the hint to the SI section and Fig. S5 in the revised main text detailed below.

Changes in the main text:

Page 6 Line 120–121

The handedness of the C-state defects can be reversed by changing the rotation direction (Supplementary Information 2.4 and Fig. S5).

Page 6, 11, 23, 32, 34, and S9

The Fig S4 mentioned in the relation to simulated optical microscope images opens some questions. Usually Jones approach do not properly present disclination lines as cores are too tiny. What was done to get rather thick dark lines in simulations of S and C states? In the Methods I do not see any note on this point!

For the Jones approach, dark areas correspond to the regions where polarized ray in the layer mostly passes areas where the director is either parallel or orthogonal to the polarization. Therefore, a disclination can be indirectly seen if the director field above the line satisfies the mentioned condition. As lines look very thin the director should have a narrow range where this condition is realized. In a Mauguin limit for a very particular director distortion, the polarization can also keep the right orientation. The numerically obtained director field can disclose this unfortunately, the presented profiles in the Supplements are too crude to see the director field. It would be good to add a discussion of this background. For thin samples, the Jones approach is good while for the thicker ones where focusing, ray deflection, and oblique rays become relevant the open code also used is better.

We thank the Reviewer for this comment.

(1) We have added some discussions in the revised main text detailed in below.

Changes in the main text:

Page 7 Line 139–142

The Jones-matrix approach is good for thin samples (Fig. S6A, B), while for thicker ones where focusing, ray deflection, and oblique rays become relevant (Fig. S6C), the director field cannot be clearly seen (Supplementary Information 6.2.5).

(2) We have also included a new discussion and a new figure in the revised **Supplementary Information**.

Changes in the Supplementary Information:

Page 10 in Sec. 2.4

Notably, the Jones-matrix approach is good for thin samples (Fig. S6A, B) while for thicker ones where focusing, ray deflection, and oblique rays become relevant, simulated images deviate from experiments (Fig. S6C). Adopting an approach developed by Goy [8], a more clear optical image for the thick cell case is given in Fig. S38.

Pages 44–45 in Sec. 6.2.5

For the Jones-matrix approach, dark areas correspond to the regions where the polarized ray in the layer mostly passes areas where the director is either parallel or orthogonal to the polarization. Therefore, a disclination can be indirectly seen if the director field above the line satisfies the mentioned condition. As lines look very thin, the director should have a narrow range where this condition is realized. In a Mauguin limit [15] for a very particular director distortion, the polarization can also keep the right orientation. The numerically obtained director field cannot disclose this, unfortunately. For thin samples, the Jones-matrix approach is good (Fig. S6A, B) while for the thicker ones where focusing, ray deflection, and oblique rays become relevant, the defect cores are brighter and the director field is less clear to see (Fig. S6C).... The open-source code can give better results in the thick cell limit, as shown in Fig. S38.

Fig. S38 on Page 45

Fig. S38 Comparison of optical images. Experimental optical image (A) and simulated optical image (using Nemaktis [8]) (B) in the W-state from the 1D cusp-like splay-bend pattern. Scale bar: 50 μm .

There is another problem which causes possible troubles. The numerical solution for the nematic structure is scaled for a factor of ~ 300 . There is no problem with the director field. The problem is that the defect core with a depressed order parameter cannot be simply scaled. It would be nice to test how relevant this is for POM. For a qualitative description, it is certainly not relevant but for optical applications, it is certainly relevant. It would be good to comment on it at some point.

We agree with the Reviewer that the defect core size cannot be scaled. Indeed, if we inspect Fig. 3H, 3I, S6, and S11, we can see that the defect lines are thinner in the experimental optical images than in the simulated optical images, regardless of which optical simulation method we use. Because in the simulation, the ratio between the defect core size and pattern periodicity $\frac{\xi_N}{L} = 1/30$, which is much smaller than 1, the presence of the disclinations does not impact the bulk nematic structure, as evidenced by the good match between experimental and simulation images. There are indeed some subtle differences between the two images, which are attributed to their difference in size. Therefore, the defect size effect should be taken into account for future optical applications. We have now added this additional discussion in the revised Supplementary Information below.

Changes in the Supplementary Information:

Page 39 in Sec 6.2.1

Because the system size in simulation is much smaller compared to the experiment, while the defect core size cannot be scaled, the defect lines appear thicker in the simulated optical images than in the experimental optical images, regardless of what optical simulation method we use, as seen in Fig. 3H, 3I, Fig. S6, and S11. Because in the simulation, the ratio between the defect core size and pattern periodicity is $\xi_N/L = 1/30$, which is much smaller than 1, the presence of disclinations does not impact the bulk nematic structure, as evidenced by the good match between experimental and simulated optical images. There are indeed some subtle differences between the two images, which are attributed to their difference in size. Therefore, the system size effect may play a role in the optical applications of nematic moirés. This is beyond the scope of the current study and we leave it for future work.

Page 7-8

The analysis of topology of disclinations on these pages refers to the introduction of disclination topology in the Materials and Methods. That text is fine for specialists but a broader readership needs a graphical presentation of all disclination profiles with typical alpha and beta angles that appear in this paper. It should go beyond what is now in Fig 1 G, H. The new figure should probably go in the Supplement or Materials and Methods if allowed.

Following the Reviewer's suggestion, a new Fig. S3 illustrating α and β shown in below has been added to the revised Supplementary Information.....

In appropriate points in the main text, Supplementary Information 2.4 and Figs S3, S4, and S5 should be mentioned.

We thank the Reviewer for this suggestion. Supplementary Information 2.4 and Fig. S3, S4 and S5 are now mentioned below in the revised main text:

Changes in the main text:

Page 6 Line 117–121

The schematics of different local profiles of disclination lines are given in Fig. S3. When $0.3 < H/L < 0.8$, curved line defects (C-state) appear as equispaced, aligned curves, each of which resembles a 3D helix and appears as a wavy line viewed in the z-direction (Fig. 1D, Fig. S4). The handedness of the C-state defects can be reversed by changing the rotation direction (Supplementary Information 2.4 and Fig. S5).

Page 8–9 Line 183–185

The director profiles of these disclinations can be characterized by two angles (51, 52), namely a twist angle $\beta \in [0, \pi]$ and a “phase shift” angle $\alpha \in [0, 2\pi)$ (Materials and Methods, Supplementary Information 2.4, Fig. S3) (2, 31, 32).

Page 9 Line 189–190

Our calculations show that line defects in the S- and W-states are of pure-twist type with $\beta \equiv \pi/2$ (Fig. 1C, 1E, Fig. S4).

Page 9 Line 195–197

However, for the two groups of parallel defect lines in the W-state, the local director field along the curve tangent is nearly constant (Fig. S4C, top plane and bottom plane).

Page 9 Line 198-200

Within the same group, all the parallel line defects share the same local profile and, therefore, the constant angle α . In the C-state, the twist angle β of the local winding varies along the defect curve (Fig. 1D, G, Fig. 2B, Fig. S4B): ...

Page 10

In the previous case the lines do not change much when psi angle is small but their separation decreases with increasing psi angle. How is here when psi angle starts from zero? Probably increasing psi loops enter the observation window from the infinity and become closed when they are small enough to fit in the window. In simulations it is important what you do with boundary conditions. It would be good to show how the aspect ratio change with growing psi? All this should be explained in more details in the supplement using theory or simulations!

Topological charge of the loop should be mention. Here loops should have charge zero so that can disappear if not prevented by surroundings.

We thank the Reviewer for these suggestions. We have calculated the aspect ratio of the series of simulations with $H/L=0.1$ in the revised Supplement Information. As we increase the angle Ψ from 0° , the defects first enter the simulation window around $\Psi=6^\circ$, as shown in Fig. S14A....

At appropriate points in the main text, Supplementary Information 3.2 & 3.3 and Figs. S12, S13, & S14 should be mentioned.

Following the Reviewer's suggestion, Supplementary Information 3.2, 3.3 and Figs. S12, S13, & S14 are now mentioned in the section **1D sinusoidal splay-bend pattern generates a 2D lattice of ring defects** in the revised main text as detailed below.

Changes in the main text:

Page 12 Line 254–260

As the cell gap increases, both simulation and experiment show that smaller defect loops (Loop-I) shrink and eventually self-annihilate, and larger loops survive (Fig. 3E, G, Fig. S12, S13, Movie S9). Simulations also find that the surviving defect loops gradually switch from a more round-like shape (Loop-II, Fig. 3E, F, Fig. S13B) to a rhombohedral shape (Loop-III, $H/L \in [0.2, 0.6]$, Fig. 3E, G, Fig. S13C) in the planar view. This transition with an out-of-plane buckling is akin to the transition from the S-state to the C-state in the 1D cusp-like pattern (Fig. S12, S13).

Page 12 Line 265–270

For even thicker cells ($H/L > 0.6$), a web-like defect is formed (Fig. 3E), akin to the thick-cell results of the cusp-like pattern (Fig. S4C, S12F). As we increase the angle Ψ from 0, the defects first enter the simulation window around $\Psi = 6^\circ$, as shown in Fig. S14A. Then we observe one array of closed defect loops at $\Psi = 12^\circ$ (Fig. S14B) and multiple arrays of defect loops with a distance of T after $\Psi \geq 24^\circ$. As Ψ increases, the aspect ratio of the loops decreases, and the loops become rounder as seen in the simulation and theory (Supplementary Information 3.3, Fig. S14I).

Page 13 Line 283–284

...whereas the end regions along the long axis of a Loop-II have a $-1/2$ wedge-like profile ($\beta > \pi/2$, Fig. 3F, S12, S13).

Page 13 Line 291–292

The topological charges of the three kinds of loops are all zero (Supplementary Information 3.2) and they can self-annihilate if not prevented by laden particles.

Page 11

Also, here the same remark as for Pages 7-8 applies!

We have now mentioned Fig. S3 in the revised manuscript.

Changes in the main text:

Pages 12–13 Line 273–275

...its β profile continuously varies from pure-twist ($\beta = \pi/2$), to +1/2 wedge ($\beta = 0$), then back to pure-twist ($\beta = \pi/2$), to –1/2 wedge ($\beta = \pi$), and finally returns to pure-twist ($\beta = \pi/2$) (Fig. S3A).

Page 12

The presented surface defects require strict planar anchoring. How strong it is in the experimental setup and what was used in simulations?

We thank the Reviewer for this question. In the experiment, the anchoring strength is in the order of $\sim \llbracket \llbracket 10 \rrbracket^{-3} \cdot 10 \rrbracket^{-4} \text{ J/m}^2$ (strong anchoring) (48). The anchoring strength W is set to be infinite in the simulations (the surface preferred director n_{sis} fixed) to approximate the strong anchoring condition in the experiment. The good match between the experiment and simulation in different scenarios validates the strong anchoring assumption used in the simulation. We have added the above clarifications in the revised manuscript.

This is OK if there are no surface defects! See remark related to page 19

We thank the Reviewer for this comment. As we have answered in the following, we set the surface points in the simulation to have equilibrium bulk order parameter S_0 . Because we do not evolve the \mathbf{Q} -tensor of the surface points and all the emerging defects are essentially bulk defects, there is no divergence issue in the simulation. We have added some comments in the **Materials and Methods** section of the main text below:

Changes in the main text:

Page 26 Line 569–572

Infinite (fixed) anchoring condition is applied to approximate the strong anchoring condition in the experiment (74). The surface points in the simulation are set to have constant equilibrium bulk order parameter $S_0 \cong 0.62$. We monitor the free energy as a function of simulation steps (Fig. S34) and do not find any divergence issue (Supplementary Information 6.2.1).

We have also included this additional discussion on in the Supplementary Information below:

Changes in the Supplementary Information:

Page 38 (last paragraph), Sec 6.2.1

We set infinite (or fixed) anchoring condition for both top and bottom substrate in the simulation to approximate the strong anchoring condition in the experiment. The patterned surface in the experiment has a well-defined easy-axis pattern, which becomes discontinuous at defect points (Fig. S34). Therefore, we set all the surface points to have

the equilibrium bulk value $S_0 \cong 0.62$. The \mathbf{Q} -tensor of the surface points is not evolved in the simulation. Bulk points next to the discontinuity points on the surface will have depressed order parameter due to the diverging director field. The disclinations that emerged in the simulation are essentially bulk line defects, with their ends connected to those discontinuity points. We further show that we do not encounter divergence issue in the simulation. For example, the ± 1 defect pattern at $\Psi = 20^\circ$ exhibits free energy convergence (Fig. S34A). The good match between the experiment and simulation (Fig. 5I, J) further validates our simulation method.

Fig. S34 on Page 39

Fig. S34 More details of ± 1 defect pattern. (A) Free energy convergence process in the ± 1 defect pattern simulation (at $\Psi = 20^\circ$). (B) Scanning electron microscope (SEM) experimental image of a +1 defect pattern adopted from [16]. Scale bar: 30 μm .

Page 13

More details are needed for $\psi=0$ structures: There are plus and minus defects and must be a coherence with neighbors how disclinations form quasi loops. It would be good to add a simulation with disclination loop profiles. What happens with a nematic between loops and surface defects where order is depressed?

We have modified Fig. S17 and added more details about the quasi-loop defects in the Supplement Information 4.2.We would like to also thank the question about the nematic between loop defects and surface defects. As shown in Fig S17C-E,...

Also, here including hints about Supplementary Information 4.2. and Fig S17 in the main text should be done.

We have now mentioned Supplementary Information 4.2 and Fig. S17 in the revised manuscript detailed in below.

Changes in the main text:

Page 15 Line 321–324

...and as shown in Fig S17C-E, a nematic between loops and patterned surface defect core positions has depressed order and a quasi-loop that connects two +1 defect cores and two -1 defect cores has $\beta \in [0, \frac{\pi}{2}]$ and $\beta \in [-\frac{\pi}{2}, 0]$, respectively (Supplementary Information 4.2).

Page 19, it applies also to page 22

Here I repeat: Anchoring strength for in-plane and out of plane deviations should be presented as it certainly strongly effects experimental structures in thin samples. What was assumed in simulations?

In the experiment, we applied strong anchoring and we don't evolve the n_s (the surface molecular orientation) in the energy minimization calculation to approximate the strong anchoring condition in the experiment (70).

Listing W values of the nematic anchoring strength and stressing that for simulation of the director fields the approximation of infinite W is good enough is OK only if there are no surface defects. The problem is the nematic order parameter S . If you assume that it is constant, local free energy diverges and the approximation of infinite W does not work anymore. Usually, finite W is used but it is true that this substantially complicates the numerical procedure. A possible (not elegant) way out is assuming that in the defect core S is zero. It should be clearly explained what was done here, probably in Methods and Supplementary.

We thank the Reviewer for this comment. In the experiment, the easy axis is locally imprinted on the surface through optical patterning. In the simulation we follow the experiment by assuming infinite anchoring condition and the surface order parameter is set to the equilibrium bulk value S_0 . Therefore, the bulk point next to the surface defect position has a depressed order parameter due to the diverging director field. The disclinations emerged in the simulation are bulk line defects, with their ends connected to easy-axis discontinuity positions. Because we use fixed anchoring condition, we do not evolve the \mathbf{Q} -tensor of the surface points. Our approach does not have divergence issue as the free energy can converge in simulations. We have included the missing information in simulation and explain the convergence issue in the revised manuscript.

We have added the following comments in the revised **main text**:

Changes in the main text:

Page 26 Line 569–572

Infinite (fixed) anchoring condition is applied to approximate the strong anchoring condition in the experiment (74). The surface points in the simulation are set to have constant equilibrium bulk order parameter $S_0 \cong 0.62$. We monitor the free energy as a function of simulation steps (Fig. S34) and do not find any divergence issue (Supplementary Information 6.2.1).

We have also included this additional discussion on **in the Supplementary** below:

Changes in the Supplementary Information:

Page 38 (last paragraph), Sec 6.2.1

We set infinite (or fixed) anchoring condition for both top and bottom substrate in the simulation to approximate the strong anchoring condition in the experiment. The patterned surface in the experiment has a well-defined easy-axis pattern, which becomes discontinuous at defect points (Fig. S34). Therefore, we set all the surface points to have the equilibrium bulk value $S_0 \cong 0.62$. The \mathbf{Q} -tensor of the surface points is not evolved in the simulation. Bulk points next to the discontinuity points on the surface will have depressed order parameter due to the diverging director field. The disclinations that emerged in the simulation are essentially bulk line defects, with their ends connected to those discontinuity points. We further show that we do not encounter divergence issue in the simulation. For example, the ± 1 defect pattern at $\Psi = 20^\circ$ exhibits free energy convergence (Fig. S34). The good match between the experiment and simulation (Fig. 5I, J) further validates our simulation method.

Fig. S34 on Page 39

Page 20-21 and 22

The paper is quite long therefore the Section "Theoretical model of numerical simulations" should be more compact as the approach is also well known. The technical details like different elastic constants and Frank form should go to the Supplement.

We thank the Reviewer for this advice. We have moved more technical details in the "Theoretical model of numerical simulations" to the Supplement (6.1.1).

This is OK

Further in this segment, I comment according to the lines of the revised text:

522: Add here an expression for the nematic coherence length that now appears later. Additionally, you must stress that flow effects are neglected and you will follow only relaxation of the orientational ordering.

We have included this additional information in the revised manuscript.

Changes in the main text:

Page 24 Line 545–546

In the simulations, we neglect flow effects and focus on the thermodynamic relaxation of the nematic order.

537-539: Here I repeat the above question related to Page 19.

We thank the Reviewer for raising this question again, please see our response in the above (page 10–11 of this response letter).

542-543: Here you should describe that Γ_s relates to the well-known Leslie's viscosities and sets up the typical timescale of the dynamical processes in the nematic at a selected length scale. From Eq.8 usually, a simple expression for a characteristic time is obtained: the ratio of the square of the particular length divided by $\Gamma_s \times L_1$.

Following the Reviewer's suggestion, we have now added this additional information about Γ_s in the revised manuscript:

Changes in the main text:

Page 26 (below Eq.7) Line 568–569

... Γ_s is a relaxation constant and is related to the rotational viscosity γ_1 via $\Gamma_s = 2S_0^2/\gamma_1$ (75), and t is time.

Page 27 Line 584–585

... $\gamma_1 = 0.078 \text{ Pa}\cdot\text{s}$...

The discussion about simulation time scales is also presented in the revised manuscript:

Changes in the main text:

Page 26 (last paragraph), Line 573–575

There are two characteristic time scales associated with the nematic. One is the nematic relaxation time $\tau_{LC} = \frac{\gamma_1 \xi_N^2}{L_1}$ with γ_1 defined in the above. The other time scale is the relaxation time associated with the nematic cell (75): $\tau_{cell} = \frac{\gamma_1 H^2}{L_1} = \frac{2S_0^2 H^2}{\Gamma_s L_1}$, with H being the cell height.

546-547: On page 6 etc. I have stressed that there is a problem with scaling, which is OK for the director field but could not be simply applied to the defect cores with a depressed order parameter. This should be commented

We thank the Reviewer for raising this comment again, please see our response in the above (page 4–5 of this response letter).

547-548 Describing the selection of lattice point distance related to the periodicity is not

enough. The best would be to present the size of the complete simulation box with lattice point distances in all three orthogonal directions. Here also boundary conditions should be explained and the expected weakness of the solution close to free boundaries. It looks like they are cut in the presented results. More details for all structures should be in the Supplement 6.2.1.

We have included the following descriptions for the boundary conditions and box size with an additional figure in the revised **Supplementary Information**.

Changes in the Supplementary Information:

Pages 37–38 in Sec 6.2.1

We do not impose periodic boundary conditions since the rotated pattern does not have translational symmetry. We instead use free boundary conditions (zero anchoring) for the sidewalls, which will inevitably induce inhomogeneities near the sidewalls. This boundary effect does not impact the nematic structure near the center of the simulation box. When showing the defect configurations, we manually cut the distorted regions near the sidewalls. In Fig. S33 we show the actual simulation box and the cropped box.

For the 1D cusp-like splay-bend pattern and 1D sinusoidal splay-bend pattern, the box width direction has 420 lattice points (N_x), and the length direction has 600 lattice points (N_y). The thickness is varied between 3 and 40 lattice points. We fix lattice constant L to be 30 points except for the electric field results (the parameters are in the caption of Fig. S26). The size of the cropped box is 300 by 300 in the $x - y$ dimension for the 1D cusp-like and sinusoidal pattern. The simulation results without cropped box can be found in Movie S1. For the 2D ± 1 defect pattern, there are 18 periodic units (Fig. 5A) along both x and y direction in the simulation box, with $H/L = 0.3$ and $L = 30$ lattice points. The simulation results without cropping are shown in Movie S11. In Fig. 5D and 5I, the cropped box size is 13 by 13, 7 by 7, respectively.

Fig. S33 on Page 38

Fig. S33 The full simulation box for the 1D cusp-like and sinusoidal pattern with the cropped box shown as a dashed box. Except for the electric field calculations (Sec. 5 in the Supplementary Information), $N_x = 420$ and $N_y = 600$ in, and H is varied between 3 to 40. Pattern constant L is set to 30. The cropped box has 300 lattice points in the x dimension and 300 lattice points in the y dimension.

We have also added a hint on in the **main text**.

Changes in the main text:

Page 24–25 (before Eq.1) Line 546–548

The simulation box dimensions are shown in Fig. S33, and considering the boundary effect, the results are shown in a slightly cropped box (Supplementary Information 6.2.1).

548-553: The discussion on time scales does not look OK. Including undefined terms, not clearly describing dimensionless quantities, mentioning Lattice Boltzmann simulations used for flows, etc. Using the above-mentioned expression for times related to characteristic scales: nematic coherence length, cell thickness, and lateral system size should be related to the needed time interval and number of repetitions. Usually time intervals are a few times shorter than the nematic coherence time. In periodic structures starting with small perturbations computing time must surpass the time related to the layer thickness, while starting from a random state would require times longer than the one related to the lateral size.

We thank the Reviewer for this comment. We have removed unrelated terms and have rewritten the discussion about simulation units in the revised manuscript as follows:

Changes in the main text:

Page 26 (last paragraph), Line 573–581

There are two characteristic time scales associated with the nematic. One is the nematic relaxation time $\tau_{LC} = \frac{\gamma_1 \xi_N^2}{L_1}$ with γ_1 defined in the above. The other time scale is the relaxation time associated with the nematic cell (75): $\tau_{cell} = \frac{\gamma_1 H^2}{L_1} = \frac{2S_0^2 H^2}{\Gamma_s L_1}$, with H being the cell height. In simulation units, $\tau_{LC} \approx 7.7$ and $\tau_{cell} \approx 623$. The simulation time step $\Delta t = 0.5$ is more than ten times smaller than τ_{LC} and the total simulation time $\tau_s \gg \tau_{cell}$ to ensure that our simulation can converge to free energy minimum. For rotation operation simulations, we choose the total simulation time $\tau_s \gg \tau_{cell}$ to approximate a quasi-static process. We have also checked the nematic structure during rotation if $\tau_s \gg \tau_{cell}$ is not satisfied (Movie S13).

554: Here it would be good to mention that the finite difference relation approach was used. Your code or public code etc. Some details can go to the supplement.

We thank the Referee for this comment. We have added this information on Page 26 in the revised manuscript as follows:

Changes in the main text:

Page 27 Line 582–585

The simulation is conducted via a finite difference approach, as described in (61). Following (61), our simulation can be mapped to the nematic 4-Cyano-4'-pentylbiphenyl at room temperature and atmospheric pressure by choosing $\xi_N \approx 6.63$ nm, $A_0 = 1.17 \times 10^5$ J/m³, $\gamma_1 = 0.078$ Pa·s, and $\epsilon_a = 11.5$. This gives rise to $\tau_{LC} \approx 0.667$ μ s and $\tau_{cell} \approx 54$ μ s.

Pages S33-S34

One usually adds quasi thermal noise to reduce such effects.

We thank the Reviewer for this comment. We have used different initial conditions including a random initial configuration, and compare their free energies instead. We have added the discussions about quasi-thermal noise in Supplement (6.2.2). ... In our simulation with electric fields, we added random noise to the initial director field (very small, nonzero component n_z) to study the Frederiks transition, which shows a good match with the theory (Section 5 in SI). Without the initial noise, the transition requires a four times larger transition threshold in the simulations.

In Supplementary 5 this is not added in detail so that one cannot reproduce the procedure.

We thank the reviewer for this comment. We have the following information **In section 5.2 on Page 31 in the revised Supplementary Information:**

Changes in the Supplementary Information:

Pages 31–32 in Sec 5.2

The relaxed structure in the absence of an electric field all have $n_z = 0$ component since (1) we apply the initial condition with $n_z = 0$, and (2) our anchoring condition is planar. Therefore, we need to add an initial noise to the director field before applying the electric field. Specifically, we have added a small random number Δn_z uniformly distributed in $[-10^{-10}, 10^{-10}]$ to the z component of all the bulk LC points and normalize the director \mathbf{n} , to allow the Frederiks transition to happen.

Reviewer #2 (Remarks to the Author):

In the revised manuscript, the authors have fully addressed the comments, questions, and suggestions of four reviewers. The title, many parts of the main text, the list of references, the supplementary information, and the videos have been substantially revised and clarified. I thank the other reviewers for their accurate and comprehensive reports, which helped the

authors improve the manuscript in many ways. I also appreciate the authors' efforts to review the effects of anisotropy of elastic constants, the use of an additional open-source package for simulating optical patterns, and their provision of clear explanations of noted deficiencies. The authors are aware of the limitations of their approach, and on the other hand, they also suggest how to use the moiré effect to address the inverse problem of designing arbitrary pixelated defect regions. I have no further comments or suggestions for improvement, and I think the manuscript is good enough to be published in Nature Communications in its present form.

We thank Reviewer #2 for the approval.

Reviewer #3 (Remarks to the Author):

The authors have answered all comments and questions that were raised in my previous report and I recommend acceptance of this manuscript in its present form.

We also thank Reviewer #3 for the approval.

Reviewer #4 (Remarks to the Author):

The authors have addressed many of the concerns raised by the four referees including me. However, there are still many problems in the presentation (several issues should have been commented in the 1st report), and therefore I do not recommend the publication of this manuscript in its present form.

As one of the referees says, this manuscript is too long. I do not accept the authors' mere statements "something has been added in the revised manuscript" that let the referees look through the whole 42-page manuscript or 44-page supplementary material.

We would like to thank Reviewer #4 for the criticism, and we apologize for the inconvenience we have incurred. We hope the newly included statements and discussions are easy to find and we hope the revised version can reach the standard of Nature Communications.

** Present tense ("enables") is natural for the title.*

Following the Reviewer's suggestion, we have now changed the title to "Moiré effect enables versatile design of topological defects in nematic liquid crystals".

** [Page 10] A_{xy} is referred to as "projected helix diameter" in lines 208-209, and "amplitude of the defects" in line 226. This could be confusing.*

We thank the Reviewer for this comment. We have made changes and referred A_{xy} to "projected helix diameter" .

Changes in the main text:

Page 11 Line 232-233

The **projected helix diameter** A_{xy} in the C-state increases as Ψ decreases (Fig. 2F, Movie S6)...

** [Page 15, lines 330-331] The statement " $\tan^{-1}(3/4)$ is a singular point in the moiré theory." does not practically say anything. What results from the fact that $\tan^{-1}(3/4)$ is a singular point? I also note that it has been already stated in lines 326-327 that $\tan^{-1}(3/4)$ is a special angle.*

We thank the Reviewer for this comment. In the moiré theory (see Sec. 3.4 in Amidror, I., 2009. The Theory of the Moiré Phenomenon: Volume I: Periodic Layers (Vol. 38). Springer Science & Business Media), a singular point has an infinitely large period T (see the divergent black curve in the blue region of Fig. 5G). According to the calculations detailed in Sec. 4.4 in the Supplementary Information, $\Psi = \tan^{-1}(3/4)$ is a singular point in the two-dimensional moiré pattern. In our nematic moiré, we call $\Psi = \tan^{-1}(3/4) \approx 36.8^\circ$ a special angle because, at this rotation angle, periodic defects can be generated from the periodic overlapping of the ± 1 defects, as shown in Fig. 5I, J, which cannot be explained by the diverging moiré periods. We use \tilde{T} to denote that period (different from T), as shown in Fig. 5I.

We have included more explanations and the above reference as Ref. 28 in the main text.

Changes in the main text:

Page 15-16 between Line 342–348

Note that $\tan^{-1}(3/4)$ is a singular point in the moiré theory (divergent black curve in the blue region in Fig. 5G) (28) with an infinitely large moiré period T (see the calculations in Supplementary Information 4.4). In the simulations and experiments, the defect structures are found aperiodic for $\Psi \in [34^\circ, 40^\circ]$ except $\Psi \approx 36^\circ$ (Fig. S21). The emerging defect period \tilde{T} and angle $\tilde{\omega}$ of the superlattice around the rotation angle $\Psi \approx 36^\circ$ agree between the experiments and simulations, and can be explained by the Pythagorean triple theory (Fig. 5I, Table 1, Fig. S22), rather than the moiré theory.

We have also modified the above-mentioned reference (Ref. 7) in **Sec. 4.4 in the revised Supplement Information** for clarification.

Changes in the Supplementary Information:

Page 25 (last paragraph) in Sec 4.4

This is consistent with the geometric theory of the moiré pattern, which shows **that there is no dominating frequency for $\Psi \in [21^\circ, 34^\circ]$ and the frequency of the $(1,2,-2,-1)$ -moiré becomes comparable with that of the $(1,0,-1,0)$ -moiré for $\Psi \in [34^\circ, 40^\circ]$ (Fig. S20) [7], frustrating the nematic and rendering it a less ordered defect structure. As $\Psi \in [34^\circ, 40^\circ]$, the dominating moiré is $(\pm 1, \pm 2, \mp 2, \mp 1)$, the higher order moiré (2nd order).**

Page 47 in References

[7] Amidror, I., 2009. The Theory of the Moiré Phenomenon: Volume I: Periodic Layers (Vol. 38)}. Springer Science and Business Media. See 3.4.1 for the theory of two superposed square lattices.

* [Materials and Methods] The authors do not mention anything about from which company they bought JXLC-20000 (or how they synthesized it), or why they used it instead of 5CB in confocal microscope observation.

We thank the Reviewer for this comment. JXLC20000 is a commercialized nematic mixture purchased from *Grandinchem, INC*. The chemical structures of the components are shown below and has been added as Fig. S30B. The reason we used JXLC20000 instead of 4'-pentyl-4-cyanobiphenyl (5CB) in Fig. 1 is due to the low birefringence of JXLC20000 which will reduce the scattering during the confocal imaging process. Birefringence of JXLC20000 is about 0.05 and the birefringence of 5CB is about 0.2.

Changes in the main text:

Page 21 Line 465

JXLC20000 is a commercialized nematic mixture purchased from *Grandinchem, INC* (Fig. S30).

Page 23 Line 511–515

The birefringence for JXLC-20000 is about 0.05 and the birefringence for 5CB is about 0.2. Hence, the color of images is different. Both liquid crystal materials of 5CB and JXLC-20000 are used in the experiments. JXLC-20000 is used in Fig. 1, and 5CB is used in the rest of the experiments.

Page 24 Line 530–532

Due to the low birefringence of LC JXLC-20000, it is used in the confocal measurement to reduce the scattering effect during the imaging process. The BTBP solution is then mixed with JXLC-20000 by a weight ratio of 1:1.

Changes in the Supplementary Information:

Page 35 in Sec 6.1

* [Page 23] It is well known that biaxiality arises in the vicinity of the defect core when the orientational order is simulated by a second-rank tensor (See e.g., Lyuksyutov, *Sov. Phys. JETP* 48, 178 (1978); Schopohl and Sluckin, *Phys. Rev. Lett.* 59, 2582 (1987); Penzenstadler, and Trebin, *J. Phys.* 50, 1027 (1989)). I agree with the authors that the order is practically uniaxial in the bulk. Nevertheless, if they are doing calculations with the constraint of $P=0$, their calculations are different from standard ones carried out in previous studies such as Ref. 61.

We thank the Reviewer for this comment. We do not force $P = 0$ in our simulation. We have included these references (now Ref. 71-73) in the Methods section and modified the discussion in the revised manuscript.

Changes in the main text:

Page 25 (below Eq. 3) Line 555–556

... Using the above Doi expression for f_{LDG} , biaxiality P is practically 0 in defect-free regions (71-73).

Page 31 Line 777–782

71. I. Lyuksyutov, Topological instability of singularities at small distances in. *Sov. Phys. JETP* 48(1), 178-179 (1978).
72. E. Penzenstadler, H.-R. Trebin, Fine structure of point defects and soliton decay in nematic liquid crystals. *Journal de Physique* 50, 1027-1040 (1989).
73. N. Schopohl, T. Sluckin, Defect core structure in nematic liquid crystals. *Physical review letters* 59, 2582 (1987).

* [Pages 24-25, eq. (6)] I understand that the Rapini-Papoular form of the surface energy is in fact not used in their calculations if W is assumed to be infinite.

We agree with the Reviewer. We have removed the introduction of f_s on Page 25 in the revised manuscript and only retained the introduction of surface nematic order parameter \mathbf{Q}_s .

Changes in the main text:

Page 24 Line 542

~~Anchoring energy induced by patterned substrates is counted by the Rapini–Papoular form (61).~~

Pages 25–26 Line 560–565

~~We use the nondegenerate Rapini–Papoular form of the surface anchoring energy density f_s (61)~~

~~$$f_s = \frac{\pm}{2} W (\mathbf{Q} - \mathbf{Q}_s)^2, \quad (6)$$~~

~~where \mathbf{Q}_s is the patterned surface field defined as $\mathbf{Q}_s = S_0 (\mathbf{n}_s \mathbf{n}_s - \frac{1}{3})$ with \mathbf{n}_s the imposed surface orientation, and W is the anchoring strength, which is set to be infinite in the simulation (\mathbf{n}_s is fixed)~~

** [Page 25] The authors say that τ_0 (and hence τ_s) does not have physical meaning. But still they compare τ_s with τ_{cell} ; the latter is given by physical parameters. This comparison does not make sense.*

We are sorry for confusing the Reviewer. We are comparing the total simulation time τ_s to a nematic relaxation time τ_{cell} . We have rewritten the discussion of the time units in the revised manuscript as follows:

Changes in the main text:

Page 26 Line 568–569

~~... Γ_s is a relaxation constant and is related to the rotational viscosity γ_1 via $\Gamma_s = 2S_0^2/\gamma_1$ (75), and t is time.~~

Page 26 (last paragraph) Line 573–581

~~There are two characteristic time scales associated with the nematic. One is the nematic relaxation time $\tau_{\text{LC}} = \frac{\gamma_1 \xi_{\text{N}}^2}{L_1}$ with γ_1 defined in the above. The other time scale is the relaxation time associated with the nematic cell (75): $\tau_{\text{cell}} = \frac{\gamma_1 H^2}{L_1} = \frac{2S_0^2 H^2}{\Gamma_s L_1}$, with H being the cell height. In simulation units, $\tau_{\text{LC}} \approx 7.7$ and $\tau_{\text{cell}} \approx 623$. The simulation time step $\Delta t = 0.5$ is more than ten times smaller than τ_{LC} and the total simulation time $\tau_s \gg \tau_{\text{cell}}$ to ensure that our simulation can converge to free energy minimum. For rotation operation~~

simulations, we choose the total simulation time $\tau_s \gg \tau_{\text{cell}}$ to approximate a quasi-static process. We have also checked the nematic structure during rotation if $\tau_s \gg \tau_{\text{cell}}$ is not satisfied (Movie S13).

Page 27 Line 582–587

The simulation is conducted via a finite difference approach, as described in (61). Following (61), our simulation can be mapped to the nematic 4-Cyano-4'-pentylbiphenyl at room temperature and atmospheric pressure by choosing $\xi_N \approx 6.63$ nm, $A_0 = 1.17 \times 10^5$ J/m³, $\gamma_1 = 0.078$ Pa·s, and $\epsilon_a = 11.5$. This gives rise to $\tau_{\text{LC}} \approx 0.667$ μ s and $\tau_{\text{cell}} \approx 54$ μ s. In the simulation, surface pattern period $L = 30$ in simulation units, corresponding to ~ 199 nm, which is compared to 75 μ m in the experiment.

** [Confocal images in Fig. 1] A fundamental issue is not addressed: What should appear dark (or bright) in their confocal images? The difference between the panels N and O should be clearly stated in the caption or the main text.*

We than the Reviewer for this question. Fluorescence molecules are mixed with liquid crystals, and they will accumulate in the disclination lines and emit the fluorescence light (see also T. Ohzono et al., Scientific Reports 6, 36477 (2016)). In our experiments, imaging was performed with an excitation wavelength of 488 nm and an emission wavelength of 530 nm. Due to the fluorescence molecules in the disclination lines, the lines are visualized as bright lines. In the experiment, the disclination lines are arranged into two separating layers in the W-state. Hence, using confocal microscopy to scan the cell from top to bottom, there are four disclination lines (1st group) along the same direction first near the top substrate, which are bright lines (Fig. 1L). With scanning along the z-axis, another four thin disclination lines (2nd group) show up near the bottom substrate in another orientation (Fig. 1M). We can also observe the two groups of disclinations at the same time in the 3D rotation views (Fig. 1N, O). The lines close to the top substrate are the bright lines as shown in Fig. 1N. When visualized from the bottom of the cell in Fig. 1O, both two groups of lines are shown due to the scattering effect, and the 2nd group of lines close to the bottom substrate looks gray because of a lower contrast.

We have added the following sentences in the revised main text on Page 7 and Page 34.

Changes in the main text:

Page 7 Line 147–153

Near the top substrate, there are four disclination lines (the first group) along the same direction, which are bright lines inside thick dark bands (Fig. 1L). With scanning along the z-axis, another four thin, gray disclination lines (the second group) show up near the bottom substrate in another orientation (Fig. 1M). Our configuration is similar to the two-layer configuration observed in (49) by the Confocal Microscopy. Therefore, when viewed from the z-direction (Fig. 1M), the two groups of defects appear as a web-like structure,

confirming the simulation prediction (Movie S3). 3D views of the web-like structure are presented in Fig. 1N, O (Movie S4).

Page 34 Line 813–821 in Fig. 1 Caption

Using confocal microscopy to scan the cell from top to bottom, we observe (L) disclination lines (first group) close to the top substrate and (M) the disclination lines (second group) appearing near the bottom substrate. (N)-(O) give two 3D rotation views of the sample to visualize the two groups at the same time. z_c is the spacing between the two groups in the scanning process, $z_c = 40 \mu\text{m}$. (N) When the two groups of disclinations are visualized from the cell top at the same time, the bright lines are the first group disclination lines, close to the top substrate. (O) Viewing the sample from the cell bottom, both two groups of lines are shown due to scattering effect and the second group of lines close to the bottom substrate is in gray because of lower contrast.

** [Fig. 5] The authors say that in Fig. 5B and D, the dashed orange (or blue) square corresponds to the same region. Obviously the relative position of the orange square with respect to the blue one is different.*

We thank the Reviewer for this criticism. We have modified Fig. 5B and 5D in the revised manuscript for clarification.

Fig. 5. 2D pattern defect structure characterization. (B) The mapping from the dot-screen pattern to the ± 1 defect lattice pattern (left) and the superposed 2D geometric square gratings formed at $\Psi = 5^\circ$. (C) Superposed 2D geometric moiré pattern at $\Psi \approx 36.8^\circ$. (D) Simulated defect structure in the nematic moiré at $\Psi = 5^\circ$. (E–F) Experimental bright field snapshots corresponding to two boxed regions in the dashed box in (B, D).

The explanation of the black theoretical lines in the blue regions of Fig. 5G and H is still missing (In the 1st report I erroneously mentioned the black lines as blue ones). They are not compared with simulations or experiments.

We thank the Reviewer for this point. We have further clarified the descriptions in the revised main text and the Supplementary Information (sec. 4.4).

In terms of convolution in the moiré theory, the dominating moiré for $\Psi \in [34^\circ, 40^\circ]$ is $(\pm 1, \pm 2, \mp 2, \mp 1)$, the higher order 2nd moiré. Comparing Fig. S20A and B, we can see that the 1st order moiré is more visible than the 2nd order moiré. In the simulations and experiments, we cannot find periodic defect structures shown in Fig. S21 following the moiré period T for $\Psi \in [34^\circ, 40^\circ]$ (this period T is plotted in the blue region in Fig. 5G). Therefore, we are not able to compare the moiré theory to the simulation and experiment in the blue region (i.e., $\Psi \in [34^\circ, 40^\circ]$) in Fig. 5G and Fig. 5H.

However, we do find periodic structure around $\Psi = \tan^{-1}(3/4)$ in simulations and experiments as shown in Fig. 5I, J. To distinguish this period from the moiré period, the period of the emergent defect structure is denoted by \tilde{T} . We have actually compared the Pythagorean triangle to the simulation and experiment in the blue region (around $\Psi = \tan^{-1}(3/4)$) in Fig. 5I and in Table 1.

We now made this clearer in the revised main text.

Changes in the main text:

Page 16 (1st paragraph) Line 344–348

In the simulation and experiment, the defect structures are found aperiodic for $\Psi \in [34^\circ, 40^\circ]$ except $\Psi \approx 36^\circ$ (Fig. S21). The emerging defect period \tilde{T} and angle $\tilde{\omega}$ of the superlattice around the rotation angle $\Psi \approx 36^\circ$ agree between the experiments and simulations, and can be explained by the Pythagorean triple theory (Fig. 5I, Table 1, Fig. S22), rather than the moiré theory.

Page 40–41 Line 918–920 in Fig. 5 Caption

... and blue is for the $\Psi \in [34^\circ, 40^\circ]$ regime with dominating moiré $(1, 2, -2, -1)$, with a singular point (infinitely large period) at $\Psi = \tan^{-1}(3/4)$.

Additional discussions are included on **Page 25** in the revised **Supplementary Information** in **Sec. 4.4** below:

Changes in the Supplementary Information:

Page 25 (last paragraph) in Sec 4.4

In the simulation and experiment, we cannot find periodic defect structures exhibiting the moiré period T (the period is plotted in the blue region in Fig. 5G) within $\Psi \in [34^\circ, 40^\circ]$ (Fig. S21). Therefore, we are not able to compare the moiré theory to the simulation and experiment in the blue region ($\Psi \in [34^\circ, 40^\circ]$) in Fig. 5G and Fig. 5H.

* [Supplementary PDF, Sec. 5.1] Three different symbols, L , d and H , have been used for the cell thickness.

We have modified the symbols. On Page 31 In the revised Supplementary we only use H for the cell thickness.

* Fig. S14 does not have panels I and J.

We appreciate the comment. We relabelled Fig. S14K to Fig. S14I on Page 20 in the revised Supplementary Information shown below:

Figure S14: **Defect structure at different Ψ ...** (I) The aspect ratio of Loop-I and Loop-II in simulations and theories.

* I recommend the authors to present Figs. S17D and E from a different viewpoint: I misunderstood that two close loops were located vertically.

We thank the Reviewer for this comment. We have modified Fig. S17D and E on Page 22 in the revised Supplementary Information as shown below.

Fig. S17: **Loop-like defect array at $\Psi = 0^\circ$ in the 2D defect pattern.** (D) shows the director field at the bottom plane ($z = 0$) and (E) gives the director field in the mid-plane ($z = H/2$).

** Fig. S23 is not referred to in the main text.*

** What one can learn from Fig. S25 is not at all mentioned.*

We thank the Reviewer for these comments. We have added more discussions on page 15 in the revised main text to mention Fig. S23 and Fig. S25.

Changes in the main text:

Page 16 Line 352–355

For the special angle $\Psi = \tan^{-1} 3/4$, however, the emergence of periodic structures comes from the overlapping of defects and therefore the defect configuration is sensitive to the choice of the rotation center (Fig. S23).

Page 17 Line 366–367

When the ratio is lower, there are more bulk defects than near-surface defects and the total volume of defects is lower (Fig. S25).

Additionally, we have also included a discussion for Fig. S25 in revised **Supplementary Information**:

Changes in the Supplementary Information:

Page 30 in Sec 4.7

Those quasi-loops in fact consist of two separate curves, as presented clearer in Fig. S25E and F. During substrate rotation, defect creation/annihilation may occur. However, since each defect core on the two patterns can generate two split curves, the total number of defect curves is constant during the rotation. Here we count the ratio between bulk defects (defect curve connecting both substrates) and near-surface defects (defect curve with its two ends on the same substrate), and plot it against the twist angle Ψ in Fig. S25. From Fig. S25B and C, we learn that when the rotation is sufficiently fast, the system doesn't have time to relax to satisfy the new anchoring pattern, and there are more near-surface defects. Since we apply $H/L = 0.3$ in the simulation, defects are shorter if they go through the bulk from top to bottom. In quasi-static simulations with the system remaining in the stable state, more defect lines choose to connect through the bulk (Fig. S25B, C), and as a result, its defect volume is lower than $\tau_s/\tau_{\text{cell}} = 35.4$ and $\tau_s/\tau_{\text{cell}} = 236$ (Fig. S25D). Here, τ_{cell} is the characteristic time scale for the relaxation of the nematic director field, $\tau_{\text{cell}} = (\gamma_1 H^2)/L_1 = (2S^2 H^2)/(\Gamma_s L_1)$ with γ_1 being the rotational viscosity, H being the cell gap, and L_1 being the elastic constant. The simulation time $\tau_s = N\tau_0$ is expressed in the simulation time unit τ_0 (N is the number of simulation steps). To achieve a quasi-static simulation, $\tau_s \gg \tau_{\text{cell}}$ has to be satisfied.

** [Fig. S26] If the cell thickness is the same for the three simulations as the authors say, the difference in the lattice constant must be clearly state there.*

The Reviewer is right. We have added the descriptions in the caption of Fig. S26 in revised Supplementary Information:

Changes in the Supplementary Information:

Fig. S26 caption on Page 32

The lattice constants (L) for the S-, C-, and W-state are $150 \times 6.63 \text{ nm} \approx 994.45 \text{ nm}$, $60 \times 6.63 \text{ nm} \approx 397.8 \text{ nm}$, and $30 \times 6.63 \text{ nm} \approx 198.9 \text{ nm}$, respectively.

** [Figs. S4, S17, S33, S34] The color scale should be for beta/pi (as in Fig. 1), not beta.*

Following the Reviewer's suggestion, we have replaced the colorbar for beta with beta/pi in Fig. S4, S17, S35 (old S33) and S36 (old S34) in the revised Supplementary Information.

** [Figs. S27-29] The "V" in "0.82V" should be roman, not italic.*

We thank the Reviewer for this comment. We have changed the format in Figs. S27–29 in the revised Supplementary Information.

** [Supplementary PDF, Sec. 6.2.5] The authors response "we did not artificially enhance the disclination lines in our simulated optical images" seems to contradict with the introduction of n'_E with a comment, "We apply this modification because the the Jones-matrix approach is not good in simulating thin disclination lines."*

The value of lambda is not given in the calculated POM images.

We thank the Reviewer for this point. The extraordinary index of refraction n'_E was chosen to be dependent on the local scalar nematic in order to improve the quality of the simulated optical images by the Jones-matrix approach. This may explain the disagreement between us and Reviewer #1 on the good quality of simulated images by the Jones-matrix approach. The system size in the simulation is smaller than in the experiment, and the wavelength λ is set to $3.8 \times 6.63 \text{ nm} \approx 25.194 \text{ nm}$ to obtain optical images from simulation results. We have also performed calculations using the open-source code suggested by Reviewer #1. For those results, we rescaled our data to the real experimental size and used the experimental wavelength, 400 – 800nm. We have included this additional information **in the revised Supplementary Information**.

Changes in the Supplementary Information:

Page 45 in Sec 6.2.5

In our simulation, the system size is smaller than in the experiment, and the wavelength λ is set to $3.8 \times 6.63 \text{ nm} \approx 25.194 \text{ nm}$ to obtain optical images based on the simulation data using the Jones-matrix approach. We have also performed calculations using the open-source code [8]. For those results, we rescaled our data to the real experimental size and used the experiment wavelength, 400 – 800 nm.

* *There are still typographical/grammatical errors:*

* *[page 2, line 30] "planck" -> "plank"? (or, "disk-like" is more commonly used in the community.)*

* *[page 15, line 334] "later" is not a verb.*

* *[page 25] The phrase "comparing to L=75um in the experiment" does not seem to be placed appropriately in this sentence.*

* *[Page 34, line 808] "Use" -> "Using"?*

* *[Supplementary PDF, page 17] "emergency" -> "emergence"?*

* *[Supplementary PDF, page 17] "they can all disappear" appears twice.*

* *[Supplementary PDF, page 31] The semicolon that follows "in the simulations" should be colon.*

* *[Supplementary PDF, page 41] "Flg. S4" -> "Fig. S4"*

We thank Reviewer #4 for pointing out these typographical/grammatical errors

Changes in the main text:

Page 2 Line 30

~~Planck~~-> disc-like

Page 16 Line 351

~~later~~-> alter

Page 27 Line 587

~~comparing to~~-> which is compared to

Page 34 Line 813

~~Use~~->Using

Changes in the Supplementary Information:

Page 17 in Sec 3.2

~~emergency~~-> emergence

~~they can all disappear~~

Page 31 in Sec.5

...in the simulations; therefore, Eq. S10 reduces to...->

...in the simulations. Therefore, Eq. S10 reduces to...

Page 43 in Sec.6.2.4

... the topological analysis of the S, C, and W-state in ~~Fig. S4~~, the ...->

the topological analysis of the S, C, and W-state in Fig. S4, the...

We would like to thank all the Reviewers again for their time and helpful suggestions to have greatly improved the quality of our manuscript.

REVIEWERS' COMMENTS

Reviewer #1 (Remarks to the Author):

The authors have done a great job. This time, they have improved numerous details of the paper and its supplement. The different parts are now better coupled, which allows the readers to comprehend numerous interesting details in a rather long text consisting of the main part and an extensive supplement. They have also done a great job of clearly presenting replies and related text changes for all reviewers. I agree with all the changes. I have observed only one misprint in the Supplement on page 10, third paragraph, where Goy instead of Poy is used for the author of Ref 8. I support the publication in Nat. Comm.!